# Quantifying the controls on potential soil production rates: A case study of the San Gabriel Mountains, California

Jon D. Pelletier

Department of Geosciences, University of Arizona, Gould-Simpson Building, 1040 East Fourth Street, Tucson, Arizona
85721-0077, USA

*Correspondence to*: Jon D. Pelletier (jdpellet@email.arizona.edu)

**Abstract.** The potential soil production rate, i.e., the upper limit at which bedrock can be converted into transportable material,
limits how fast erosion can occur in mountain ranges in the absence of widespread landsliding in bedrock or intact regolith.
Traditionally, the potential soil production rate has been considered to be solely dependent on climate and rock characteristics.
Data from the San Gabriel Mountains of California, however, suggest that topographic steepness may also influence potential
soil production rates. In this paper I test the hypothesis that topographically induced stress opening of pre-existing fractures in
the bedrock or intact regolith beneath hillslopes of the San Gabriel Mountains increases potential soil production rates in steep
portions of the range. A mathematical model for this process predicts a relationship between potential soil production rates
and average slope consistent with published data. Once the effects of average slope are accounted for, a small subset of the
data suggest that cold temperatures may limit soil production rates at the highest elevations of the range due to the influence
of temperature on vegetation growth. These results suggest that climate and rock characteristics may be the sole controls on
potential soil production rates as traditionally assumed, but that the porosity of bedrock or intact regolith may evolve with
topographic steepness in a way that enhances the persistence of soil cover in compressive-stress environments. I develop an
empirical equation that relates potential soil production rates in the San Gabriel Mountains to the average slope and a climatic
index that accounts for temperature limitations on soil production rates at high elevations. Assuming a balance between soil
production and erosion rates at the hillslope scale, I illustrate the interrelationships among potential soil production rates, soil
thickness, erosion rates, and topographic steepness that result from the feedbacks among geomorphic, geophysical, and
pedogenic processes in the San Gabriel Mountains.

*Keywords: soil production, cosmogenic radionuclides, topographically induced stress, San Gabriel Mountains*

## 1 Introduction

The potential soil production rate (denoted herein by $P_r$) is the highest rate, achieved when soil cover is thin or absent,
that bedrock or intact regolith can be converted into transportable material at each point on Earth's surface. $P_r$ values are the
rate-limiting step for erosion in areas where landsliding in bedrock or intact regolith is not widespread, because soil must be

produced before it can be eroded. Slope failures in bedrock or intact regolith are common in some fine-grained sedimentary rocks (e.g., Griffiths et al., 2004; Roering et al., 2005) but may be less common in massive lithologies such as granite.

Despite its fundamental importance, the geomorphic community has no widely accepted conceptual or mathematical model for potential soil production rates. Pelletier and Rasmussen (2009) took an initial step towards developing such a model by relating $P_r$ values in granitic landscapes to mean annual precipitation and temperature values. The goal of that model was to quantify how water availability and vegetation cover control the potential soil production rate across the extremes of Earth's climate. The Pelletier and Rasmussen (2009) model predicts $P_r$ values consistent with those reported in the literature from semi-arid climates, where $P_r$ values typically range from ~30-300 m/Myr. In humid climates, the Pelletier and Rasmussen (2009) model predicts $P_r$ values greater than 1000 m/Myr (Fig. 2A of Pelletier and Rasmussen, 2009). This is broadly consistent with measured soil production rates in the Southern Alps of New Zealand where the mean annual precipitation (MAP) exceeds 10 m (Larsen et al., 2014). The Pelletier and Rasmussen (2009) model was a useful first step but clearly not all granites are the same. In particular, variations in mineralogy (Hahm et al., 2014) and bedrock fracture density (Goodfellow et al., 2014) can result in large variations in soil production rates in granites within the same climate.

The San Gabriel Mountains (SGM) of California (Fig. 1) have been the focus of many studies of the relationships among tectonic uplift rates, climate, geology, topography, and erosion (e.g., Lifton and Chase, 1992; Spotila et al., 2002; DiBiase et al., 2010; 2012; DiBiase and Whipple, 2011; Heimsath et al., 2012; Dixon et al., 2012). These studies take advantage of a significant west-to-east gradient in exhumation rates in this range. Spotila et al. (2002) documented close associations among exhumation rates, mean annual precipitation (MAP) rates, and the locations and densities of active tectonic structures. Mean annual precipitation (MAP) rates vary by a factor of two across the elevation gradient and exhibit a strong correlation with exhumation rates (Spotila et al., 2002, their Fig. 10). Lithology, which varies substantially across the range (Fig. 1), also controls exhumation rates. Spotila et al. (2002) demonstrated that exhumation rates are lower, on average, in rocks relatively resistant to weathering (i.e., granite, gabbro, anorthosite, and intrusive rocks) compared to the less resistant schists and gneisses of the range (Spotila et al., 2002, their Fig. 9). This lithologic control on long-term erosion rates can control drainage evolution. For example, Spotila et al. (2002) concluded that the San Gabriel River has exploited the weak Pelona Schist to form a rugged

canyon between ridges capped by more resistant Cretaceous granodiorite (e.g., Mount Baden Powell). Spotila et al. (2002) concluded that landscape evolution in the SGM was controlled by a combination of tectonics, climate, and rock characteristics.

Heimsath et al. (2012) provided a millennial-time-scale perspective on the geomorphic evolution of the SGM. These authors demonstrated that soil production rates ($P$) and erosion rates ($E$) in rapidly eroding portions of the SGM greatly exceed $P_r$ values in slowly eroding portions of the range. Heimsath et al. (2012) concluded that high erosion rates, triggered by high tectonic uplift rates and the resulting steep topography, cause potential soil production rates to increase above any limit set by climate and bedrock characteristics. Their results challenge the traditional view that $P_r$ values are controlled solely by climate and rock characteristics.

Recent research, stimulated by shallow seismic refraction and drilling campaigns, has documented the importance of topographically induced stresses on the development of new fractures (and the opening of pre-existing fractures) in bedrock or intact regolith beneath hillslopes and valleys (e.g. Miller and Dunne, 1996; Martel, 2006; 2011; Slim et al., 2014; St. Clair et al., 2015). In this process, the bulk porosity of bedrock and intact regolith evolves with topographic ruggedness (i.e., topographic slope or curvature). In a compressive-stress environment such as the SGM, topographically induced stresses can result in lower compressive stresses, or even tensile stresses, in rocks near ridgetops. As an elastic solid is compressed, surface rocks undergo outer-arc stretching where the surface is convex-outward (i.e., on hillslopes), reducing the horizontal compressive stress near the surface and eventually inducing tensile stress near ridgetops in areas of sufficient ruggedness. Such stresses can generate new fractures or open pre-existing fractures in the bedrock or intact regolith, allowing potential soil production rates to increase. In this paper I test whether potential soil production rates estimated using the data of Heimsath et al. (2012) are consistent with the topographically induced stress fracture opening hypothesis in the SGM. This hypothesis predicts a relationship between $P_r$ values and average slope that is consistent with the data of Heimsath et al. (2012). Once the effects of average slope are accounted for, I test the hypotheses that climate, lithology, and local fault density also influence $P_r$ values. I then use the resulting empirical model for $P_r$ values to map the spatial variations in potential soil production rates, soil thickness, erosion rates, and topographic steepness across the range in order to illustrate the interrelationships among these variables.

**2 Data analysis and mathematical modeling**

**2.1 Controls on potential soil production rates in the SGM**

Estimates of the maximum or potential soil production rate (i.e., the soil production rate obtained when the buffering effects of soil, if present, are factored out of the measured soil production rate) for the SGM can be estimated using the residuals obtained from the regression of soil production rates to soil thicknesses reported by Heimsath et al. (2012) (their Fig. 3). The exponential form of the soil production function quantifies the decrease in soil production rates with increasing soil thickness:

$$P = P_r e^{-h/h_0}, \tag{1}$$

where $h$ is soil thickness and $h_0$ is a length scale quantifying the relative decrease in soil production rates for each unit increase in soil thickness. Regressing their data to equation (1), Heimsath et al. (2012) obtained $h_0 = 0.32$ m for locations with an average slope, $S_{av}$, less than or equal to 30° and $h_0 = 0.37$ m for locations with $S_{av} > 30°$. Values of the potential soil production rate (Supplementary Table 1) can be estimated as the residuals obtained by dividing the $P$ values measured by Heimsath et al. (2012) by the exponential term in equation (1):

$$P_r = \begin{array}{ll} P e^{h/0.32\,\text{m}} & \text{if } S_{av} \leq 30° \\ P e^{h/0.37\,\text{m}} & \text{if } S_{av} > 30° \end{array}. \tag{2}$$

Note that equation (2) is equivalent to subtracting the logarithms of the exponential term from the logarithms of $P$ values, since division is equivalent to subtraction under log transformation. Log transformation is appropriate in this case because $P$ values are positive and positively skewed (i.e., there are many $P$ values in the range of 50-200 m/Myr and a smaller number of values in the range of 200-600 m/Myr that would be heavily weighted in the analysis if the data were not log-transformed). $P_r$ values estimated from equation (2) increase slowly with increasing $S_{av}$ until an abrupt increase at $S_{av} \approx 30°$ (Fig. 2A).

Heimsath et al. (2012) did not include data points from locations without soil cover in their regressions because these data points appear (especially for areas with $S_{av} > 30°$) to fit below the trend of equation (1). This implies that a humped production function may be at work in some portions of the SGM. The mean value of $P$ from areas with $S_{av} \leq 30°$ that lack soil cover is 183 m/Myr, i.e., slightly higher than, but within $2\sigma$ uncertainty of, the $170 \pm 10$ m/Myr value expected based on the exponential soil production function fit by Heimsath et al. (2012). As such, the evidence indicates that for areas with $S_{av} \leq$ 30°, data from locations with and without soil cover are both consistent with an exponential soil production function. The mean value of $P$ from areas with $S_{av} > 30°$ that lack soil cover is 207 m/Myr, i.e., significantly lower than the $370 \pm 40$ m/Myr

expected based on the exponential soil production function. This suggests that a hump may exist in the soil production function for steep ($S_{av} > 30°$) slopes as they transition to a bare (no soil cover) condition. To account for this, I estimated $P_r$ to be equal to $1.78P$ (i.e., the ratio of 370 to 207) at locations with $S_{av} > 30°$ that lack soil cover.

The SGM has horizontal compressive stresses of ~10 MPa in an approximately N-S direction at depths of less than a few hundred meters (e.g., Sbar et al., 1979; Zoback et al., 1980; Yang and Hauksson, 2013). The development of rugged topography can lead to topographically induced fracturing of bedrock or opening of pre-existing fractures near ridgetops in compressive-stress environments (e.g., Miller and Dunne, 1996; Martel, 2006; Slim et al., 2014; St. Clair et al., 2015). Given the pervasively fractured nature of bedrock in the SGM (e.g., Dibiase et al., 2015), I assume that changes in the stress state of bedrock or intact regolith near ridgetops leads to the opening of pre-existing fractures (i.e., an increase in the bulk porosity of bedrock or intact regolith) rather than the fracturing of intact rock. I adopt the analytic solutions of Savage and Swolfs (1986), who solved for the topographic modification of regional compressive stresses beneath ridges and valleys oriented perpendicular to the most compressive stress direction. Savage and Swolfs (1986) demonstrated that the horizontal stress ($\sigma_{xx}$) in bedrock or intact regolith becomes less compressive near ridgetops as the average slope (measured over a spatial scale that includes ridgetops and side slopes) increases (Fig. 3).

Savage and Swolfs (1986) studied the role of topography in modifying local stresses in a model ridge-and-valley geometry that uses a conformal transformation that includes length scales $b$ and $a$ that define the vertical and horizontal extents of the ridge, respectively. Because the data from Heimsath et al. (2012) are acquired from locations at or near ridgetops, I focused only on the portion of the Savage and Swolfs (1986) solution between the ridgetop and the point of maximum slope, i.e., the broad, U-shaped valley bottoms flanking the central ridge were not considered. The average slope, $S_{av}$, computed from between the ridgetop and the point of maximum slope, is equal to $b/4a$ in the mathematical framework of Savage and Swolfs (1986). A key result of Savage and Swolfs (1986) is their prediction of a gradual decline in the horizontal compressive stress near ridgetops as $b/a$ increases between 0 and 2 (their Figure 4) based on their equation (36):

$$\frac{\sigma_{xx}}{N_1} = \frac{2-b/a}{(2+b/a)(1+b/a)} \tag{3}$$

where $N_1$ is the regional maximum compressive stress and $S_{av}$ has units of m/m in equation (3). Substituting $4S_{av}$ for $b/a$ in equation (3) yields:

$$\frac{\sigma_{xx}}{N_1} = \frac{2-4S_{av}}{(2+4S_{av})(1+4S_{av})} \tag{4}$$

Note that the tangent of the slope angle (units of m/m) is averaged to obtain $S_{av}$ in all cases in this paper. However, after this averaging, $S_{av}$ is reported in degrees in some cases to facilitate comparison with the results of Heimsath et al. (2012).

In landscapes with $S_{av} > 27°$ or atan(0.5), bedrock or intact regolith that would otherwise be in compression develops

tensile stresses close to the surface near ridgetops (Fig. 3A). An average slope of 27° is close to the threshold value of 30° that represents the transition from low to high $P_r$ values in the SGM (Fig. 2). Therefore, the abrupt increase in $P_r$ values at $S_{av} \approx$ 30° is consistent with a transition from compressive to tensile stresses in bedrock or intact regolith near ridgetops of the SGM.

The average slope computed from the model geometry is consistent with the average slope computed by Heimsath et al. (2012). The average slope in the model is computed from the ridgetop to the point of maximum slope in the model geometry.

In the SGM, as with any region of narrow, V-shaped valleys, the steepest portion of the hillslope tends to occur at or near the slope base. Heimsath et al. (2012) computed their average slope from hillslope patches (valley bottoms were excluded) over a length scale that included ridgetops and side slopes. As such, the two calculations are consistent.

It is important to note that the local stress modification in the Savage and Swolfs (1986) model is a function of both local curvature and the slope averaged over a spatial scale that includes ridgetops and side slopes. Within an individual

hillslope, local curvature controls the sign of stress modification, with a reduction in compressive stress (and development of tensile stress in sufficiently rugged terrain) occurring beneath ridgetops and an increase in compressive stress occurring beneath valley bottoms. The compressive-stress reduction that occurs beneath ridgetops is the most important response of the model for the purposes of this paper since the $P_r$ data are from locations at or near ridgetops (i.e., 24 of the 57 data points are on ridgetops, with the remaining data points located within approximately 100 m from ridgetops). The magnitude of the extension

near ridgetops is controlled by the landscape-scale slope, quantified by Savage and Swolfs (1986) as $b/a$. Since $b$ and $a$ are length scales that define the vertical and horizontal extents of the ridge rather than the slope at any one location, the average slope computed over a length scale that includes ridgetops and side slopes is the variable most consistent with $b/a$.

Figure 3 illustrates the effects of topography on tectonic stresses only, i.e., gravitational stresses are not included. Gravitational stresses can be included in the model by superposing the analytic solutions of Savage and Swolfs (1986) (their

equations (34) and (35)) with the solutions of Savage et al. (1985) that quantify the effects of topography on gravitational

stresses (their equations (39) and (40)). The result would be a three-dimensional phase space of solutions corresponding to different values of the regional tectonic stress $N_1$, the characteristic gravitational stress $\rho g b$ (where $\rho$ is the density of rock, $g$ is the acceleration due to gravity, and $b$ is the ridge height), and the Poisson ratio $\mu$. Qualitatively, the effects of gravitational stresses would be 1) to increase the compression at depth via the lithostatic term (at soil depths this corresponds to an addition

of ~10 kPa, which is negligible compared to the regional compressive stress of ~10 MPa in the SGM), and 2) to increase the compressive stresses near the point of inflection on hillslopes (e.g., Fig. 2a of Savage et al., 1985). These modifications do not alter the first-order behavior illustrated in Figure 3 for locations near ridgetops. Section 3 provides additional discussion of the assumptions and alternative approaches to modeling topographically induced stresses.

      The fit of the solid curve in Figure 2A to $P_r$ values is based on equation (4), together with an assumption that the

transition from compressive to tensile stresses triggers a step increase in $P_r$ values over a small range of $S_{av}$ values in the vicinity of the transition from compression to tension:

$$P_{r,S} = \begin{array}{ll} P_{r,l}\left(1 - \frac{\sigma_{xx}}{N_1}\right) & \text{if } S_{av} \leq S_l \\ P_{r,h}\left(1 - \frac{\sigma_{xx}}{N_1}\right) & \text{if } S_{av} > S_h \\ \left(P_{r,l} + \left(P_{r,h} - P_{r,l}\right)\frac{S_{av}-S_l}{S_h-S_l}\right)\left(1 - \frac{\sigma_{xx}}{N_1}\right) & \text{if } S_l \leq S_{av} < S_h \end{array} \tag{5}$$

where $P_{r,S}$ denotes the model for the dependence of $P_r$ values on $S_{av}$, $P_{r,l}$ and $P_{r,h}$ are coefficients defining the low and high values of $P_r$, and $S_l$ and $S_h$ are the average slopes defining the range over which $P_r$ values increase from low to high values

across the transition from compression to tension. $P_{r,l}$ and $P_{r,h}$ were determined to be 170 m/Myr and 500 m/Myr based on least-squares minimization to the data (data from elevations above 2300 m were excluded because of the climatic influence described below). $S_l$ and $S_h$ were chosen to be 30˚ and 32˚, respectively, to characterize the abrupt increase in $P_r$ values in the vicinity of 30˚. The null hypothesis that $P_{r,S}$ values can be fit as well or better by a linear relationship can be rejected: the reduced-$\chi^2$ value, which takes into account different numbers of degrees of freedom, of the log-transformed values of equation

(5), is less than half (45%) of the reduced-$\chi^2$ for a least-squares linear fit. It should be noted that there is effectively no theory or model prediction for how $P_r$ values increase above the abrupt increase in values at $S_{av} \approx 30°$. Equation (5) makes the parsimonious assumption that $P_r$ values continue to increase proportionally to $1 - \sigma_{xx}/N_1$ above the abrupt increase at $S_{av} \approx 30°$.

More sophisticated models would be required to make a more informed prediction regarding how weathering rates might be modified by an increasing magnitude of tensile stress.

In addition to the average slope control associated with the topographically induced stress fracture opening process, a climatic control on $P_r$ values is suggested by the results of a cluster analysis. This type of analysis involves identifying

clusters in the data that are sampled from distinct sets or processes based on the dissimilarity of the means values of the clusters, taking into account the variation within each cluster. The four points colored in blue in Figure 2A are the four highest elevation samples in the dataset, with elevations $\geq 2300$ m a.s.l. The logarithms (base 10) of this cluster have a mean value of -0.40 after subtracting the logarithms of $P_{r,S}$ to account for the average slope control on $P_r$ values, compared with a mean of 0.00 for the logarithms of the remaining data points with $S_{av} > 30°$ (also with the logarithms of $P_r$ subtracted). Assuming a significance

level of 0.05, the null hypothesis that the cluster of blue points is sampled from the same set as that of the remaining points with $S_{av} > 30°$ (i.e., that both sets are governed by the same process or controlling variables) can be rejected based on the standard t test with unequal variances ($t = 0.021$).

Figures 4A-4C illustrate the mean annual temperature (MAT), mean annual precipitation (MAP), and existing vegetation height (EVH) for the central portion of the SGM. Above elevations of approximately 1800 m a.s.l., vegetation

height decreases systematically with increasing elevation (Fig. 4D). This limitation is likely to be primarily a result of temperature limitations on vegetation growth because MAP increases with elevation up to and including the highest elevations of the range. Figure 4E plots the ratio of $P_r$ to $P_{r,S}$ as a function of elevation. The closed circles are binned averages of the data (each bin equals 100 m in elevation). The ratio of $P_r$ to $P_{r,S}$ (equivalent to the residuals under log transformation after the effects of average slope are removed) increases, on average, and then decreases within the range of elevations between 1500 and 2600

m, broadly similar to the trend of EVH (Fig. 4D). Some differences between the curves are to be expected due to the fact that EVH is influenced by the recent fire history, which temporarily reduces EHV in locations that have experienced fire in recent decades. Despite that complication, the fact that both EHV and $P_r/P_{r,S}$ exhibit broadly similar increases and then decreases suggests a causal connection between vegetation cover and weathering rates consistent with a temperature/vegetation limitation on $P_r$ values at the highest elevations of the SGM.

Local variability in $P_r$ estimates due to variations in soil thickness, mineralogical variations within a given lithology, spatial variations in fracture density, etc. can be minimized by averaging $P_r$ values (not including the four highest-elevation points because of the climatic control) from locations that have the same average slope (Fig. 2C). This process tends to average data from the same local cluster since local clusters often have average slopes that are both equal within the cluster and different from other clusters. Figure 2C demonstrates that the predictions of the topographically induced stress fracture opening hypothesis are consistent with the observed dependence of $P_r$ values on $S_{av}$ values.

The average slope and climatic controls on $P_r$ values can be combined into a single predictive equation for $P_r$ values:

$$P_{r,pred} = P_{r,s} C \hspace{4cm} (6)$$

where $P_{r,pred}$ denotes predicted values for $P_r$, $C$ is a climatic index defined as 1 for $z < 2300$ m and 0.4 (i.e., the ratio of the mean of the logarithms of the data for $z > 2300$ m to the mean of the logarithms of remaining data points with $S_{av} > 30°$) for $z > 2300$ m. A regression of $P_{r,pred}$ values to $P_r$ values yields an $R^2$ of 0.50 (Fig. 2D). When data with equal $S_{av}$ values are averaged (i.e., the filled circles in Fig. 2D), the resulting $R^2$ value is 0.87.

The results of this section demonstrate that average slope and possibly climate exert controls on $P_r$ values in the SGM. Although I did not find additional controls that were clearly distinct from these, it is worth discussing additional controls that I tested for. The data points colored in gray in Figure 2B are from the three rock types most resistant to weathering as determined by Spotila et al. (2002): granite, anorthosite, and the Mount Lowe intrusive suite. Spotila et al. (2002) also identified gabbro as a relatively resistant rock in the SGM, but no soil production rates are available from this rock type. Figure 2B suggests that lithology might exert some control on $P_r$ values. Specifically, 7 samples from the more resistant lithologies sit above the least-squares fit of equation (4) to the data, while 13 (including the 7 lowest $P_0$ values) sit below the least-squares fit. However, the null hypothesis that the residuals of the gray cluster (after the effects of average slope are removed) has a mean that is indistinguishable from the residuals of the remaining points (colored black in Figure 2B) cannot be rejected ($t = 0.21$).

Many studies have proposed a relationship between fracture density and bedrock weatherability on the basis that fractures provide additional surface area for chemical weathering and pathways for physical weathering agents to penetrate into the bedrock or intact regolith (e.g., Molnar, 2004; Molnar et al., 2007; Goodfellow et al., 2014; Roy et al., 2016a,b). The

difference in erosion rates between the SGM and adjacent San Bernadino Mountains, for example, has been attributed in part to differences in fracture density between these ranges (Lifton and Chase, 1992; Spotila et al., 2002). As such, it is reasonable to hypothesize that differences in $P_r$ values might result from spatial variations in fracture density within each range. I computed a bedrock damage index $D$ based on the concept that $P_r$ values increase in bedrock that is more pervasively fractured,

together with the fact that bedrock fracture densities are correlated with local fault density in the SGM (Chester et al., 2005; Savage and Brodsky, 2011). Savage and Brodsky (2011) documented that bedrock fracture density decreases as a power-law function of distance from small isolated faults, i.e. as $r^{-0.8}$ where $r$ is the distance from the fault. Fracture densities around larger faults and faults surrounded by secondary fault networks can be modeled as a superposition of $r^{-0.8}$ decays from all fault strands (Savage and Brodsky, 2011). Chester et al. (2005) documented similar power-law relationships between bedrock fracture

density and local fault density in the SGM specifically. I define the bedrock damage index $D$ (Fig. 5A) as the sum of the inverse distances, raised to an exponent 0.8, from the point where the $D$ value is being computed to every pixel in the study area were a fault is located:

$$D = \sum_{\mathbf{x}'} \left( \Delta x / |\mathbf{x} - \mathbf{x}'| \right)^{0.8} \tag{7}$$

where $\Delta x$ is the pixel width, $\mathbf{x}$ is the map location where bedrock damage is being computed, and $\mathbf{x}'$ is the location of each

mapped pixel in SGM where a fault exists. $D$ has units of length since it is the sum of all fault lengths in the vicinity of a point, weighted by a power function of inverse distance. Equation (7) honors the roles of both the distance to and the local density of faults documented by Savage and Brodsky (2011) because longer faults or more mature fault zones with many secondary faults have more pixels that contribute to the summation. The fact that a relationship exists between $P_r$ values and $D$ (Fig. 5B, $p = 0.035$) and between $D$ and $S_{av}$ (Fig. 5C, $p = 0.015$) suggests that some of the control by average slope that I have attributed

to the topographically induced stress fracture opening process may reflect differences in the density of pre-existing fractures related to local fault density. However, the much higher $R^2$ value of the relationship between $P_r$ and $P_{r,pred}$ ($R^2 = 0.50$) compared to that for the relationship between $P_r$ and $D$ ($R^2 = 0.08$) suggests that the topographically induced stress fracture opening process is the dominant mechanism controlling $P_r$ values in the SGM. In addition, this process has a stronger theoretical foundation.

**2.2 Relating potential soil production rates to erosion rates and topographic steepness in the SGM**

In this section I invoke a balance between soil production and transport at the hillslope scale in order to illustrate the interrelationships among potential soil production rates, erosion rates, soil thicknesses, and average slopes spatially across the SGM. The conceptual model explored in this section is based on the hypothesis that the average slope depends on the long-term difference between uplift and erosion rates. Uplift rates (assumed for the purposes of this discussion to be equal to exhumation rates) are lower in the western portion of the SGM and higher in the eastern portion (Spotila et al., 2002, their Fig. 7b). As average slope increases in areas with higher uplift rates, erosion rates increase and soils become thinner. Both of these responses represent negative feedback mechanisms that tend to decrease the differences that would otherwise exist between uplift and erosion rates and between erosion rates and soil production rates. If the uplift rate exceeds the potential soil production rate, soil thickness becomes zero and soil production and erosion rates can no longer increase with increasing slope (in the absence of widespread landsliding in bedrock or intact regolith). In such cases, topography with cliffs or steps may form (e.g., Wahrhaftig, 1965; Strudley et al., 2006; Jessup et al., 2010). However, if the potential soil production rate increases with average slope via the topographically induced stress fracture opening process, the transition to bare landscapes can be delayed or prevented as Heimsath et al. (2012) proposed. This represents an additional negative feedback or adjustment mechanism beyond the increase in soil production rates in steep terrain made possible by the exponential form of the soil production function. At the highest elevations of the range, soil production is slower, possibly due to temperature limitations on vegetation growth. The interrelationship between these variables can be quantified without explicit knowledge of the uplift rate since the relationship between soil thickness and average slope implicitly accounts for the uplift rate (i.e., a smaller difference between uplift and erosion rates is characterized by a thinner soil). This conceptual model predicts positive correlations among potential soil production rates, erosion rates, and topographic steepness, and negative correlations of all of these variables with soil thickness.

Equation (6), in combination with modified versions of equations (9)&(11) of Pelletier and Rasmussen (2009), i.e.,

$$P_r e^{-h/h_0} = E \tag{8}$$

and

$$\frac{\kappa S_{av}}{1-(S_{av}/S_c)^2} = EL, \tag{9}$$

predict spatial variations in erosion rates and topographic steepness associated with spatial variations in $P_r$ values. In equations (7)&(8), $\kappa$ is a sediment transport coefficient (m$^2$/Myr) and $L$ is a mean hillslope length (m). Equation (9) assumes a steady state balance between soil production and erosion, modeled via the nonlinear slope-dependent sediment flux model of Roering et al. (1999) at the hillslope scale. Equation (9) also assumes that the mean slope gradient at the base of hillslopes (where the

sediment flux leaves the slope) of a given area can be approximated by the average slope.

Spatial variations in erosion rates can be estimated using $P_r$ values predicted by equation (6) if spatial variations in soil thickness can also be estimated. To do this, I developed an empirical relationship between soil thickness and slope gradient derived from the Heimsath et al. (2012) dataset (Fig. 6):

$$h = \frac{h_1}{s_{av}^b}, \tag{10}$$

with best-fit coefficients of $b = 1.0$ and $h_1 = 0.06$ m ($R^2 = 0.18$, $p = 0.001$). For this regression, I shifted the soil thickness in areas with no soil upward to a small finite value (0.03 m). These areas have no soil today, but must have had some transportable material (i.e., soil) at some point in the past or else no erosion would occur. Also, without some shift, the 10 data points with $h = 0$ cannot be used, biasing the analysis towards areas that have soil cover today. The 0.03 m value was chosen because this is the minimum finite soil thickness measured by Heimsath et al. (2012).

Using equation (10) as a substitution, equations (8)&(9) can be combined to obtain a single equation for $S_{av}$:

$$\frac{S_{av}}{1-(S_{av}/S_c)^2} = \frac{L}{\kappa} P_{r,pred} \exp\left(-\frac{h_1}{h_0 s_{av}^b}\right) \tag{11}$$

Given a map of steepness obtained by solving equation (11), soil thicknesses and erosion rates can be mapped using equations (10) and (8), respectively. Note that the $S_{av}$ value obtained by solving equation (11) is not a prediction in the usual sense since $S_{av}$ is an input to equation (11) via $P_{r,pred}$.

Equations (8)&(9) are the same as equations (9)&(11) of Pelletier and Rasmussen (2009) except that their equation (9) included a term representing the bedrock-soil density contrast related to a slightly different definition of $P_r$ (termed $P_0$ in Pelletier and Rasmussen (2009)) and their equation (11) assumed a depth- and slope-dependent transport relation. Here I use a soil-depth-independent transport relation because such models are highly sensitive to the presence/absence of soil and areas

of thin or no soil are likely to have episodic cover (e.g., rapid mass wasting following incipient soil production) that makes measuring or estimating long-term-averaged soil depths difficult.

The $S_{av}$ values predicted by equation (11) (Fig. 7C) reproduce the observed first-order patterns of topographic steepness (Fig. 7C) if $L/\kappa = 0.005$ Myr/m and $S_c = 0.8$ are used. The value $S_c = 0.8$ was chosen because it is in the middle of the range of values (i.e., 0.78-0.83) that Grieve et al. (2016) obtained for steep landscapes in California and Oregon. With this value for $S_c$, the best-fit value for $L/\kappa$ was determined by minimizing the least-squares error between the model prediction (Fig. 7B) and observed variations in average slope (Fig. 7C). Predicted and measured $S_{av}$ values are lowest in the Western block and higher in the Sierra Madre, Tujunga, and Baldy blocks. Soil thicknesses predicted by the model correlate inversely with slopes and $P_r$ values (Fig. 7D). Erosion rates (Fig. 7E) closely follow $P_r$ values, but are lower in absolute value, reflecting the buffering effect of soil on bedrock physical weathering processes.

The model can be further tested by comparison to the catchment-averaged erosion rates reported by DiBiase et al. (2010). Figure 8 plots catchment-averaged erosion rates (unfilled circles) as a function of catchment-averaged $S_{av}$ values. As with the $P_r$ values plotted in Figure 2C, I averaged the data in bins of slope in order to minimize local variability related to factors besides average slope. The solid curve represents the model prediction for erosion rate, i.e., equation (8) with $P_r$ values predicted by equation (5) and $h$ values predicted by equation (10). Catchment-averaged erosion rates follow a similar pattern as predicted values, remaining constant or increasing slightly with increasing $S_{av}$ until $S_{av} \approx 30°$, beyond which erosion rates increase abruptly. The similarity between $E$ and $P_r$ values (Figs. 8 and 2) reflects the important influence of $S_{av}$ on both variables, the coupling between these variables (i.e., in the absence of widespread landsliding in bedrock or intact regolith, soil must be produced in order for erosion to occur), and the modest impact that differences in soil thickness have on soil production rates across landscapes of different relief in the SGM. Except for several data points of relatively high erosion rates at both the lowest ($S_{av} = 10\text{-}15°$) and highest slopes ($S_{av} > 35°$), the model reproduces the absolute values and the slope dependence of the measured erosion rates reasonably well. The underprediction of the model at the highest slopes may be due, in part, to the fact that the $P_r$ values used to calibrate the model has relatively few data points near the highest end. For example, 4 of 57 $P_r$ values are above 500 m/Myr, while 11 of 50 catchments have erosion rates above this value. The comparison of the predicted curve to the model is not meant to imply that the model prediction is the best or only mathematical expression that represents the

data. Rather, Figure 8 (and Figure 2 for the model-data comparison of $P_r$ values) is intended only to demonstrate consistency with the threshold increase at $S_{av} \approx 30°$ predicted by the Savage and Swolfs (1986) model.

## 3 Discussion

This paper adopts a stepwise regression and cluster analysis approach that builds upon the regression analysis that Heimsath et al. (2012) used to characterize the dependence of soil production rate on soil thickness. Stepwise regression is the process of computing the residuals of a regression and testing for additional controls, via additional regression and the calculation of a new set of residuals, until no additional explanatory variable can be identified. Stepwise regression is one method for testing the residuals of a regression for additional controls, which is a recommended step in all regression analyses. I did not apply simultaneous multivariate linear regression (with or without log transformation) because such an approach would have been inconsistent with the complex nonlinear relationships in the data documented by Heimsath et al. (2012) and the analyses presented here.

Estimating $P_r$ values using the residuals of the regressions of Heimsath et al. (2012) assumes that $h_0$ has sufficiently limited variation within the two subsets of the study site considered by Heimsath et al. (2012) (i.e., those with $S_{av}$ values above and below 30°) that any such variation would not affect the conclusions of the paper. For example, in order for the relationship between $P_r$ and $S_{av}$ (i.e., Figs. 2A-2C) to be significantly affected by variations in $h_0$, $h_0$ would have to have a systematic dependence on $S_{av}$. For example, if systematically lower values of $h_0$ occur at steeper slopes and this effect is not accounted for, the result could be a biasing of $P_r$ values downward in such regions. Heimsath et al. (2012) clearly demonstrated that no such systematic dependence exists. These authors considered two end member slope regimes and found that the average $h_0$ values for these two regions differed by only 0.05 m (0.32 m vs. 0.37 m). At a soil thickness of 0.3 m, this difference corresponds to $P_r$ differences of approximately 10% (i.e., exp(-0.30/0.32) vs. exp(-0.30/0.37)). This difference is more than 100 times smaller than the variation in $P_r$ values. The difference becomes even smaller for soils thinner than 0.3 m.

Savage and Swolfs (1986) used a convex-concave geometry, defined by a conformal transformation, in which the slope increases linearly with distance from the divide to the steepest point on the hillslope. In higher-relief portions of the SGM characterized by more planar hillslopes, slopes increase abruptly over a relatively short distance from the ridgetop, then

more slowly with increasing distance from the ridgetop. This difference introduces some uncertainty into the application. The model might overestimate the magnitude of topographically induced stress in high-relief portions of the SGM because a more planar slope has a lower curvature than a more parabolic slope and larger curvatures tends to increase extensional stress. On the other hand, more planar hillslopes localize curvature near the ridgetops, which might tend to increase bending stresses that

drive extension over and above that predicted by the model for locations near ridgetops.

The effect of topographically induced stresses on the production of intact regolith or soil is a rapidly evolving field at the boundaries among geomorphology, geophysics, and structural geology. The results presented here, based on the Savage and Swolfs (1986) model, represents just one possible approach to the problem. Miller and Dunne (1998), for example, modified the Savage and Swolfs (1986) solutions to account for cases with vertical compressive stress gradients (their

parameter $k$) larger than 1. Data from the SGM and the adjacent southwestern Mojave Desert indicate that the vertical gradient of horizontal stress in the SGM is likely less than one, so the modification of Miller and Dunne (1998) may not be necessary for the SGM. Sbar et al. (1979) measured mean maximum compressive stresses at the surface equal to 16 MPa, which is similar to values measured at depths of 100-200 m obtained by Zoback et al. (1980) (their Figs. 7&10). In addition to the effects of variations in the depth gradient of stress, fractures can open beneath hillslopes in a direction perpendicular to the slope, parallel

to the slope, or in shear. The criteria for each of these strains depends on different components or derivatives of the stress field. For example, Martel (2006, 2011) emphasized the vertical gradient of vertical stress, which depends on the topographic curvature instead of the slope, in driving fracturing parallel to the surface, while St. Clair et al. (2015) emphasized the ratio of the horizontal stress to the spacing between ridges and valleys. More research is needed in the SGM and elsewhere to better understand the response of bedrock and intact regolith to the 3D stress field. However, all studies agree that the extent of one

or more fracture opening modes increases with topographic slope and/or curvature, often with a threshold change from compression to tension above a critical value of topographic ruggedness.

The results presented here provide a possible process-based understanding of the dependence of potential soil production rates on topographic steepness documented by Heimsath et al. (2012) in the SGM. These authors proposed a negative feedback in which high erosion rates trigger higher potential soil production rates, with the result that soil cover may

more persistent than previously thought. The results presented here show that previous models of topographically induced

stresses suggest transitions from compressive to tensile strength at hillslope angles similar to those at which $P_r$ values increase. This similarity suggests that in the SGM, the release of compressive stress in steep landscapes may cause fractures beneath ridges to open, thereby allowing weathering agents to penetrate into the bedrock or intact regolith more readily. The fact that this process requires a regional compressive stress state suggests that it is not likely to be equally important everywhere on Earth. In cases of low regional compression or extension, the development of rugged topography in rocks with pre-existing fractures is not likely to be significant in promoting fracture opening in the rocks beneath hillslopes.

Heimsath et al. (2012) argued that $P_r$ values (analogous to what they termed $SPR_{max}$ values) increase with erosion rates not just in the SGM, but globally based on the strong correlation between $P$ and $E$ values (their Fig. 4b). However, the results of this paper argue against a global increase in $P_r$ values with $E$ values. The process described in this paper, i.e., compressive-stress reduction near ridgetops in compressive-stress environments, does not apply to extensional or neutral-stress settings. As such, other factors might explain the global correlation between $P$ and $E$ values. For example, erosion rates may be limited by $P_r$ values (since erosion cannot occur faster than soil is produced in the absence of widespread landsliding in bedrock or intact regolith). Also, $P_r$ values are a function of climate, with values exceeding 1000 m/Myr in humid climates (Pelletier and Rasmussen, 2009; Larsen et al., 2014). As such, the global correlation between $P$ and $E$ values may, in part, be a result of water availability being important for both soil production and erosion processes. If soil production rates cannot keep pace with erosion rates, stepped topography can and does form in many cases (e.g., Wahrhaftig, 1965; Strudley et al., 2006; Jessup et al., 2010), leading to a reduction in erosion rates (as evidenced by lower soil production rates in bare areas relative to soil-covered areas (Hahm et al., 2014)) despite locally steeper slopes. In such cases, $P$ and $E$ values are still correlated because erosion cannot occur at rates higher than $P_r$.

**4 Conclusions**

In this paper I estimated spatial variations in the potential soil production rate, $P_r$, using cosmogenic-radionuclide-derived soil production rates from the central San Gabriel Mountains of California published by Heimsath et al. (2012). The results demonstrate that trends in the data are consistent with the hypothesis that topographically induced stresses cause pre-existing fractures to open beneath steeper hillslopes. This model predicts an abrupt increase in $P_r$ values close to the average

slope (approximately 30˚) where an increase is observed in the data. After the effects of topographically induced stress are accounted for, a limitation on $P_r$ values is detectable at the highest elevations of the range where vegetation growth is limited by temperature. There is some evidence that lithology and local fault density may also influence potential soil production rates, but the null hypotheses that these processes are not significant cannot be ruled out with given a threshold statistical significance (false positive rate) of 0.05, or they cannot be clearly distinguished from other controls. The results of this paper demonstrate that $P_r$ values may be solely dependent on climate and rock characteristics as has been traditionally assumed, but that rock characteristics evolve with topographic ruggedness in compressive-stress environments. These results provide a useful foundation for additional targeted cosmogenic-radionuclide analyses in the San Gabriel Mountains and for the incorporation of methods that can further test the topographically induced stress fracture opening hypothesis such as shallow seismic refraction surveys and 3D stress modeling.

### Acknowledgements

I thank Katherine Guns for drafting Fig. 1. I wish to thank Arjun Heimsath, Kelin Whipple, Simon Mudd, Jean Braun, and six anonymous reviewers for reviews of this and earlier versions of the manuscript.

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

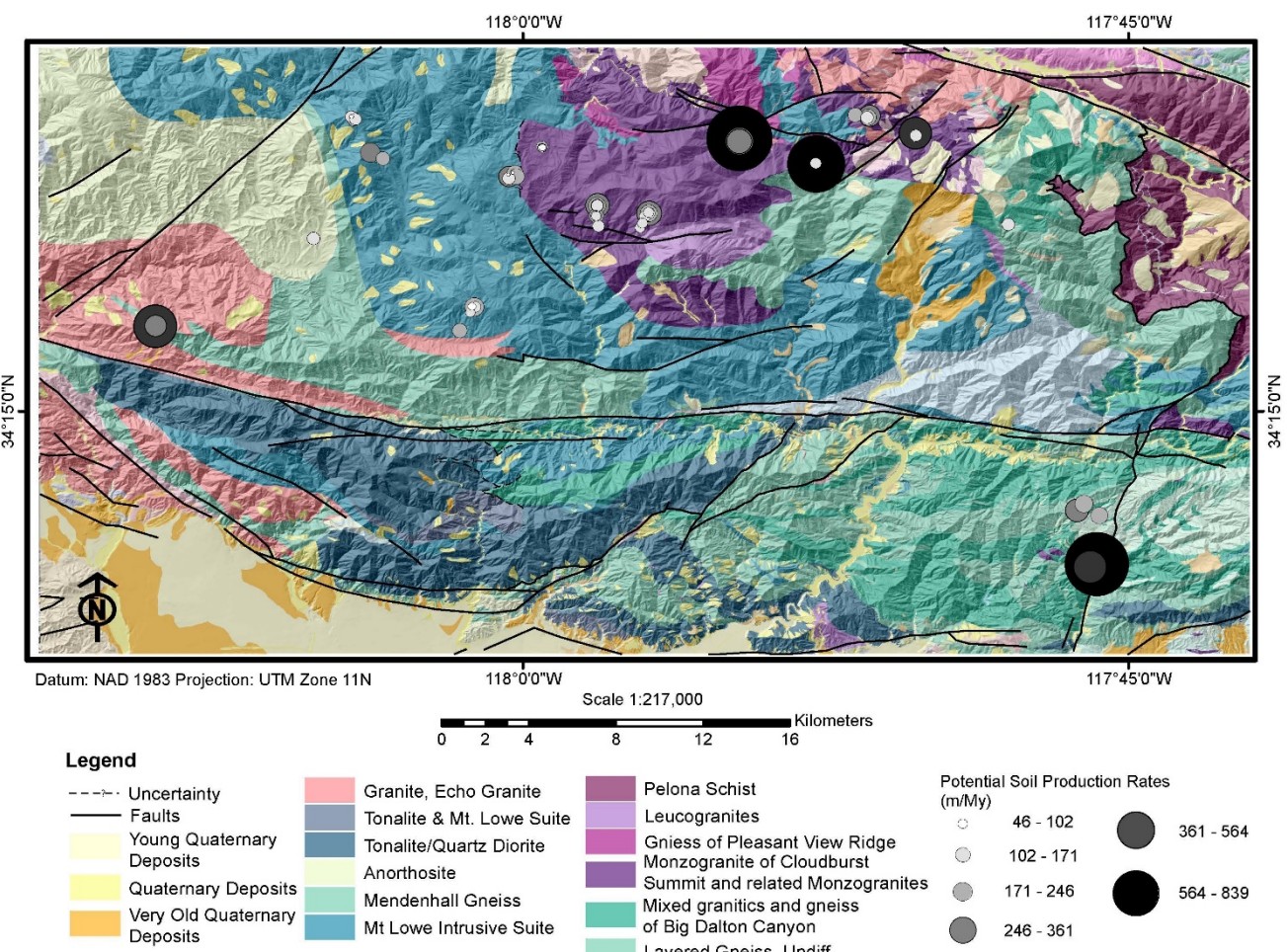

**Figure 1. Geologic map of the central San Gabriel Mountains, California. Potential soil production rates inferred from the data of Heimsath et al. (2012) are also shown. Lithologic units were compiled using Yerkes and Campbell (2005), Morton and Miller (2003), and Figure 3 of Nourse (2002). Faults were mapped from Morton and Miller (2003) and the Quaternary fault and fold database of the United States (U.S. Geological Survey and California Geological Survey, 2006).**

,

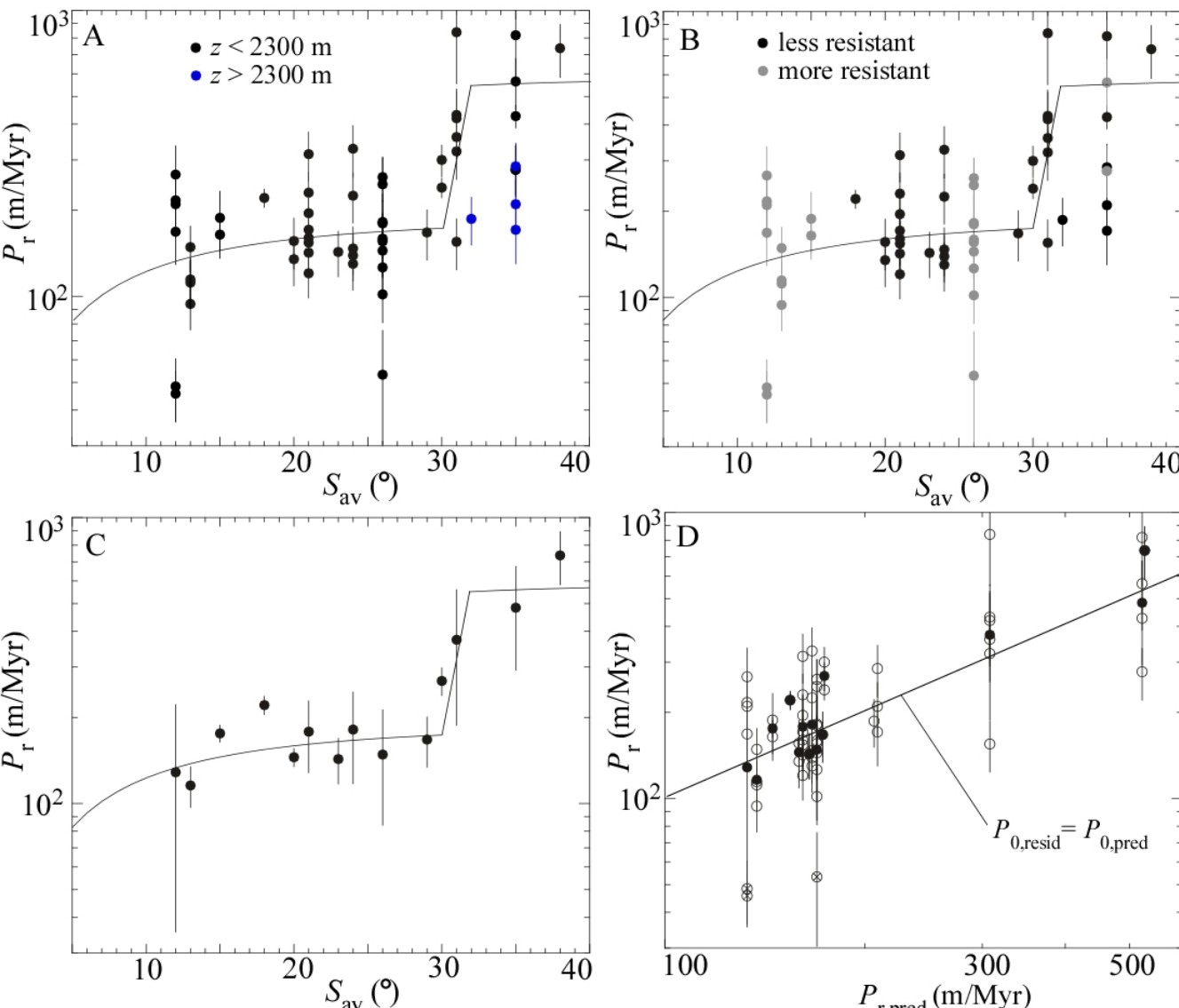

**Figure 2. Plots of $P_r$ and their relationship to average slope, $S_{av}$, and other potential controlling factors. (A)** Plot of $P_r$ values versus $S_{av}$. Data points colored blue are from the highest elevations of the range ($z > 2300$ m). The piece-wise curve plots equation (4), with the three segments of the curve corresponding to the three conditions in the equation. **(B)** The same plot as (A), except that data points are colored according to whether they from rocks that are relatively more resistant (gray) or less resistant (black) to weathering. **(C)** Plot of $P_r$ values averaged for each value of $S_{av}$. In (A) and (B), error bars represent the uncertainty of each data point, while in (C) the error bar represents the standard deviation of the data points averaged for each $S_{av}$ value. **(D)** Plot of $P_r$ versus values predicted from equation (5). Unfilled circles show individual data points, while filled circles represent the averaged data plotted in (C).

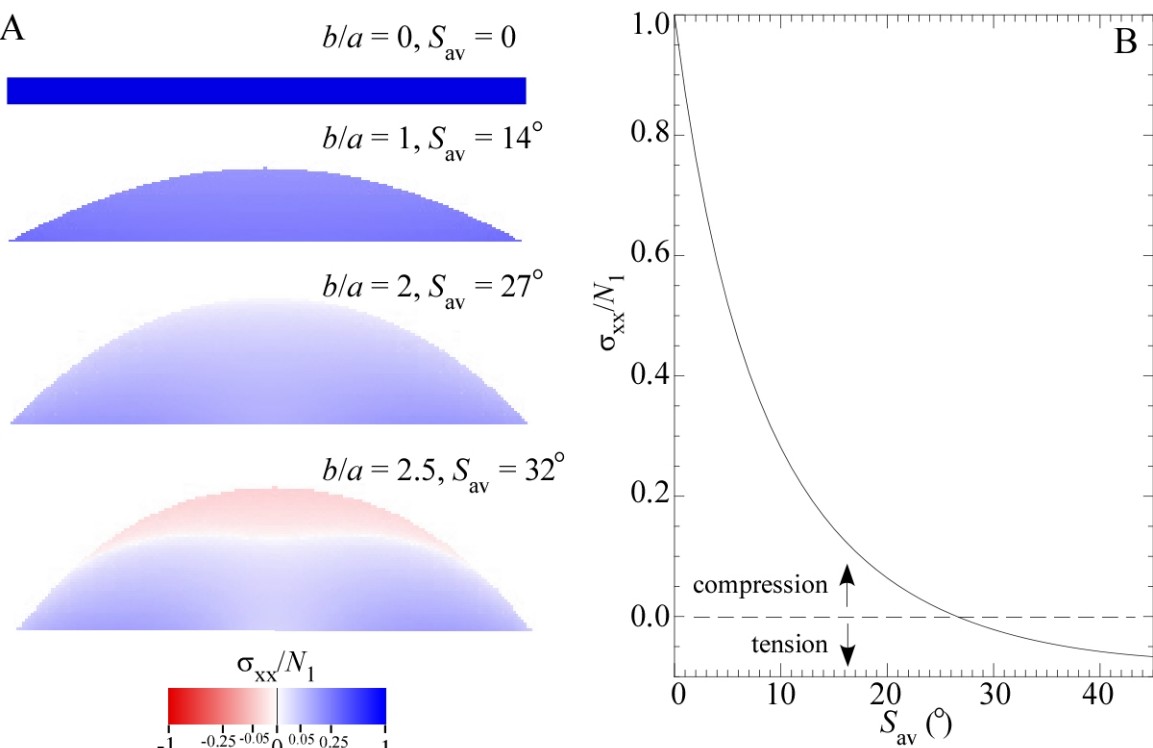

**Figure 3.** Analytic solutions illustrating the perturbation of a regional compressive stress field by topography. **(A)** Color maps of the horizontal normal stress, $\sigma_{xx}$ (normalized to the regional stress, $N_1$), as a function of ridge steepness (defined by the shape factor $b/a$ of Savage and Swolfs (1986) and the average slope $S_{av}$) using equations (34) and (35) of Savage and Swolfs (1986). The hillslopes are plotted with no vertical exaggeration. **(B)** Plot of $\sigma_{xx}$ directly beneath the ridge as a function of $S_{av}$ using equation (36) of Savage and Swolfs (1986). The plot illustrates the decrease in compressive stress with increasing average slope and the transition to tensile stresses at a $S_{av}$ value of approximately 27°.

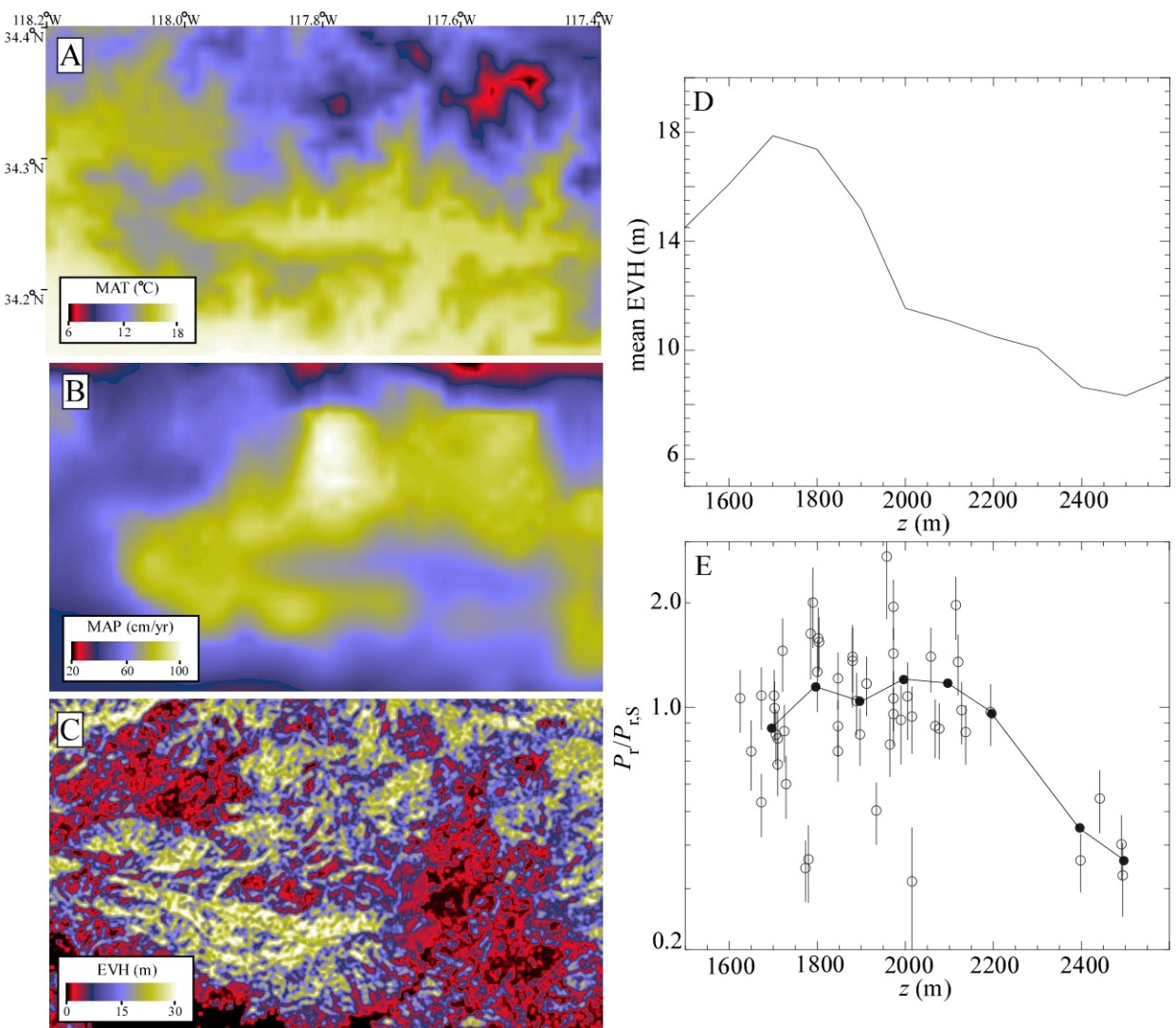

**Figure 4. Climate and vegetation cover of the central San Gabriel Mountains.** Color maps of (A) mean annual temperature (MAT) and (B) mean annual precipitation (MAP) from the PRISM dataset (Daly et al., 2001). (C) Color map of mean existing vegetation height (EVH) from the U.S. Geological Survey LANDFIRE database (U.S.G.S., 2016). (D) Plot of mean EVH versus elevation above sea level, $z$, using the data illustrated in (C). (E) Plot of the ratio of $P_r$ to $P_{r,S}$ as a function of elevation. Filled circles are binned averages of the data (each bin equals 100 m in elevation).

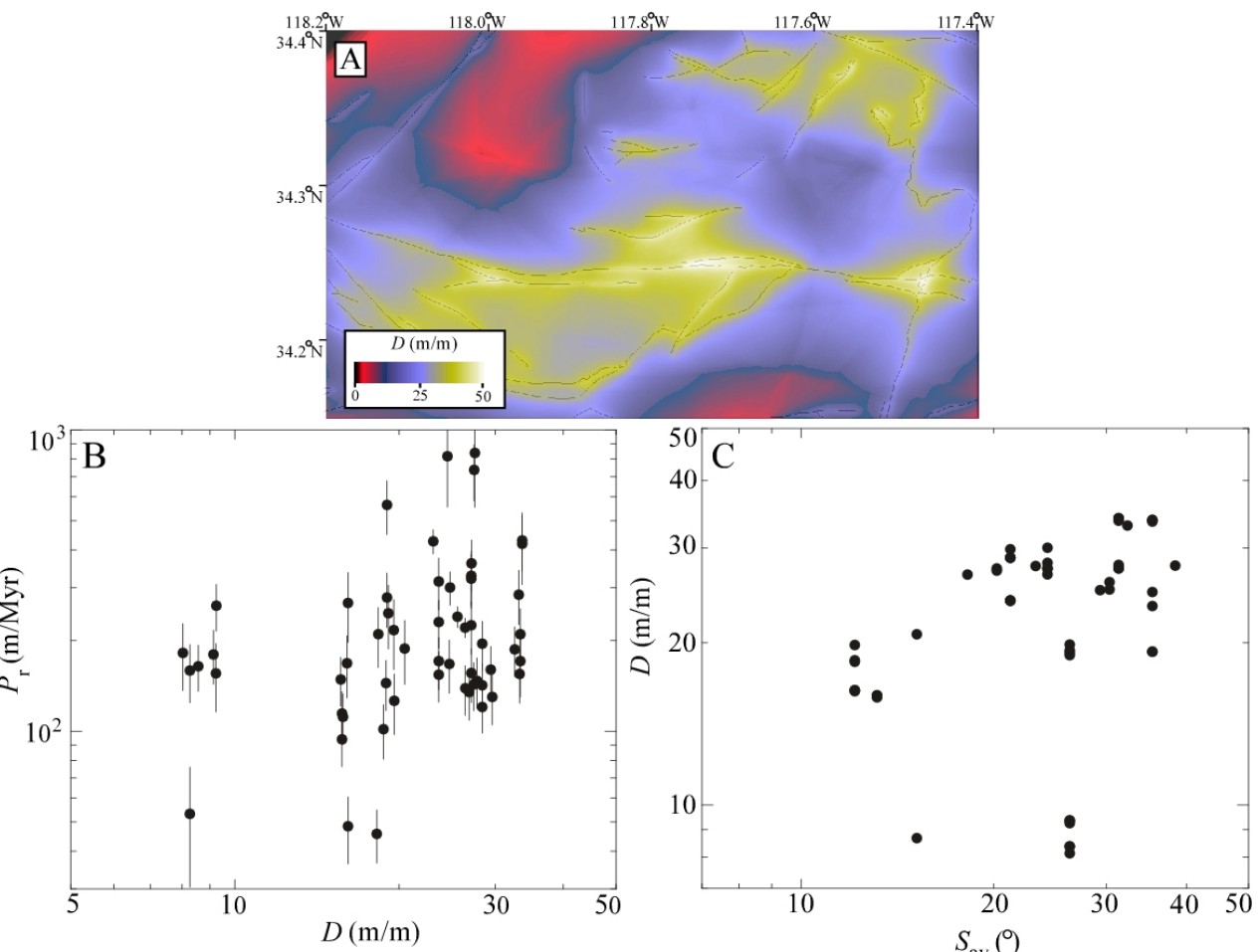

**Figure 5.** Map of the bedrock damage index, $D$, and its correlation with $S_{av}$. **(A)** Color map of spatial variations $D$. **(B)** Plot of $D$ versus $S_{av}$ for the 57 sample locations of Heimsath et al. (2012).

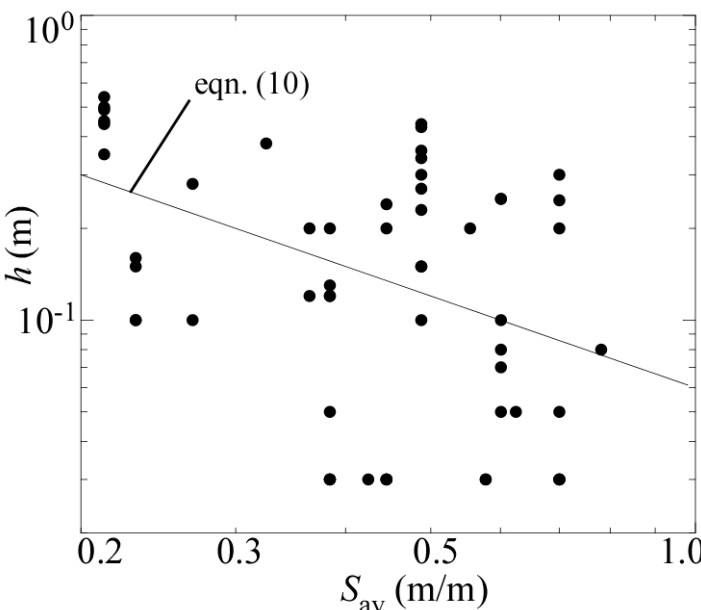

**Figure 6. Plot of soil thickness, *h*, as a function of average slope, $S_{av}$. The least-squares power-law fit to the data (eqn. (10)) is also shown.**

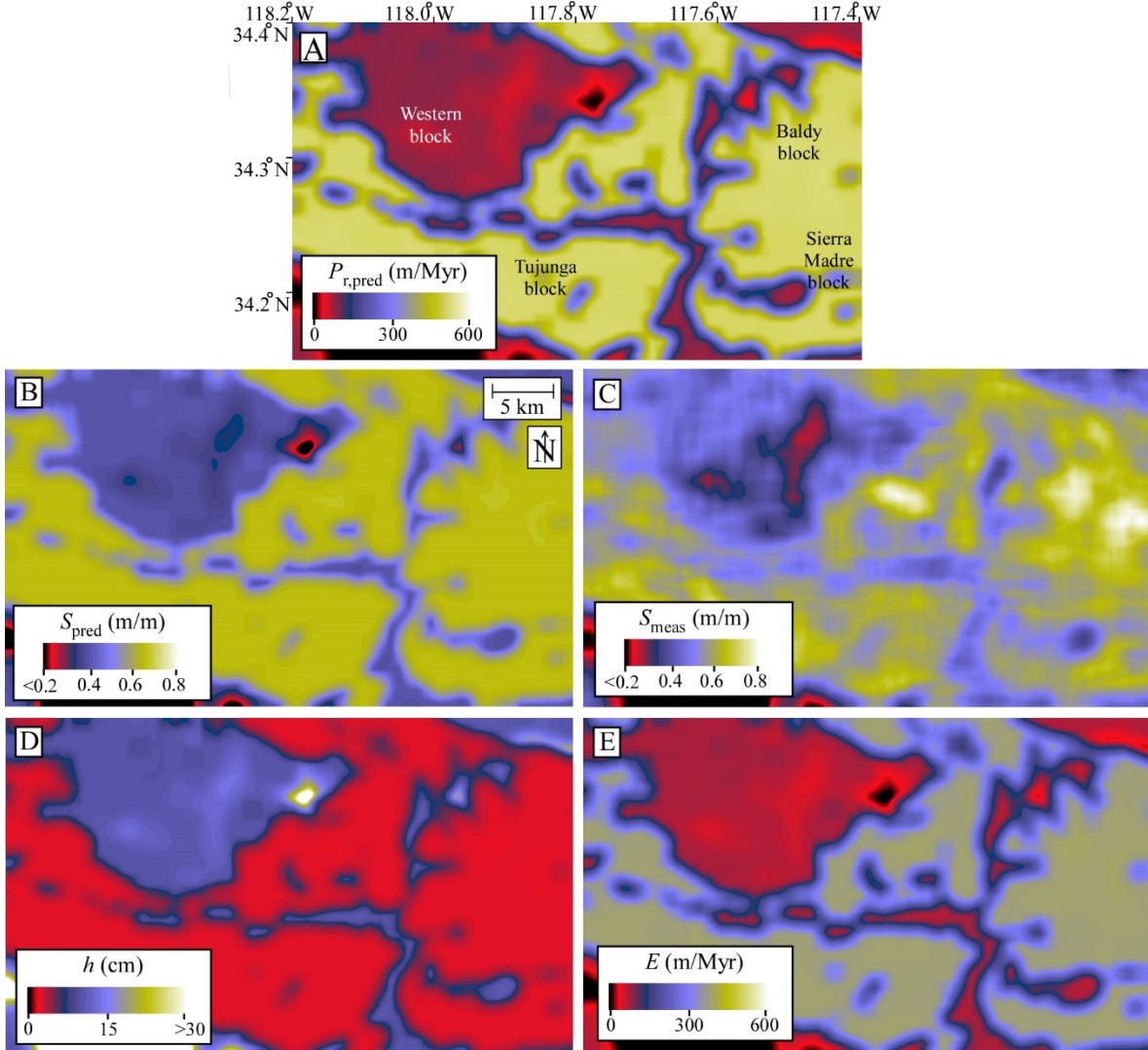

**Figure 7.** Color maps illustrating the predicted potential soil production rate from equation (6) ($P_{r,pred}$), predicted and observed values of average slope, $S_{av}$, soil thickness, $h$, and erosion rate, $E$. (A) Color map of $P_{r,pred}$ values estimated from equation (6). (B) Color map of $S_{av}$ values predicted by equation (11), smoothed by a moving average filter with a 1-km length scale to emphasize patterns at the landscape scale. (C) Color map of actual (DEM-derived) $S_{av}$ values, smoothed in the same manner as (B). (D) Color map of soil thicknesses, $h$, predicted by equation (10). (E) Color map of erosion rates, $E$ predicted by equation (8).

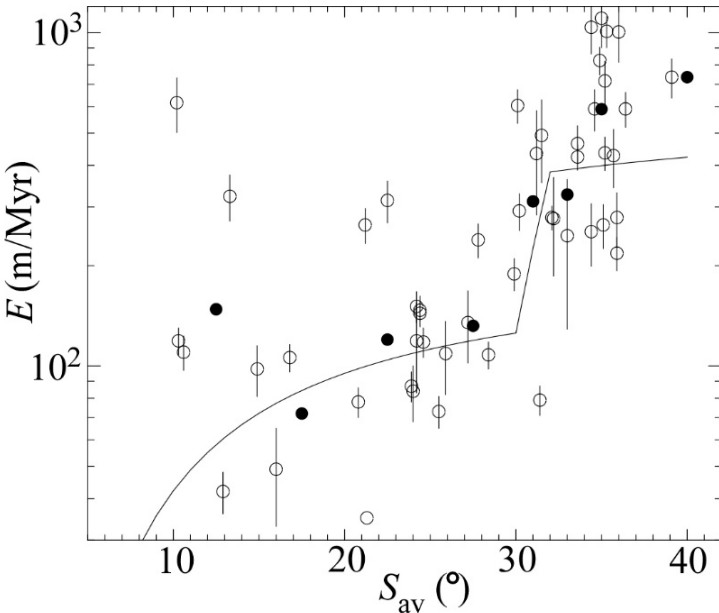

**Figure 8.** Plot of the catchment-averaged erosion rates of DiBiase et al. (2010) (unfilled circles) versus catchment-averaged $S_{av}$. Filled circles represent log-transformed averages of data within the following bins: 10-15°, 15-20°, 20-25°, 25-30°, 30-32°, 32-34°, and 34-36°. The curve plots the model prediction, i.e., equation (8) with $P_r$ values predicted by eqn. (5) and $h$ values predicted by equation (10).