# Peer review of "Quantifying the controls on potential soil production rates: A case study of the San Gabriel Mountains, California"

_Earth Surface Dynamics, 2016_

## Referee Comment (RC1) · A. Heimsath (Referee) · 23 Sep 2016

This manuscript by Pelletier is focused on a reinterpretation of soil production rates measured in the San Gabriel Mountains, CA by Heimsath et al. (2012), and in particular the controls on the intercept of the soil production function, P0 in the present manuscript, which is the peak conversion rate from rock to soil and a key parameter that determines the presence and thickness of soils in upland landscapes.

Heimsath et al. (2012) used soil production functions determined from low and high-uplift zones in the San Gabriel Mountains to show that P0 is higher in rapidly eroding

areas with steep slopes and high local relief, with the implication that soil production rates increase in concert with regional erosion rates to help sustain a soil mantle in steep landscapes. Implicit in the claim of Heimsath et al. (2012) is the notion, supported by data, that for the San Gabriel Mountains, spatial gradients in rock uplift rate and thus relief are dominant compared to variations in rock material properties or climate, which also likely influence soil production rates (See for example Owen et al. 2011 ESPL, 36: 117-135). The Heimsath et al (2012) sampling strategy was designed specifically to minimize climatic and lithologic differences between sites.

Here, Pelletier proposes that the variability in P0 present in the San Gabriel Mountains (SGM) dataset is due instead to spatial variations in rock material properties driven by faulting-related damage and aspect-dependent microclimate. While it is well worth exploring how rock properties and microclimate may influence soil production rates both in the SGM and elsewhere, this paper presents no data on either and the analysis provides scant evidence to support the proposed re-interpretation. Moreover, the theoretical underpinnings of the empirical microclimate-fault-damage "model" are tenuous at best. In the absence of either sound theory or any new data, we cannot recommend this manuscript for publication. Below we articulate our 4 main concerns as concisely as possible without belaboring the points:

(1) Poor theoretical basis for Equation 2. The core of the paper is centered on an empirical expression for P0 as a function of a damage index D and microclimate A. However, it is completely unclear how these parameters are connected to the actual process of soil production. For example, the damage index is based on empirical studies of fracturing in bedrock as a function of distance from a fault over scales typically <100m. There is no reason to believe these relationships can be meaningfully applied at landscape scale. Moreover, why should the relationship between fracture density and P0 be linear? There are no data to substantiate this, and no mechanistic justification. Indeed one might anticipate a non-linear dependence. Second, the microclimate factor "A", is defined as a function of the product of aspect and slope. Again, there is little

justification for this treatment beyond a weak empirical fit to the Heimsath et al. (2012) data. A major concern here is that the microclimate factor is strongly slope-dependent , and thus cannot be untangled from the slope controls proposed by Heimsath et al. (2012). Additionally, "microclimate" is obviously a factor that modifies the local "macroclimate" (Precipitation and Temperature) but there is no representation of these factors in the analysis.

(2) Inappropriate data handling. Computing P0 values from every single measurement of P is unwise. Normal practice is to collect many measurements of P under different thicknesses of soil, thus defining the "soil production function" and enabling estimation of P0 via regression analysis. Heimsath et al. (2012) only felt that their data justified two estimates of P0. One might attempt a finer-subdivision of the data as a function of rock properties, microclimate, or catchment-mean erosion rate to perhaps generate a handful of potentially robust P0 estimates, but what is done here in this manuscript has little value. The scatter in P0 values obtained in this manner is not particularly meaningful – it only reflects in a less-than-quantitative way the uncertainty in our ability to determine what P0 is and how it might vary with environmental conditions.

(3) Suspect empirical basis for Equation 2. Ultimately such an empirical approach might be justified if the proposed driving factors were quantified and showed a sound correlation with P0. However, this is not the case. The damage index D is determined from fault traces on existing geologic maps, and likely has little bearing on the actual pattern of rock damage. At a minimum, we expected to see at least some validation of this ad hoc treatment beyond the extremely weak correlation presented in Figure 3A. (as a side note, there is no justification for throwing out the cluster of five points described on Page 5, Line 20 besides "area with an unusually high density of mapped landslides"). The empirical basis for the microclimate factor is similarly weak, particularly given that any correlation is likely due to the slope-dependence (which may lead to a spurious interpretation of climate control due to its primary dependence on rock uplift rate). It should also be noted that there is not a strict correspondence between

precipitation (which varies primarily with elevation) and rock uplift rate – the presence of elevated, low-relief surfaces in the San Gabriel Mountains allows the decoupling of potentially confounding climatic and tectonic controls on soil production (see Dixon et al., 2012 EPSL).

(4) Weak logic chain in discussion. Even ignoring the limited theoretical and empirical basis for Equation 2, and the inappropriate handling of local measurements of P, the discussion does not present any particularly useful new insight. The discussion of focused tectonic uplift in regions with high P0 (Page 10, Line1-9) is off base and may be confusing to many readers. It is true that at the scale of an orogenic wedge rock uplift rates – and slip rates on associated faults – may increase in response to rapid erosion (e.g., Willett, 1999), but this feedback simply cannot operate at the scale of the SGM block. It is simply invalid to argue that differences in P0 driven by proximity to faults and microclimate can enhance erosion rates and thus induce more rapid uplift and create the steeper topography of the eastern SGM. There are very good structural reasons to expect more rapid tectonic rock uplift at the eastern end of the SGM. This more rapid rock uplift produces the higher elevations, greater relief, and drives more rapid erosion. Certainly until proven otherwise this should be considered the most likely and simplest scenario. Once this gradient in elevation, relief and erosion rate is established, there will be associated differences in climate (precipitation and temperature) and potentially the degree of rock damage (though this is unproven with any data), which may have some influence on P0. Further study of these potential influences is certainly warranted, but useful progress will require the formulation of hypotheses based on a clear theoretical basis as well as the collection of data appropriate to testing these specific hypotheses.

– Arjun Heimsath & Kelin Whipple

---

## Author Comment (AC1) · 25 Sep 2016

Heimsath and Whipple criticize me for not presenting a comprehensive theory for how rock damage and microclimate influence $P_0$ values (their point 1). I chose to focus the theory/modeling portion of my paper on the relationships among $P_0$ values, erosion rates, soil thicknesses, and topographic slopes, all of which I mapped across the central portion of the SGM as a test of my model. A lot of relevant theoretical concepts exist, well-known to everyone in the hillslope geomorphic community, regarding the relationships among weathering rates and the fracture densities of bedrock and intact regolith. For example, it is a textbook result that chemical weathering rates of fresh bedrock increase proportionally to the surface area available for weathering reactions, which depends linearly on the density of vertically oriented fractures. It is also well known that a higher density of vertically oriented fractures provides proportionally more incipient weaknesses that weathering agents acting on the contact between soil and intact regolith (or between soil and bedrock) from above (e.g., infiltrating water and tree roots) can exploit to produce intact regolith and soil (e.g., Langston et al., 2015). Regarding microclimate and its influence on wildfire intensity and severity, the fact that intense heating associated with wildfires can fracture bedrock and intact regolith in areas of thin or no soil cover is well established both experimentally and in the field, as the references demonstrate (Blackwelder, 1927; Goudie et al., 1992; Dorn, 2003; Shtober-Zisu et al., 2010). It is far beyond the scope of the paper to develop a comprehensive theory for how microclimate relates to vegetation cover, wildfire frequency and severity, and soil production rates, assuming such a theory is even possible. I will, however, add additional discussion of the available theory and concepts related to the relationship between soil production rates and bedrock fracture density to my revision.

Heimsath and Whipple's contention that $P_0$ values cannot be estimated point-by-point (point 2) is contradicted by the fact that the relative decrease in soil production rates with increasing soil thickness is nearly identical throughout the portions of the SGM sampled by Heimsath et al. (2012). My analysis assumes that there is only minor variation (compared to the 1.5 order of magnitude variation in $P_0$ values) in how soil production rates depend on soil thickness in the SGM. This is precisely what the data show: soil production rates in slowly and rapidly eroding portions of SGM with finite soil thickness (which include a wide range of aspects and distances from faults) are well represented by an exponential function of soil thickness with decay parameters that are almost identical, i.e., 0.027 and 0.031 cm$^{-1}$, based on Figure 3 of Heimsath et al. (2012). On a more practical level, I don't understand how we, as a community, could make significant progress on understanding the controls on $P_0$ values if we accept the logic of Heimsath and Whipple that only two $P_0$ values can be reliably determined from 57 CRN analyses.

Heimsath and Whipple criticize my inclusion of slope as a control on $P_0$ values on the basis that microclimate cannot thus be untangled from the slope controls proposed by Heimsath et al. (2012) (point 3). However, Heimsath et al. (2012) demonstrated a slope control on *soil production rates* (*P* values) whereas I am considering *potential soil production rates* ($P_0$ values). Slope exerts a strong control on *P* values because steeper slopes drive higher erosion rates (up to a maximum value of $P_0$), thinner soils, and hence higher soil production rates as a result of the generally inverse relationship between soil production rate and soil thickness. In contrast, as I explained on p. 3, lines 17-20, $P_0$ values "do not depend on soil thickness and its controlling factors; hence, they isolate the effects, if present, of environmental factors (e.g., water availability, vegetation cover,

wildfire severity and frequency) and material factors (e.g., bedrock fracture density and lithology/mineralogy) that influence soil production rates." Slope appears as an input to the model of section 2.1 for south-facing slopes. Its inclusion is necessary to capture a real feedback in the natural system, i.e., that steep, south-facing slopes of the SGM, as the type habitat for chaparral (Holland, 1986), are prone to frequent, high-severity wildfires (Keeley and Zedler, 2009) that, in turn, have high rates of soil production if soil cover is thin or absent (Blackwelder, 1927; Goudie et al., 1992; Dorn, 2003; Shtober-Zisu et al., 2010). High rates of soil production, in turn, permit higher erosion rates and thus steeper landscapes because rock uplift rates increase where rates of erosional unloading are higher. I demonstrated that $P_0$ values are controlled by slope aspect, especially at the extremes of directly south- and directly north-facing slopes (Fig. 3B), so it is impossible that the dependence of $P_0$ values on microclimate is some type of spurious result based on slope gradient alone.

The conceptual model presented by Heimsath and Whipple (point 4) is missing a key element: soil production. They propose that more rapid rock uplift produces the higher elevations, greater relief, and more rapid erosion in the eastern portion of the SGM. Heimsath and Whipple mention soil production as a relevant process only once the gradient in elevation, relief and erosion rate is established based on structurally controlled spatial variations in rock uplift rates. However, *soil must be produced* in order for erosion to occur (in the absence of widespread landsliding within bedrock or intact regolith). If $P_0$ values are lower than rock uplift rates in granitic landscapes for a sustained period, cliffs, tors, and stepped topography form (e.g., Wahrhaftig, 1965; Jessup et al., 2010), features that occur in many granitic mountain ranges but that are especially common in arid environments where $P_0$ values tend to be low. The formation of cliffs, tors, and stepped topography may be enhanced by a feedback in which steeper slopes result in thinner soils and *lower* erosion and soil production rates, as evidenced by numerical models (e.g., Strudley et al, 2006) and global CRN datasets that demonstrate lower average erosion rates in bare bedrock landscapes compared to soil-mantled landscapes (Hahm et al., 2014; Figure 4). As such, the correlation between topographic steepness and erosion rates and the relative absence of cliffs, tors, and stepped topography in the SGM is not inevitable but is directly related to the fact that $P_0$ values increase with erosion rates. The central question, then, is *how $P_0$ values and erosion rates are correlated. Is it that erosion rates control $P_0$ values, as Heimsath et al. (2012) proposed? Or, is it that $P_0$ values, controlled by bedrock material properties (e.g., lithology and fracture density) and the abundance of weathering agents/events (e.g., water availability, vegetation cover, wildfire severity and frequency), control erosion rates because erosion cannot occur faster than soil is produced, together with the fact that $P_0$ values and erosion rates have similar bioclimatic controls? I leave it for the reader to decide, but note that Heimsath et al. (2012) and Heimsath and Whipple have not identified a mechanism by which erosion rates might control $P_0$ values, nor have they evaluated the possibility that a factor besides erosion rates (e.g., climate, microclimate, lithology, rock damage, etc.) might control $P_0$ values in the SGM.

In my conceptual model I emphasized the importance of spatial variations in rock uplift rate (e.g., p. 9, line 16). These variations depend, of course, on fault and fold geometries but also on erosion rates because the erosional unloading of faults and crustal roots drives deformation and rock uplift (including both active and isostatic uplift). Heimsath and Whipple state that it is impossible for

rock uplift rates to be influenced by erosion rates at the spatial scale of an SGM block (~10 km) (point 4). Since they provide no explanation, I can only guess what they might be thinking. If Heimsath and Whipple are referring to the fact that the rigidity of the lithosphere resists deflection in response to sufficiently small-scale spatial variations in erosional unloading, I agree in general but note that in a mountain range such as the SGM with pervasive crustal-scale faults, the effective elastic thickness of the crust can be as low as ~3-5 km and the associated flexural wavelength can be as low as ~5-10 km (e.g., Lowry and Smith, 1995). As such, spatial variations in erosion rate can certainly result in spatial variations in rock uplift rate at these scales. All numerical models of mountain belt evolution clearly demonstrate that erosional unloading affects the force balance on faults and crustal roots, which, in turn, affects spatial patterns of rock deformation and uplift.

Heimsath and Whipple argue that some of the relationships I claim to be linear might be nonlinear (point 1). My analysis explicitly included the possibility of a nonlinear dependence between $P_0$ and $D$ (and also between $P_0$ and $A$) by fitting the data to a power-law function. The relevant text on this point (p. 6, last paragraph) is as follows: "To constrain the mathematical form of the relationships among $P_0$, $D$, and $A$, I performed a multivariate linear regression of the logarithms of $P_0$ to the logarithms of both $D$ and $A$. Transformed in this way, the best-fit coefficients obtained by the regression are equivalent to the exponents of power-law relationships of $P_0$ (the dependent variable) to $D$ and $A$ (the independent variables). This regression yielded exponents of $1.1 \pm 0.4$ and $1.1 \pm 0.3$ for the relationships of $P_0$ to $D$ and $A$, respectively. These values are sufficiently close to 1 that I chose to fix the values of the exponents to 1 (i.e., eqn. (2)) for simplicity and reanalyze the data to determine the value of $c_1$ that yields the best fit of equation (2) to data." The fact that the exponents of the best-fit relationship are 1 (within uncertainty) is a *data-driven result* that excludes the possibility of a significantly nonlinear relationship between $P_0$ and either $D$ or $A$. Regarding slope aspect, I am using the *standard microclimatic index* used by many previous studies and that has been shown to correlate strongly with variations that depend on slope aspect and gradient (e.g., ground surface temperatures, potential evapotranspiration rates, vegetation cover).

There is an implicit criticism throughout Heimsath and Whipple's review that I am using proxies for rock damage and microclimate rather than more fundamental variables such as fracture density, soil temperatures during wildfires, etc. Clearly it is impossible to measure these variables over the spatial scale of an entire mountain range, hence the use of proxies is essential if progress is to be made at these spatial scales (and progress should be attempted at all spatial scales). The use of proxies to constrain properties that cannot readily be measured due to challenges associated with large spatial or temporal scales is as old as Geosciences itself.

Heimsath and Whipple state that there is no reason why the power-law relationship between bedrock fracture density and the distance from faults established by Chester et al. (2005) and Savage and Brodsky (2011) for portions of the SGM should apply at the landscape scale (point 1). Heimsath and Whipple are correct that most of the data establishing the scaling relationship between fracture density and the distance from faults that I referenced extend only to distances of ~100 m from faults. To address this point, in my revision I will expand the discussion to include studies from other regions that demonstrate power-law scaling between fracture density and the

distance from faults out to larger distances (e.g., Sturzenegger et al., 2007). In addition, I will include a discussion of the geometric properties of fault networks (e.g., length, spacing, slip) that demonstrate robust power-law scaling up to spatial scales of ~10 km in southern California (e.g., Bonnet et al., 2001). Despite Heimsath and Whipple's dismissal of my approach, the use of a scaling relationship between fracture density and distance from faults, together with the best-available geologic maps to identify the faults, is both reasonable and perhaps the only possible approach to quantifying range-scale variations in rock damage at this time. Heimsath and Whipple further state that "the damage index $D$ is determined from fault traces on existing geologic maps, and likely has little bearing on the actual pattern of rock damage." How would these reviewers suggest that rock damage be more precisely quantified at these large spatial scales? I am not sure why "existing" geologic maps are insufficient, as they reflect careful mapping by many scientists over many decades.

Heimsath and Whipple criticize me for not including precipitation and temperature in my analysis (point 1). My analysis includes vegetation height, which is an effective integrator of precipitation and temperature that has the advantage of being mappable at high resolution over the entire mountain range, unlike precipitation and temperature, which must be interpolated from a few climate stations. The influence of range-scale climate variations was explicitly addressed in the last paragraph of page 8: "The largest $P_0$ values increase and then decrease with elevation between 1.5 and 2.5 km elevation, as indicated by the dashed curve that defines the envelope of the data in Figure 4A…. Mean canopy height, constrained from the Existing Vegetation Height layer of the U.S.G.S. LANDFIRE database (U.S. Geological Survey, 2016), follows a similar pattern to that of $P_0$ (Fig. 4B), correlating positively with elevation below 1.8 km a.s.l. and negatively with elevation above 1.8 km due to limited energy availability, especially in the cold-season months when most precipitation falls in the SGM. Figures 4A&4B suggest that $P_0$ may have some dependence on range-scale climate or vegetation." Moreover, I explicitly addressed why relationships between erosion rates and precipitation are difficult to interpret uniquely: "… the strong correlation that Spotila et al. (2002) documented between exhumation rates, elevation, and MAP. Spotila et al. (2002) cautioned, however, that this correlation could be coincidental as 'prevailing winds happen to deliver the most precipitation along the southern range front where the most active structures are.'"

I agree with Heimsath and Whipple that range-scale climate variations likely control $P_0$ values (point 4), and that spatial variations in uplift play an important role in controlling range-scale climate variations. I proposed that range-scale climate variations may control $P_0$ values (p. 9, lines 3-4) and I also emphasized the importance of tectonic uplift rates in my conceptual model (p. 9, lines 18-20), but I did not explicitly connect the two points as perhaps I should have. I did not include range-scale variations in climate and explicitly in my conceptual model because I could not document a statistically significant relationship between range-scale climate and $P_0$ values (in part because the relationship between $P_0$ values and range-scale climate is not simple or monotonic) and because I could not clearly separate range-scale variations in climate from rock damage, as discussed in the last paragraph of p. 8. I did explicitly note that higher potential soil production rates tend to increase erosion rates because erosion rates are limited by $P_0$ values and because both have similar bioclimatic controls. In the revision I will further emphasize uplift as an

important mechanism for generating range-scale variations in climate and vegetation that likely control $P_0$ values and, as a result, erosion rates.

Heimsath and Whipple state that I provide no new data. To their dataset I added proxies for rock strength and microclimate, predicted the interrelationships among $P_0$ values, erosion rates, soil thicknesses, and topographic slopes throughout SGM, and compared those predictions to data. As noted in the manuscript, I also worked hard to find relationships with lithology (using quantitative indices related to mineralogy developed by Spotila et al., 2002 together with the best-available geologic maps) and vegetation cover.

Heimsath and Whipple further accuse me to throwing out data (point 3). All of the statistical analyses I present, save one, include all of the data. As a thought experiment (clearly presented as such), I observed that the correlation between $P_0$ and $D$ values would improve (to 99.9%) *if* five data points from an area with an unusually high density of mapped landslides were removed. It is well established in the literature that landsliding, as a more episodic transport process than creep or bioturbation, can adversely influence soil production rates determined from CRN analyses. This point may be a distraction, so I will remove it from the revision.

Heimsath and Whipple declare that the correlations I obtained are extremely weak and not sound. I agree with Heimsath and Whipple that the correlation coefficients are low, a point that I discussed at length on p. 10, lines 10-16. Low correlation coefficients do not change the fact that the null hypotheses that $P_0$ values are unrelated to $D$ and $A$ can be rejected with 98.6% and 99.9% confidence, respectively.

Heimsath et al. (2012) worked hard to create a remarkable dataset and, of course, deserve wide latitude in deciding how the data are interpreted. However, the importance of soil production as a process controlling the evolution of mountain ranges, together with my natural desire to defend my work, stimulated me to take the unusual step of attempting to reinterpret their data. In Heimsath et al. (2012), the authors attacked a paper that I published in 2009 with Craig Rasmussen, referring to it as part of an "entrenched paradigm inconsistent with data" that "greatly exaggerates changes in critical-zone processes to tectonic uplift." In this paper I have shown that the model I developed with Craig is consistent with the data of Heimsath et al. (2012) (even though it was not developed specifically for the SGM) if the effects of rock damage and microclimate are taken into account using the best-available proxies.

I wish to thank Heimsath and Whipple for an engaging set of comments.

**References not cited in the discussion paper**

Bonnet, E., O. Bour, N. E. Odling, P. Davy, I. Main, P. Cowie, and B. Berkowitz (2001), Scaling of fracture systems in geological media, Rev. Geophys., 39(3), 347–383, doi:10.1029/1999RG000074.

Jessup, B. S., S. N. Miller, J. W. Kirchner, and C. S. Riebe (2010), Erosion, Weathering and Stepped Topography in the Sierra Nevada, California; Quantifying the Dynamics of Hybrid (Soil-Bedrock) Landscapes, American Geophysical Union, Fall Meeting 2010, abstract #EP41D-0736.

Langston, A. L., Tucker, G. E., Anderson, R. S., and Anderson, S. P. (2015), Evidence for climatic and hillslope-aspect controls on vadose zone hydrology and implications for saprolite weathering. Earth Surf. Process. Landforms, 40, 1254–1269. doi: 10.1002/esp.3718.

Lowry, A. R., and R. B. Smith (1995), Strength and rheology of the western U.S. Cordillera, J. Geophys. Res., 100(B9), 17947–17963, doi:10.1029/95JB00747.

Strudley, M. W., A. B. Murray, and P. K. Haff (2006), Regolith thickness instability and the formation of tors in arid environments, J. Geophys. Res., 111, F03010, doi:10.1029/2005JF000405.

Sturzenegger, M., M. Sartori, M. Jaboyedoff, and D. Stead (2007), Regional deterministic characterization of fracture networks and its application to GIS-based rock fall risk assessment, Engineering Geology, 94(3-4), 201-214, doi:10.1016/j.enggeo.2007.08.002.

Wahrhaftig, C. (1965), Stepped Topography of the Southern Sierra Nevada, California, Geol. Soc. Am. Bull, 76(10), 1165-1190, doi: 10.1130/0016-7606(1965)76[1165:STOTSS]2.0.CO;2.

---

## Referee Comment (RC2) · A. Heimsath (Referee) · 30 Sep 2016

We stand by our criticisms of the manuscript in review. We do not offer a point-by-point response to the rebuttal of our review comments, but we do offer a few clarifications that may help.

1. Heimsath et al. (2012) did not attack any papers. They argued that models built on the previous paradigm would greatly exaggerate changes in critical zone processes in response to tectonic uplift and commensurate increases in erosion rate. The previous paradigm is that $P_0$ is dictated by climate and rock properties and is independent of

erosion rate. When that paradigm is used models require that soils thin as erosion rates increase such that landscapes become rocky and soil-free as soon as erosion rates exceed P0. Stripping off the soil is a rather dramatic change to the critical zone. One main point of Heimsath et al. (2012), described in point 2, is that soils in fact persist in areas of high erosion rate, and thus critical zone processes and conditions are much less sensitive to tectonics than predicted by most landscape evolution models that embed a fixed rule for soil production.

2. Heimsath et al. (2012) emphasized controls on P0, which they termed SPmax. In fact, the sentence in the author's rebuttal, ".... the correlation between topographic steepness and erosion rates and the relative absence of cliffs, ... in the SGM is not inevitable but is directly related to the fact that P0 values increase with erosion rates", is a good summary of the main point made in that paper. The paper also expressed that this link has not been mechanistically explained – indeed it has yet to be replicated or confirmed empirically – and merits further investigation.

3. Given the inherent scatter in soil thickness and local P values (soil production rates) and the uncertainty in 10Be measurements, in most circumstances at least 10 P values are needed to define a robust soil production function, from which a single estimate of P0 and its uncertainty can be determined. In the approach used in the reviewed manuscript, local scatter attributable to many potential influences is interpreted in terms of variation in P0. This is unwise. At best one might hope to subdivide the SGM data into 2-5 subsets for evaluation of P0, had the data been collected to effectively sample across transects in the variables of interest, such as aspect. In preparation of Heimsath et al. (2012) we explored many potential subdivisions of the data to look at controls on P0, but settled on only having confidence in the two subsets presented. It is possible that further subdivision in terms of rock properties, climate (or microclimate), or erosion rate may indeed prove useful. However, additional data would be needed to refine estimates of P0 as a function of the variables of interest.

Arjun Heimsath and Kelin Whipple

**ESurfD**

Interactive
comment

---

## Referee Comment (RC3) · Anonymous Referee #2 · 7 Oct 2016

Anonymous review of Pelletier, eSurf submission 2016

I probably should start by disclosing the fact that also I reviewed a previous version of this manuscript that the author submitted for publication in Geology. This is relevant because It is very clear to me that the author took advantage of relaxed space constraints here at eSurf to address some of my and other reviewers' suggestions. This is commendable, and as a result, this version of the MS is significantly improved in my opinion. Nevertheless, there are still some major flaws – including an especially big one that I pointed out in my review of the Geology MS but that the author did not

address here.

Thus, despite the many changes that the author incorporated in this revision, and despite the fact that the manuscript is topically very well suited to the audience of eSurf, I do not think that the key conclusions are justified by the data and the analysis as presented. Hence, I do not think it is suitable for publication at this time. Big revisions are needed – and I frankly think a major overhaul of both the approach and discussion are needed to remove the impediments to publication in a revised submission to eSurf or a new submission to a different journal.

*********Here are my big general concerns in no particular order:

(1) First, I concur with reviewers Heimsath and Whipple on the fact that it is nonstandard at best to assign a separate P0 value to each measured P value using exponential scaling relationship in Eq 1. It would be equivalent to predicting different values of a y intercept in a linear regression of y on x when you know the slope of the regression and the value of y and x for each data point. There is only one y intercept per regression through a cloud of data. This business of inferring one y-intercept per data point strikes me as – at best – a kooky way (i.e., that differs from established norms) of quantifying the uncertainty in the y intercept.

Based on what I can see in their reviewer comments, Heimsath and Whipple had the same reaction. And the author's response to their comment – i.e., "On a more practical level, I don't understand how we, as a community, could make significant progress on understanding the controls on P0 values if we accept the logic of Heimsath and Whipple that only two P0 values can be reliably determined from 57 CRN analyses" – is not compelling. The alternative logic of Pelletier seems to be that we should suspend conventions of statistics and stretch data farther than they can be stretched just to support some as yet non mechanistic formulation that he has presented here. I prefer the less radical option of recognizing the limits of data and working to overcome them in more traditionally acceptable ways – i.e., with new measurements and perhaps a more clever

analysis approach. For example, as an alternative to the methods presented here, the author might think of ways to model P rather than P0 using some sort of multiple regression analysis that includes h explicitly in a model of rock damage and microclimatic effects. This business of calculating a P0 effectively corrects for the exponential-with-depth variation in P before the modeling begins. In a true multiple nonlinear regression, one could account for h, D, A and everything else simultaneously, and as an outcome of the approach also quantify the relative importance (leverage) of each variable in the regression. If the outcome is that h dominates while D and A add little to the predictive power of the model, then the author would be forced to confess that D and A are not strong predictors of P and thereby P0. And as I point out below, there is good reason to suspect that that is precisely what he would learn.

(2) Second – and this is a bigger concern in my view – is the degree to which the predicted values of P0 ***diverge*** from the observed values in the dataset. It seems like a key goal in this paper is to use indices of rock damage and aspect to predict P0 and ultimately map the variations in P0 and some additional offshoots of it (E and E*L) onto the landscape. Starting at section 2 and continuing through to the end of this paper, this is actually ***the*** central focus of results and discussion. The trouble is, one must be willing to believe that the model in Eq 2 is a good predictor of P0 in order to confidently follow the author in this vital leap of faith.

Personally, after reading this paper, I am not willing to make that leap. Nor should any self-respecting data analyst, once he/she realizes that the predicted values are not actually a very good match to the observations. Sure – as the reviewer points out – there is a highly significant correlation between the measured and predicted values P0, but the existence of such a correlation is not sufficient in and of itself to demonstrate that the predictions are good enough to explain the variations in P0. Recognizing this challenge nearly fifty years ago, hydrologists Nash and Sutcliffe (1970) developed their own measure of model efficiency in their quest to objectively evaluate whether their models of river discharge were good predictors of observations. Though this Nash-

Sutcliffe (N-S) statistic – as it later became known – was developed for models of river flow, it has also been widely used to assess model efficiency for other natural variables, including erosion rates and nitrogen and phosphorus loading.

My quick calculation of a N-S efficiency statistic for the model of predicted P0 yields a value of 0.18 based on data provided in the supplemental table. For context, realize that the maximum value of the coefficient, which is 1, would indicate that the model explains \*\*\*all\*\*\* of the observed variation in the data. By contrast a value of 0 would indicate that the model is \*\*\*just as good as\*\*\* the average value of the data at explaining observed variations across the data set. Values less than 0 imply that the model \*\*\*is worse than\*\*\* the average. In this case, my estimated value of 0.18 indicates that \*\*\*the model in Eq. 2 is a little bit better than the average P0\*\*\* at predicting the distribution of measured P0 across the dataset. For this reason, I think the machinations of the predictive modeling exercise (i.e., most of the paper) are not really warranted , irrespective of the significance of any inferred correlations between P0 and D and between P0 and A. Importantly the reader should only commit to believing those correlations to the extent that he/she can overlook the suspect exercise of calculating P0 for each measured value of P.

Ultimately, it is not clear to me that the author understands that there is a vital difference between documenting a statistically significant correlation between a measured and predicted value and demonstrating that a model is good at doing what it is supposed to do. If he does, he is hiding it at the top of page 7, where he seems to suggest that statistical significance in classical regression metrics is sufficient. I will not deny that the correlation coefficient and thus the coefficient of determination by themselves provide a very loose first approximation of model fitness. But even then, this is true only to the extent that high coefficients of determination (close to one) imply better correlations and low coefficients imply poor correlations (irrespective of whether they are statistically significant). To understand the problem with using $R^2$ in the way the author seems to want to use it here, consider the toy example in which P0 observed

is exactly 0.2 times the value of P0 predicted for each inferred value of P0; in that case, the coefficient of determination of P0 predicted and P0 observed would be 1.0 with a very very low p value even though the predicted value of P0 is 5 times higher than the observed value at each site. P0 predicted is dead wrong but the coefficient of determination is fantastically good. This illustrates how simple correlation indices for predicted versus observed data sets can (and probably often do) fall short on gauging the predictive power of a model.

Disclosure: This comment is more elaborate but ultimately very similar to one that I made in my previous review of the Geology submission. Yet the author did nothing to address this concern in this revised manuscript. Although the predicted values are a bit different in this iteration (mostly reflecting changes in chosen values of L/k and Sc, I think), my comment still stands. Nothing about the revised paper has changed the fundamental problem that the model does little better than the average value of P0 at predicting P0 measured here. This seriously undermines the story. While it is probably true that the author is no longer quite as assertive as he was in round 1 about the role of microclimate and faulting in controlling landscape evolution, the words in the abstract and conclusion sections leave no doubt that he really believes – and wants his readers to buy the idea – that they play "subequal" roles along with tectonics in controlling P0. And he unambiguously (but unjustifiably) points to the statistical significance of $R^2$ values of the relationships between predicted and observed values of P and P0 as his metrics of model fitness (page 7 lines 1 and 2) – which, as noted above, are not definitive, even when the $R^2$ is much higher than the low values reported here.

—- On a side note, when I plotted the P0 measured and P0 predicted values in the supplemental table against each other, I get a pattern that looks slightly different than the one shown in the figure. The differences are not big enough to explain away the problem of low Nash-Sutcliffe statistics (Fig 3D and 4C), but it made me worry that the author has some version inconsistencies between his figures and the data he provided in the table. Not sure which version is "correct."

(3) Like Heimsath and Whipple, I am unimpressed with the theoretical basis of Eq 2, and moreover, I am not compelled by the author's response – i.e., "It is far beyond the scope of the paper to develop a comprehensive theory for how microclimate relates to vegetation cover, wildfire frequency and severity, and soil production rates, assuming such a theory is even possible." However, whereas Heimsath and Whipple rightly seem very worried about how D might connect to rock damage at landscape scales, and how those variations in D would actually connect to P0 in a mechanistic way, I am stuck on the fact that the authors never actually showed me that aspect should matter at SGM.

The references cited on page 3 have nothing to do with the effect of aspect on vegetation or the effect of either vegetation or aspect on fire intensity or severity in the SGM. Where is the proof that vegetation, fire frequency, and slope steepness vary with aspect in the SGM? It seems it would be crucial to demonstrate this is the case before motivating the paper and the formulation of equation 2 more specifically. The aspect story fits with some of the author's work in other landscapes but not here – at least not according to the references cited here. If anything, the Keeley and Zedler study seems to suggest that the current regime – in which the landscape is prone to large fires that sweep through the landscape with indifference to aspect – has been the norm for a long time

Additionally, this study seems to hang a lot of its motivation on the idea that fire promotes weathering. But - despite the good investigative work cited on page 3 - I am not sure I concur that the connection has been well documented at SGM. All of the studies cited here are fascinating but ultimately just anecdotal investigations of weathering of boulders – not weathering of rock under soil, which presumably is important here since much of the SGM area is covered by soil. Moreover, they do not report faster weathering rates on fire-prone versus not-fire-prone slopes. In fact, none of the studies actually report rates (focusing instead on processes) and none compare fire-prone versus not-fire-prone slopes. Shtober-Zisu et al. comes close to reporting a rate but ultimately says it is hard to say how the boulder spalls in carbonate outcrops influence denudation rates across the landscape. And again, there is no comparison to a landscape that is not fire prone, so there is no control in the experiment – and importantly no support for the author's claim here that weathering is faster in fire-prone versus not-fire-prone landscapes.

However strong the correlation between P and A may be, I think it is very important for the author to step back from this generic claim that aspect-driven differences in wildfire are driving the show and more precisely drill in on how anecdotal studies from the SGM in particular support the slope aspect idea. Bottom line is there needs to be some stronger motivation here – hopefully shored up some sound mechanistic explanations for why both D (measured in the S&B 2011 approach) and A should matter. I do NOT think it is "beyond the scope" of this paper to justify the formulations that it presumes to impose broadly on the landscape.

(4) The statistical analyses are nonstandard. My discomfort with them is very high. My discomfort started with the first indication – I think on page 5 – that the author thinks of statistical significance as the logical and quantitative complement to a calculated p value. This is not the case, of course. Rather "significance" is commonly reserved referring to the threshold false positive rate that is allowed in a statistical hypothesis test. So the idea that the author thinks that a calculated $p = 0.001$ corresponds to a "statistical significance" of 99.9% set me on edge. This misappropriation of terminology was repeated many times throughout the text. But that was just the start. The author also evidently thinks it is ok to calculate a y-intercept for each measured value in a dataset using an overall regression slope that was calculated from the entire data set – and which also yields an overall regression intercept. To be honest, this seems akin to data fabrication to me, but I can settle on the gentler view of Heimsath and Whipple that it is really just of a crude way to estimate the uncertainty in the intercept. Next, the author follows a rather stilted approach to quantifying the relationship between P0 and D and A. I personally think it should be P versus D, A and h, thus recognizing h as a factor regulating P and avoiding the problem of getting just two P0 values from

57 values of P. In addition, I think the author missed an opportunity to perform a very standard multiple regression analysis on log-transformed variables and instead opted for a multi stage approach that undoubtedly underestimates errors and fails to produce vital outputs like leverage plots and partial regression coefficients which would help the audience gauge the relative importance of the different factors in the regression. In addition, there is no attempt to propagate uncertainties through any of this. This is a major oversight that needs to be fixed. Last and not least, the author also thinks it is ok to use the significance of $R^2$ for the relationship between predicted and observed values to judge the performance of his model. In the hydrology community that idea has been rejected for nearly half a century. I am very concerned about the strength of the analyses for these reasons.

*********Specific comments keyed to page numbers and line numbers. For example 5.2 is line 2 on page 5 of the PDF. Note: Although some of these comments pertain to minor issues, others are just as important to address as the ones outlined above.

2.10. I see that Heimsath and Whipple have provided a review of the manuscript and will defer to them as experts on evaluating this paragraph as a motivating theme for the paper. They did not call attention to any problems here. However, as I read line 21 on this page, I guess I have to say that this was not the take home message I got from Heimsath et al., 2012. Higher frequency of disturbance?

3.6-3.10. This study seems to hang a lot of its motivation on the unsupported idea that aspect promotes differences in vegetation which in turn promote differences in fire that promote differences in weathering in the SGM area. See general comment above.

3.23-3.25. This is non-standard to say the least. See general comment above.

4.11. I think I understand what the author is trying to do here (correct the measured P0 for the hump in the SPF), but on reading this, I am confused. You used 1.78P for P0? Not 1.78P0? The way I want to read it is the author is correcting the "measured" P0 – which is inferred from the exponential function to the data – by some correction factor.

But again, I am confused by this statement.

4.12. "This modification of equation (1) affects 4 of the 57 data points." This would only be comforting if there was actually a very strong trend across all the data. Instead, it seems that the data form really loose clouds of correlations that are hinged entirely on a few points. So the fact that this affects 4 of the points is actually troubling – not comforting – to me.

5.3. This equation does not include the fault specific constant of Savage and Brodsky. So I think this assumes that the constant is the same across the study area. Is this justified? Also, to make D dimensionless wouldn't delta x need to be raised to the 0.8 power too?

5.10. ***This is very important.*** The line plotted in Fig. 3A is a log-log regression that ignores the cluster of five data points circled in the figure. There is NO justification for ignoring these points!!! He says in line 5.20 that they occur in an area of unusually dense landslides. I do not see this in figure 1!!! Even if I did, it would not justify excluding them from the analysis. Heimsath and Whipple seem to agree. I think it is complete nonsense. Makes the line look steeper than is should be. Sweeping these points under the rug does not make them go away. Including them in the regression would undermines his story that D plays a "subequal" role with tectonics.It not only looks suspicious. It is suspicious. Author needs to HONOR the data in this study and in his other work and not try to sweep data points away like this.

6.10. I do not understand why the correlation would shut off on north-facing slopes. Is there a mechanistic/theoretical basis for this? If not than the relationship is purely empirical.

6.20. Some more non-standard statistical machinations. The author does a regression that suggests that the power law exponents of A and D are 1.1 +/- some error. Then he reanalyzes things assuming that they are 1 to determine the value of c – the constant in front of A and D in Eq. 2. I am at a loss here. I know the author to be very bright and
competent quantitatively. Yet here he invoking using some unnecessary, non-standard, and potentially misleading steps to avoid what would be a fairly straightforward multiple regression analysis of all of the parameters (slopes and intercepts) implied by a power law formulation of Equation 2. Doing this in a more standard way would yield some very useful metrics like partial correlation coefficients and leverage plots. Perhaps his approach seemed easier to explain at the time he wrote it. But I would argue that the community deserves and expects more.

7.1-7.2. This is actually not a very good correlation for predicted versus observed – especially since it is strangely for a log-log plot. To understand this, look at the plot. There is almost an order of magnitude of variation in predicted P0 at any given value of P measured. To evaluate this model, rather than see an R^2 for a log-log observed versus predicted plot, I think we need to see something like a Nash-Sutcliffe statistic, which would tell us how good the model is compared to simply assuming that we could use the average P measured to estimate P everywhere.

7.5 What are the assumptions inherent in simplifying the equations in this way? Simply citing off to previous work here is not sufficient. What are the assumptions inherent in doing this? For equation 6 you assume slopes are planar, right? Is that reasonable here? What are the limitations of removing the higher order terms of Roering et al.?

7.19. Why 0.03? Just because this is the minimum finite thickness measured? But the whole point is they have no thickness!!! The mathematical inconvenience of having a value of 0 on what you want to plot on a log scale does not justify making up a value that ***drives*** a regression that you then plot through the data. Importantly it is very true that these points have a lot of leverage on the regression. Since calculating un-derstanding the relationship between h and S is vital to calculating E from topography, this ends up being key to the paper. And I really do not think it is well justified.

8.8-8.17. I don't buy the predicted values of P0 so I guess none of this mapping of E and E*L onto the landscape really resonated with me.

9.8. If this is the key result, then you need to demonstrate it using more conventional statistical approaches. A multiple linear regression of the log of P versus log D and h and log A would be a good place to start. This would avoid the strange – and thus hard-to-justify – correction of P to P0 that you have employed here. It would also avoid the strange practice of finding a 1.1 +/- error power slope and then redoing the regression assuming the slopes are 1 to find the best fit intercept term. This whole analysis seemed like a contorted and potentially error-prone way of doing what could have been a textbook application of multiple linear regression analysis on transformed variables.

10.12. This is misleading at best. I see a factor of 2 to 3 in either direction, so a factor of 4 to 6 overall. For example, in Fig 3D, at a value of P0 observed of ∼150 m/My I see a range of predicted values running from 85 to 450 m/My. That's a factor of nearly 6 range in predictions for a single value of P observed. That is NOT a good prediction in my book and my assertion is asserted by the very low N-S statistic for this modeling exercise.

11.6. I disagree. This has \*\*\*not\*\*\* been documented. The fundamental data are suspect (calculations of P0 from P and slope of exponential regression) and the analysis of relative effects of D and A is non-standard.

11.8. Definitely not shown. The business of mapping E*L onto the landscape and comparing it to topography is fundamentally undermined by the lack of predictive power of equation 2, which is demonstrated by the very low N-S stat for the comparison between predicteds and observeds.

---

## Editor Comment (EC1) · J. Braun (Editor) · 7 Oct 2016

Reading through the two reviews we have received concerning your manuscript, it is clear that it raises many questions from the reviewers. Their comments and criticisms concern both the methods you have used to interpret the data and the assumptions on which the interpretation is based. At this stage, it is critical that you respond to the second reviewer's comments and criticisms before considering preparing a revised version of the manuscript. It seems to me that both reviewers (or sets of reviewers as both Heimsath and Whipple signed the first review) request major modifications

to the methodology you have used which might strongly impact your conclusions. I look forward to your detailed rebuttal and/or comments following the second reviewer's detailed assessment of your manuscript for which I thank him.

**ESurfD**

---

## Referee Comment (RC4) · S. M. Mudd (Referee) · 12 Oct 2016

In this paper Jon Pelletier has used the dataset from Heimsath et al 2012 to explore controls on soil production. In the original paper, Heimsath and colleagues argued that rapid erosion rates could affect the $P_0$ term in the soil production function. The obvious follow on question is: by what mechanisms does erosion rate modulate $P_0$? As stated by Heimsath and Whipple's comment (doi:10.5194/esurf-2016-37), the original 2012 paper did not mechanistically explain observed trends. So, does Pelletier's paper give insight into the mechanisms? Firstly we can look at the damage indicator. I found this

interesting since many authors have speculated on the role of fracturing in controlling weathering rates, and the implementation of equation (3) is a novel attempt to translate mapped faults into a metric for fracture density using results from detailed field studies. To compare this metric with soil production, Pelletier calculates $P_0$ from every data point by regressing the soil production function, using a slope of $h_0$ previously regressed in the Heimsath et al paper, to its $h = 0$ intercept. To do this, one must assume that the individual $P_0$ results are meaningful and not simply the results of scatter in the data due to local heterogeneities in shielding and erosion history; Heimsath and Whipple feel this unwise, a point I will revisit later in this comment. However once Pelletier follows this thread he finds a weak correlation between the $D$ metric and $P_{0,regressed}$ data (I'm not sure if I'd be so bold as to call it measured). One can explain 10What about aspect? There are a few rather high $P_{0,regressed}$ values for south facing slopes. Of the 11 points with $P_{0,regressed}$ values greater than 300 m/Myr, 8 of them are on south facing slopes. But there are also a large number of points on south facing slopes that don't have $P_0$ values that are higher than the mean $P_0$ value. The model combining topographic gradient and aspect again shows a correlation between it and the $P_0$ values, this time explaining 30I am somewhat confused by section 2.2. It seems strange to generate a map of steepness and from that calculate the spatial distribution of $h$ and $E$. Global topographic maps are readily available so why calculate $S$ from equation (8), which contains many assumptions, rather than just use topographic data? It also seems quite odd to use equation (7) since theory suggests that for a given erosion rate and $P_0$, hillslope-scale gradient will vary as a function of hillslope length. More explanation of these choices is warranted. It is worth commenting on the use of scatter in soil production data to regress $P_0$ values for individual samples. Because these numbers were collected at specific points in the landscape (i.e., they are not basin-averaged data), one must consider if the local sources of scatter. Suppose one measured 10 $P$ values in close proximity (e.g., in a 15 m radius): how variable would those $P$ values be? We don't actually know how representative the $P$ values are on a local scale, but we know soil thickness can have quite a bit of local variability, chemical weathering

can have substantial local variability, and you can have substantial local variability in the production of 10Be (from where snow falls, any transience in erosion history, etc.). So I do not think Heimsath and Whipple's concern about interpreting the $P_0$ values is unwarranted: I share this concern. So, in summary, I am worried that the potential uncertainties in P values makes it difficult to come to strong conclusions about influences of other factors on $P_0$, that even if you believe the $P_0$ values are representative the correlation with D is rather weak, and that I do not feel the effects of aspect have been sufficiently separated from gradient effects.

---

## Author Comment (AC2) · 16 Nov 2016

I provided a preliminary response to Heimsath and Whipple on Sept 25. Below is a brief summary of the main points of the Heimsath and Whipple review and my revised responses.

My $P_0$ estimates are simply the residuals obtained from the regressions of Heimath et al. (2012). Far from being "inappropriate" and "unwise" (reviewer 1), "kooky" and "akin to data fabrication" (reviewer 2), or any of the other descriptions employed by the reviewers, computing residuals and testing for additional controls is a recommended step in regression analysis. Heimsath and Whipple define $P_0$ as the y intercept of regression of soil production rates to soil thickness. This definition excludes the residuals of the regression without any basis. I define $P_0$ (in the first sentence of the abstract and again in the first sentence of the introduction) as the maximum soil production rate at each point on Earth's surface. My definition honors the fact that $P_0$ values may vary continuously in space and that regressions of soil production rates to soil thickness yield a set of residuals that can and should be tested for additional controls. Residuals are estimates, since the regression used to compute the residuals has uncertainty. However, the fact that there is local variability in P values and uncertainty in 10Be measurements does not provide a basis for ignoring the residuals of this or any other regression. If local variability and/or data uncertainty dominate a soil production rate dataset, then no statistically significant landscape-scale controls will be identified in the residuals. For example, if large spatial variations existed in $h_0$ (the decay length scale of the soil production function) in the SGM, $P_0$ variations would be highly uncertain and controlling factors impossible to detect. However, Heimsath et al. (2012) estimated that $h_0$ values differ by 0.05 m (0.32 m vs. 0.37 m) between portions of the SGM with the largest difference in $P_0$ values. At a soil thickness of 30 cm, this difference corresponds to $P_0$ differences of approximately 10% (i.e., exp(-0.30/0.32) vs. exp(-0.30/0.37)). This difference is more than 100 times smaller than the variation in $P_0$ values. This difference becomes even smaller for soils thinner than 0.3 m.

Heimsath and Whipple question the processes included in my model. I have thought hard about what factors, besides variations in fault density and vegetation cover (and its associated wildfire regime), may explain the patterns in the data. Near-surface rocks in the SGM are in a highly compressive state (~10 MPa). In compressive-stress environments, the development of rugged topography leads to a reduction in compressive stress (and even the development of tensile stress in sufficiently steep areas) in the rocks beneath hillslopes. This change in stress state can increase the bulk porosity of the rock, allowing weathering agents to penetrate more readily into the rock, thus increasing the rate of weathering for a given soil thickness. In my proposed revision, I demonstrate that the predictions of the topographically induced stress fracture opening hypothesis are more consistent with the data than my previous model. This hypothesis has the benefit of a strong theoretical foundation. Once the data are modeled based on this hypothesis, temperature clearly emerges as a limiting factor for $P_0$ values at the highest elevations of the range.

I regret not nailing this problem in the discussion paper and having to make major changes to the revision (in part because this entails more work for the reviewers). However, major changes were called for by the reviewers and a major overhaul of a manuscript is sometimes a positive outcome of negative reviews (the proposed revision to Section 2.1 is provided below and the proposed revision of the entire manuscript is provided as a separate document). I believe my revised paper provides a needed process-based understanding of the controls on $P_0$ values documented by Heimsath et al. (2012) and establishes a climatic control on $P_0$ values at the highest elevations of the SGM. These results provide a useful foundation for additional targeted 10Be analyses and for the incorporation of new methods that can further test the topographically induced stress fracture opening hypothesis (e.g., shallow seismic refraction surveys, 3D stress modeling, etc.).

In my opinion, the truth that has emerged from this review and my response is an interesting middle ground in which Heimsath et al. (2012) have been vindicated on their fundamental point that P_0 values can increase with topographic ruggedness in some (i.e., compressive-stress) settings, but that also supports the hypothesis they rejected, i.e., that P_0 values are controlled solely by climate and rock characteristics. The evidence remains that it is changes to rock characteristics, i.e., an increase in bedrock or intact regolith porosity in areas of more rugged topography, that lead to higher P_0 values, together with a climatic limitation on P_0 values at the highest elevations of the range.

Proposed revision to Section 2.1:

[revised manuscript text omitted]

---

## Author Comment (AC3) · 16 Nov 2016

**Reviewer 2 (anonymous):**

I wish to thank this reviewer for his thoughtful comments (I will use the male pronoun since the AE identified this reviewer as male). Although I do not agree with some of his comments, I agree with many and the manuscript has been significantly improved based on his review. I greatly appreciate the time he took to engage with the manuscript, both in this round of review and in a prior round for a different journal.

Comment: (1) First, I concur with reviewers Heimsath and Whipple on the fact that it is nonstandard at best to assign a separate P0 value to each measured P value using exponential scaling relationship in Eq 1. It would be equivalent to predicting different values of a y intercept in a linear regression of y on x when you know the slope of the regression and the value of y and x for each data point. There is only one y intercept per regression through a cloud of data. This business of inferring one y-intercept per data point strikes me as – at best – a kooky way (i.e., that differs from established norms) of quantifying the uncertainty in the y intercept. Based on what I can see in their reviewer comments, Heimsath and Whipple had the same reaction. And the author's response to their comment – i.e., "On a more practical level, I don't understand how we, as a community, could make significant progress on understanding the controls on P0 values if we accept the logic of Heimsath and Whipple that only two P0 values can be reliably determined from 57 CRN analyses" – is not compelling. The alternative logic of Pelletier seems to be that we should suspend conventions of statistics and stretch data farther than they can be stretched just to support some as yet non mechanistic formulation that he has presented here. I prefer the less radical option of recognizing the limits of data and working to overcome them in more traditionally acceptable ways – i.e., with new measurements and perhaps a more clever analysis approach. For example, as an alternative to the methods presented here, the author might think of ways to model P rather than P0 using some sort of multiple regression analysis that includes h explicitly in a model of rock damage and microclimatic effects. This business of calculating a P0 effectively corrects for the exponential-withdepth variation in P before the modeling begins. In a true multiple nonlinear regression, one could account for h, D, A and everything else simultaneously, and as an outcome of the approach also quantify the relative importance (leverage) of each variable in the regression. If the outcome is that h dominates while D and A add little to the predictive power of the model, then the author would be forced to confess that D and A are not strong predictors of P and thereby P0. And as I point out below, there is good reason to suspect that that is precisely what he would learn.

Response: I am not asking the reader to suspend the conventions of statistics. A soil production function is the outcome of a regression analysis. A regression analysis yields two types of outputs: the coefficients of the regression equation and a set of residuals. Computing residuals and testing for additional controls is a recommended step in regression analysis.

I defined $P_0$ in the paper as the maximum soil production rate at each point on Earth's surface. To estimate $P_0$ values defined in this way, one begins by accounting for the effects of soil cover (which has the effect of decreasing the soil production rate below its maximum or potential value) by regressing log-transformed P values to soil thickness. Following regression, P values are divided by the regression equation (which is equivalent to subtracting the regression of the log-transformed data) to obtain a set of residuals that can and should be interrogated for additional controls. That is all I have done to estimate $P_0$ values. If the regression of P data to soil thickness yields no statistically significant trend, then there is no statistically significant regression to soil thickness and hence no residuals ($P_0$ values) to study. That is not the case here, as Heimsath et al. (2012) clearly demonstrated that gently and steeply sloping portions of the landscape fit exponential soil production functions with nearly identical decay constants.

I think it is reasonable to ask reviewers to at least consider my definition of $P_0$. However, they simply define $P_0$ differently (as the y-intercept of the soil production function) and then criticize me on the basis

of that alternative definition. I think the two definitions are complementary. I don't think see any reason why the residuals of this particular regression should be ignored when the output of this or any other regression is a set of regression coefficients and a set of residuals, both of which contain important information.

Reviewer 2 joins Heimsath and Whipple in criticizing me for not providing a new suite of CRN-based soil production rate data. I think there is broad agreement in the scientific community that it is appropriate for some studies to focus on measurements and data analysis (e.g., Heimsath et al., 2012) and others to focus on analyses of existing data and modeling/process-based interpretation (this paper). Science would move forward more slowly and with a less diverse range of perspectives if, for example, every study of soil production required a new *in situ* CRN dataset. In this specific case I think it is clear that what is most needed is a process-based understanding of trends in the existing data that can be used to guide additional targeted 10Be analysis. I agree with the reviewers that my *ESurfD* paper did not provide such an analysis, in part because it did not consider the potentially important process of topographically induced stress fracture opening. However, I believe that my proposed revision does provide a process-based understanding that is both consistent with trends in the data and well-grounded in theory.

The reviewer also calls for a multivariate regression to all of the potential controlling variables including soil thickness. I think it is more appropriate to honor the work of Heimsath et al. (2012) (as the reviewer recommends in many of his comments) by using the residuals of their regression as a starting point. The combination of stepwise regression and cluster analysis I use in the revision is based on standard statistical methods. The reviewer may not agree with every step of my revised analysis, but I respectfully ask that he consider it.

 Comment: (2) Second – and this is a bigger concern in my view – is the degree to which the predicted values of P0 ***diverge*** from the observed values in the dataset. It seems like a key goal in this paper is to use indices of rock damage and aspect to predict P0 and ultimately map the variations in P0 and some additional offshoots of it (E and E*L) onto the landscape. Starting at section 2 and continuing through to the end of this paper, this is actually ***the*** central focus of results and discussion. The trouble is, one must be willing to believe that the model in Eq 2 is a good predictor of P0 in order to confidently follow the author in this vital leap of faith. Personally, after reading this paper, I am not willing to make that leap. Nor should any self-respecting data analyst, once he/she realizes that the predicted values are not actually a very good match to the observations. Sure – as the reviewer points out – there is a highly significant correlation between the measured and predicted values P0, but the existence of such a correlation is not sufficient in and of itself to demonstrate that the predictions are good enough to explain the variations in P0. Recognizing this challenge nearly fifty years ago, hydrologists Nash and Sutcliffe (1970) developed their own measure of model efficiency in their quest to objectively evaluate whether their models of river discharge were good predictors of observations. Though this Nash- Sutcliffe (N-S) statistic – as it later became known – was developed for models of river flow, it has also been widely used to assess model efficiency for other natural variables, including erosion rates and nitrogen and phosphorus loading. My quick calculation of a N-S efficiency statistic for the model of predicted P0 yields a value of 0.18 based on data provided in the supplemental table. For context, realize that the maximum value of the coefficient, which is 1, would indicate that the model explains ***all*** of the observed variation in the data. By contrast a value of 0 would indicate that the model is ***just as good as*** the average value of the data at explaining observed variations across the data set. Values less than 0 imply that the model ***is worse than*** the average. In this case, my estimated value of 0.18 indicates that ***the model in Eq. 2 is a little bit better than the average P0*** at predicting the distribution of measured P0 across the dataset. For this reason, I think the machinations of the predictive modeling exercise (i.e., most of the paper) are not really

warranted, irrespective of the significance of any inferred correlations between P0 and D and between P0 and A. Importantly the reader should only commit to believing those correlations to the extent that he/she can overlook the suspect exercise of calculating P0 for each measured value of P. Ultimately, it is not clear to me that the author understands that there is a vital difference between documenting a statistically significant correlation between a measured and predicted value and demonstrating that a model is good at doing what it is supposed to do. If he does, he is hiding it at the top of page 7, where he seems to suggest that statistical significance in classical regression metrics is sufficient. I will not deny that the correlation coefficient and thus the coefficient of determination by themselves provide a very loose first approximation of model fitness. But even then, this is true only to the extent that high coefficients of determination (close to one) imply better correlations and low coefficients imply poor correlations (irrespective of whether they are statistically significant). To understand the problem with using R^2 in the way the author seems to want to use it here, consider the toy example in which P0 observed is exactly 0.2 times the value of P0 predicted for each inferred value of P0; in that case, the coefficient of determination of P0 predicted and P0 observed would be 1.0 with a very very low p value even though the predicted value of P0 is 5 times higher than the observed value at each site. P0 predicted is dead wrong but the coefficient of determination is fantastically good. This illustrates how simple correlation indices for predicted versus observed data sets can (and probably often do) fall short on gauging the predictive power of a model.

Response: I did not include the Nash-Sutcliffe efficiency for the simple reason that it does not apply to regression models. The definition of the Nash-Sutcliffe efficiency is

$$E = 1 - \frac{\sum_{n=1}^{N}(X_{n,obs} - X_{n,sim})^2}{\sum_{n=1}^{N}(X_{n,obs} - X_{mean,obs})^2}$$

where $X_{n,sim}$ are the values predicted by a simulation or other type of model that is *not based on regression*. In cases where the predicted values are based on a regression (as is the case here), the closest analog of the Nash-Sutcliffe efficiency is the coefficient of determination, R^2, defined as

$$R^2 = 1 - \frac{\sum_{n=1}^{N}(X_{n,obs} - X_{n,reg})^2}{\sum_{n=1}^{N}(X_{n,obs} - X_{mean,obs})^2}$$

where $X_{n,reg}$ are the predicted values based on the regression. If these two equations look almost identical it is because they are. When using a regression model, one uses R^2, which I did. When not using a regression model, one uses E.

The reviewer criticizes me for not reporting a Nash-Sutcliffe efficiency despite his request for one in a review of a prior version of the paper for the journal *Geology*. I am not going to include a statistic that, by definition, does not apply to the method I am using.

I don't want to antagonize this reviewer, but I would like to point out that his discussion of the *Geology* review violates GSA's ethical guidelines for publication, which require reviewer confidentiality. I am, of course, glad that the paper was rejected by *Geology* and that my *ESurfD* paper was also negatively reviewed, because this has prompted me to take a fresh look at the problem and redouble my efforts to understand the process basis for the trends in P_0 values in the SGM. This type of rethinking and major revision is often a positive outcome of a negative review. That said, I still think it is reasonable to ask that the review process follow established ethical guidelines. I don't think it is fair or accurate that the literature now suggests that I am not a careful scientist who carefully considers reviewer comments. There are simple reasons, identified here, why I did not explicitly address some of his concerns from the prior review.

The 0.18 value computed by the reviewer is inappropriate because it is weighted towards the errors associated with higher P_0 values since the reviewer did not log-transform the data. Such weighting is

appropriate for many applications, such as modeling discharges of water, sediment, or contaminants, in which the performance of the model must be judged on its ability to predict both individual data points and the integrated value of the quantity under study. Since the integrated value is dominated by the largest values in the dataset, it is appropriate to weigh the errors associated with larger values more heavily in such cases. That is not the case here. There is no reason to weigh samples from areas with larger potential soil production rates more heavily than samples from areas with low potential soil production rates in judging the model fitness. Because the data have a positive skew, it is more appropriate to log-transform the data.

The reviewer poses the case of an independent variable regressed to a dependent variable offset by a factor of 5. A regression of the logarithms of the independent variable to the dependent variable yields a model with no offset (the unique result of the regression to the hypothetical data posed by the reviewer is ln y = ln(0.2) + ln x, $R^2 = 1$). Therefore, the supposed counterexample suggested by the reviewer is impossible using the method I am using (regression of log-transformed data).

I understand very well that there is value in having a low value of p and a high value of $R^2$ (or the Nash-Sutcliffe efficiency, if one is evaluating a simulation model). However, there are many geomorphology papers that are based on regressions with $R^2$ values lower than the ones I obtained (one example: $R^2 = 0.17$ in Nature Geosciences, v. 8, p. 462-465, 2015, Fig. 3a). In my revised paper I obtained $R^2 = 0.50$ ($R^2 = 0.87$ when data with the same $S_{av}$ value are averaged to minimize local variability). I don't know whether this will satisfy the reviewer since I don't know what he considers an acceptable value of E or $R^2$.

Comment: On a side note, when I plotted the P0 measured and P0 predicted values in the supplemental table against each other, I get a pattern that looks slightly different than the one shown in the figure. The differences are not big enough to explain away the problem of low Nash-Sutcliffe statistics (Fig 3D and 4C), but it made me worry that the author has some version inconsistencies between his figures and the data he provided in the table. Not sure which version is "correct."

Response: I could not reproduce this error. I am as certain as I can be that the data presented in the table and plotted in the figures of the proposed revision are the same.

Comment: (3) Like Heimsath and Whipple, I am unimpressed with the theoretical basis of Eq 2, and moreover, I am not compelled by the author's response – i.e., "It is far beyond the scope of the paper to develop a comprehensive theory for how microclimate relates to vegetation cover, wildfire frequency and severity, and soil production rates, assuming such a theory is even possible." However, whereas Heimsath and Whipple rightly seem very worried about how D might connect to rock damage at landscape scales, and how those variations in D would actually connect to P0 in a mechanistic way, I am stuck on the fact that the authors never actually showed me that aspect should matter at SGM. The references cited on page 3 have nothing to do with the effect of aspect on vegetation or the effect of either vegetation or aspect on fire intensity or severity in the SGM. Where is the proof that vegetation, fire frequency, and slope steepness vary with aspect in the SGM? It seems it would be crucial to demonstrate this is the case before motivating the paper and the formulation of equation 2 more specifically. The aspect story fits with some of the author's work in other landscapes but not here – at least not according to the references cited here. If anything, the Keeley and Zedler study seems to suggest that the current regime – in which the landscape is prone to large fires that sweep through the landscape with indifference to aspect – has been the norm for a long time. Additionally, this study seems to hang a lot of its motivation on the idea that fire promotes weathering. But - despite the good investigative work cited on page 3 - I am not sure I concur that the connection has been well documented at SGM. All of the studies cited here are fascinating but ultimately just anecdotal investigations of weathering of boulders – not weathering of rock under soil, which presumably is important here since much of the SGM area is covered by soil. Moreover, they do not report faster weathering rates

on fire-prone versus not-fire-prone slopes. In fact, none of the studies actually report rates (focusing instead on processes) and none compare fire-prone versus not fire-prone slopes. Shtober-Zisu et al. comes close to reporting a rate but ultimately says it is hard to say how the boulder spalls in carbonate outcrops influence denuda tion rates across the landscape. And again, there is no comparison to a landscape that is not fire prone, so there is no control in the experiment – and importantly no support for the author's claim here that weathering is faster in fire-prone versus not-fire-prone landscapes.

However strong the correlation between P and A may be, I think it is very important for the author to step back from this generic claim that aspect-driven differences in wildfire are driving the show and more precisely drill in on how anecdotal studies from the SGM in particular support the slope aspect idea. Bottom line is there needs to be some stronger motivation here – hopefully shored up some sound mechanistic explanations for why both D (measured in the S&B 2011 approach) and A should matter. I do NOT think it is "beyond the scope" of this paper to justify the formulations that it presumes to impose broadly on the landscape.

Response: I agree with the criticism that the processes I invoked in my previous model were not necessarily the best or only controls on P_0 values, so I have thought hard about this issue, tested the topographically induced stress fracture opening process, and found it to be a better explanation of trends in the data. Once this control is accounted for, a climatic control on P_0 values becomes apparent at the highest elevations of the SGM.

I did not state or imply that it was beyond the scope of the paper to justify the formulations I am invoking. Rather, the point I made was that there is value in documenting statistically significant correlations between P_0 values and controls that are based on reasonable process-based models given that the literature has only identified one control (average slope) on P_0 in this dataset and no process-based understanding for even that trend. My proposed revision is an advance because it identifies topographically induced stress fracture opening as the process most likely responsible for the average slope control on P_0 values in the SGM.

My reanalysis of the data shows that, contrary to my *ESurfD* paper, the null hypothesis that P_0 values are independent of slope aspect cannot be rejected. Ten of the sample locations are ridgetops where the local slope is zero and slope aspect should be undefined. However, my initial extraction routine did not account for this fact. Instead, the routine returned values that in several cases indicated that the slopes faced nearly directly south or north (which was correct given the location data, which in some cases is 10-30 m from the ridge due to roundoff error in the sample location). When the data are reanalyzed to include only areas that are not ridgetops (47 of 57 samples), P_0 values are slightly higher, on average, on south-facing slopes, but the null hypothesis that P_0 values are independent of slope aspect cannot be rejected.

Note: The AE has instructed me to respond to reviewer 2 prior to drafting a revision. However, I don't think it is possible to fully respond to their concerns without drafting a revision, since the requested changes were so extensive and fundamental.

Comment: (4) The statistical analyses are nonstandard. My discomfort with them is very high. My discomfort started with the first indication – I think on page 5 – that the author thinks of statistical significance as the logical and quantitative complement to a calculated p value. This is not the case, of course. Rather "significance" is commonly reserved referring to the threshold false positive rate that is allowed in a statistical hypothesis test. So the idea that the author thinks that a calculated p = 0.001 corresponds to a "statistical significance" of 99.9% set me on edge. This misappropriation of terminology was repeated many times throughout the text. But that was just the start. The author also evidently thinks it is ok to calculate a y-intercept for each measured value in a dataset using an overall regression slope that was calculated from the entire data set – and which also yields an overall regression intercept. To be honest,

this seems akin to data fabrication to me, but I can settle on the gentler view of Heimsath and Whipple that it is really just of a crude way to estimate the uncertainty in the intercept. Next, the author follows a rather stilted approach to quantifying the relationship between P0 and D and A. I personally think it should be P versus D, A and h, thus recognizing h as a factor regulating P and avoiding the problem of getting just two P0 values from 57 values of P. In addition, I think the author missed an opportunity to perform a very standard multiple regression analysis on log-transformed variables and instead opted or a multi stage approach that undoubtedly underestimates errors and fails to produce vital outputs like leverage plots and partial regression coefficients which would help the audience gauge the relative importance of the different factors in the regression. In addition, there is no attempt to propagate uncertainties through any of this. This is a major oversight that needs to be fixed. Last and not least, the author also thinks it is ok to use the significance of R^2 for the relationship between predicted and observed values to judge the performance of his model. In the hydrology community that idea has been rejected for nearly half a century. I am very concerned about the strength of the analyses for these reasons.

Response: The reviewer is correct that it is more accurate to define a threshold false positive rate (typically 0.05) and then compare the p value to this threshold to determine whether the null hypothesis is accepted or rejected. I have rephrased my discussion of statistical significance accordingly in the revision. An example from the proposed revision is as follows: "Assuming a significance level of 0.05, the null hypothesis that the cluster of blue points has a mean that is indistinguishable from that of the remaining points with $S_{av} > 30°$ can be rejected based on the standard t test with unequal variances (t = 0.021)."

The reviewer's claim that computing residuals is "akin to data fabrication" is troubling. Given that even a whiff of fabrication can ruin a scientist's career, this is language that, if taken out of context, could be very damaging. I am stunned that anyone would invoke this charge on a fellow scientist in an open review without any evidence of actual fabrication.

Comment: 2.10. I see that Heimsath and Whipple have provided a review of the manuscript and will defer to them as experts on evaluating this paragraph as a motivating theme for the paper. They did not call attention to any problems here. However, as I read line 21 on this page, I guess I have to say that this was not the take home message I got from Heimsath et al., 2012. Higher frequency of disturbance?

Response: "… a greater frequency of disturbance for a given soil thickness" is a defining phrase in the concluding paragraph of Heimsath et al. (2012). As such, I think it is appropriate to include it in a review of the relevant literature. However, I have rephrased this text as follows: "Heimsath et al. (2012) concluded that high erosion rates, triggered by high tectonic uplift rates and the resulting steep topography, cause potential soil production rates to increase above any limit set by climate and bedrock characteristics. Their results challenge the traditional view that $P\_0$ values are controlled solely by climate and rock characteristics."

Comment: 3.6-3.10. This study seems to hang a lot of its motivation on the unsupported idea that aspect promotes differences in vegetation which in turn promote differences in fire that promote differences in weathering in the SGM area. See general comment above.

Response: Text removed.

Comment: 4.11. I think I understand what the author is trying to do here (correct the measured P0 for the hump in the SPF), but on reading this, I am confused. You used 1.78P for P0? Not 1.78P0? The way I want to read it is the author is correcting the "measured" P0 – which is inferred from the exponential function to the data – by some correction factor. But again, I am confused by this statement.

Response: A humped production function means that the maximum or potential soil production rate is higher than the P value measured on bare ground. Hence P_0, defined as the maximum soil production rate at a point, has to be higher than P for these four cases. As explained in the paper, the data suggest that the factor increase is 1.78. Hence P_0 = 1.78P. I don't see how P0=1.78P0 could possibly be an alternative way of estimating P0, as the reviewer suggests.

Comment: 4.12. "This modification of equation (1) affects 4 of the 57 data points." This would only be comforting if there was actually a very strong trend across all the data. Instead, it seems that the data form really loose clouds of correlations that are hinged entirely on a few points. So the fact that this affects 4 of the points is actually troubling – not comforting – to me.

Response: Text removed.

Comment: 5.3. This equation does not include the fault specific constant of Savage and Brodsky. So I think this assumes that the constant is the same across the study area. Is this justified? Also, to make D dimensionless wouldn't delta x need to be raised to the 0.8 power too?

Response: Savage and Brodsky found no relationship between the fault specific constant and fault displacement (which correlates strongly with fault length). That is, there was variation from fault to fault in terms of their effect on fracture density in nearby rocks, but no systematic variations that one could use in a predictive equation. They stated "When we plot the entire data set shown in Figure 5, there is no clear relationship between c (the fault-specific constant) and displacement"). Savage and Brodsky did propose a weak pattern for faults in siliciclastic rocks, which is clearly not relevant for SGM.

Regarding the units of D, I have thought about this more and run some tests to determine how D should be defined so that the results are most nearly independent of grid resolution. I have found that D should not be dimensionless but should have units of length since it represents the total length of fault segments in a region (albeit weighted by an inverse power-law function of distance). The proposed revision addresses this point as follows: "I define the bedrock damage index D (Fig. 5A) as the sum of the inverse distances, raised to an exponent 0.8, from the point where the D value is being computed to every pixel in the study area were a fault is located:

$$D = \sum_{\mathbf{x'}} \Delta x \left( \Delta x / |\mathbf{x} - \mathbf{x'}| \right)^{0.8} \tag{6}$$

where $\Delta x$ is the pixel width, $\mathbf{x}$ is the map location where bedrock damage is being computed, and $\mathbf{x'}$ is the location of each mapped pixel in SGM where a fault exists. $D$ has units of length since it is the sum of all fault lengths in the vicinity of a point, weighted by a power-law function of inverse distance."

Comment: 5.10. ***This is very important.*** The line plotted in Fig. 3A is a log-log regression that ignores the cluster of five data points circled in the figure. There is NO justification for ignoring these points!!! He says in line 5.20 that they occur in an area of unusually dense landslides. I do not see this in figure 1!!! Even if I did, it would not justify excluding them from the analysis. Heimsath and Whipple seem to agree. I think it is complete nonsense. Makes the line look steeper than is should be. Sweeping these points under the rug does not make them go away. Including them in the regression would undermines his story that D plays a "subequal" role with tectonics. It not only looks suspicious. It is suspicious. Author needs to HONOR the data in this study and in his other work and not try to sweep data points away like this.

Response: The line plotted in Figure 3A was the linear trend predicted by a simultaneous multivariate regression of P_0 to D and A that included all of the data points. My discussion of these 5 points was limited to a thought exercise in which I reported p values of the relationship between P_0 and D with and without

these points included. That thought exercise did not extend to the multivariate regression or any other part of the paper. I made it clear in my Sept 25 response to Heimsath and Whipple that any mention of the cluster of five points would be removed from the proposed revision.

Comment: 6.10. I do not understand why the correlation would shut off on north-facing slopes. Is there a mechanistic/theoretical basis for this? If not than the relationship is purely empirical.

Response: My reanalysis of the data shows that, contrary to my *ESurfD* paper, the null hypothesis that $P_0$ values are independent of slope aspect cannot be rejected. In my earlier analysis I extracted slope aspect using the location data provided by Heimsath et al. (2012), which identify sample locations to an accuracy of approximately 10-30 m. Ten of the sample locations were ridgetops where the local slope is zero and slope aspect should be undefined. However, my initial extraction routine did not account for the local slope, hence my routine returned a slope aspect close to directly south- or directly north-facing for some of these ridgetop samples. When the data are reanalyzed to include only areas that are not ridgetops, $P_0$ values are slightly higher, on average, on south-facing slopes, but the null hypothesis that $P_0$ values are independent of slope aspect cannot be rejected. All of the discussion of aspect has therefore been removed from the proposed revision.

Comment: 6.20. Some more non-standard statistical machinations. The author does a regression that suggests that the power law exponents of A and D are 1.1 +/- some error. Then he reanalyzes things assuming that they are 1 to determine the value of c – the constant in front of A and D in Eq. 2. I am at a loss here. I know the author to be very bright and competent quantitatively. Yet here he invoking using some unnecessary, non-standard, and potentially misleading steps to avoid what would be a fairly straightforward multiple regression analysis of all of the parameters (slopes and intercepts) implied by a power law formulation of Equation 2. Doing this in a more standard way would yield some very useful metrics like partial correlation coefficients and leverage plots. Perhaps his approach seemed easier to explain at the time he wrote it. But I would argue that the community deserves and expects more.

Response: When a power-law relationship has an exponent of 1.1 +/- 0.3, I think it is appropriate to assume a linear relationship for simplicity (since 1.1 and 1.0 are indistinguishable, within uncertainty). However, this text has been removed as it is no longer included in the revised analysis.

Comment: 7.1-7.2. This is actually not a very good correlation for predicted versus observed – especially since it is strangely for a log-log plot. To understand this, look at the plot. There is almost an order of magnitude of variation in predicted P0 at any given value of P measured. To evaluate this model, rather than see an R^2 for a log-log observed versus predicted plot, I think we need to see something like a Nash-Sutcliffe statistic, which would tell us how good the model is compared to simply assuming that we could use the average P measured to estimate P everywhere.

Response: As I have already noted, the Nash-Sutcliffe statistic does not apply to regression models, and I have provided the closest analogous statistic ($R^2$). I don't know what the reviewer means by a "strange" log-log plot. When plotting data that have a large positive skew, it is common to plot log-log simply so that the points that would otherwise cluster in the lower left corner of a linear-linear scale can be resolved in the graph.

Comment: 7.5 What are the assumptions inherent in simplifying the equations in this way? Simply citing off to previous work here is not sufficient. What are the assumptions inherent in doing this? For equation 6 you assume slopes are planar, right? Is that reasonable here? What are the limitations of removing the higher order terms of Roering et al.?

Response: I have clarified the assumption as follows: "Equation (8) assumes that the mean slope gradient at the base of hillslopes (where sediment leaves the slope) can be approximated by the average slope, $S_{av}$." Roering et al. proposed that sediment flux is proportional to slope with a one minus slope squared term in the denominator. If the divergence of the flux is computed, the result is a complex expression with higher-order terms, but I am using the same equation Roering et al. proposed for flux. I am happy to clarify further but I would need more information from the reviewer to do so.

Comment: 7.19. Why 0.03? Just because this is the minimum finite thickness measured? But the whole point is they have no thickness!!! The mathematical inconvenience of having a value of 0 on what you want to plot on a log scale does not justify making up a value that \*\*\*drives\*\*\* a regression that you then plot through the data. Importantly it is very true that these points have a lot of leverage on the regression. Since calculating understanding the relationship between h and S is vital to calculating E from topography, this ends up being key to the paper. And I really do not think it is well justified.

Response: These locations have no thickness *today* but must have episodically had soil in the past or else they would never erode (absent landsliding in bedrock or intact regolith, which can certainly occur but are not widespread in granitic rocks). It is common practice to add a small constant (comparable to the uncertainty of the data) prior to performing a linear regression of log-transformed data. I don't think the alternative (leaving out these values entirely from the analysis, thereby biasing the results to those with finite soil thickness) is a better choice. If the reviewer would please provide a suggestion as to how these data could be included in a way that would satisfy him, I would be willing to try whatever alternative he proposes.

Comment: 9.8. If this is the key result, then you need to demonstrate it using more conventional statistical approaches. A multiple linear regression of the log of P versus log D and h and log A would be a good place to start. This would avoid the strange – and thus hard-to-justify – correction of P to P0 that you have employed here. It would also avoid the strange practice of finding a 1.1 +/- error power slope and then redoing the regression assuming the slopes are 1 to find the best fit intercept term. This whole analysis seemed like a contorted and potentially error-prone way of doing what could have been a textbook application of multiple linear regression analysis on transformed variables.

Response: In the proposed revision I have used conventional statistical approaches throughout.

Comment: 10.12. This is misleading at best. I see a factor of 2 to 3 in either direction, so a factor of 4 to 6 overall. For example, in Fig 3D, at a value of P0 observed of _150 m/My I see a range of predicted values running from 85 to 450 m/My. That's a factor of nearly 6 range in predictions for a single value of P observed. That is NOT a good prediction in my book and my assertion is asserted by the very low N-S statistic for this modeling exercise.

Response: The sentence is correct as stated. When saying that a prediction is correct to within a factor of 2 from the observed value for 72% of the data points, that includes differences both above and below the prediction (resulting in a factor of 4 difference between the max and min predictions at a given observed value of P_0). However, I have removed the sentence because it is not central to the argument.

Proposed revision to Section 2.1:

[revised manuscript text omitted]

---

## Author Comment (AC4) · 16 Nov 2016

**Reviewer 3 (Simon Mudd):**

Comment: In this paper Jon Pelletier has used the dataset from Heimsath et al 2012 to explore controls on soil production. In the original paper, Heimsath and colleagues argued that rapid erosion rates could affect the P0 term in the soil production function. The obvious follow on question is: by what mechanisms does erosion rate modulate P0? As stated by Heimsath and Whipple's comment (doi:10.5194/esurf-2016-37), the original 2012 paper did not mechanistically explain observed trends. So, does Pelletier's paper give insight into the mechanisms? Firstly we can look at the damage indicator. I found this interesting since many authors have speculated on the role of fracturing in controlling weathering rates, and the implementation of equation (3) is a novel attempt to translate mapped faults into a metric for fracture density using results from detailed field studies. To compare this metric with soil production, Pelletier calculates P0 from every data point by regressing the soil production function, using a slope of h0 previously regressed in the Heimsath et al paper, to its $h = 0$ intercept. To do this, one must assume that the individual P0 results are meaningful and not simply the results of scatter in the data due to local heterogeneities in shielding and erosion history; Heimsath and Whipple feel this unwise, a point I will revisit later in this comment. However once Pelletier follows this thread he finds a weak correlation between the D metric and P0;regressed data (I'm not sure if I'd be so bold as to call it measured). One can explain

Response: Simon's point seems to end abruptly. However, I gather from his comments that he is somewhat convinced that a statistically significant relationship exists between $P\_0$ values and D values. In the proposed revision, this point the model has been modified to be based on the topographically induced stress hypothesis.

Comment: What about aspect? There are a few rather high P0;regressed values for south facing slopes. Of the 11 points with P0;regressed values greater than 300 m/Myr, 8 of them are on south facing slopes. But there are also a large number of points on south facing slopes that don't have P0 values that are higher than the mean P0 value. The model combining topographic gradient and aspect again shows a correlation between it and the P0 values, this time explaining

Response: The revised analysis has demonstrated that aspect is not statistically significant. In the *ESurfD* paper I extracted slope aspect using the location data provided by Heimsath et al. (2012). Ten of the sample locations were ridgetops where the local slope is zero and slope aspect is undefined. When the data were reanalyzed to include only areas that were not ridgetops, $P\_0$ values are slightly higher, on average, on south-facing slopes, but the null hypothesis that $P\_0$ values are independent of slope aspect cannot be ruled out. The revised analysis is focused on average slope and climatic controls on $P\_0$ values. I apologize for making more work for the reviewers with this major change to the manuscript, but I believe that the revised paper makes a convincing case.

Comment: I am somewhat confused by section 2.2. It seems strange to generate a map of steepness and from that calculate the spatial distribution of h and E. Global topographic maps are readily available so why calculate S from equation (8), which contains many assumptions, rather than just use topographic data? It also seems quite odd to use equation (7) since theory suggests that for a given erosion rate and P0, hillslope-scale gradient will vary as a function of hillslope length. More explanation of these choices is warranted.

Response: The purpose of section 2.2 is to model demonstrate that a model based on a combination of soil production functions and nonlinear slope-dependent sediment flux can reproduce the observed spatial variations and interrelationships among geomorphic and pedogenic variables in the SGM. I think the full power of the model is not clear until it can be shown to reproduce the full suite of variables across the range. This requires that slope be modeled first, then compared to an independent dataset. Equations (6)-(8) includes all the variables mentioned, (hillslope length, hillslope-scale gradient, erosion rate, and $P\_0$), so I

think the model is consistent with the theory Simon is referring to. In the proposed revision section 2.2 is motivated using the following text: "In this section I invoke a balance between soil production and transport at the hillslope scale in order to illustrate the interrelationships among potential soil production rates, erosion rates, soil thicknesses, and average slopes across the SGM. The conceptual model explored in this section is based on the hypothesis that the average slope depends on the difference between uplift and erosion rates. Uplift rates (assumed to be equal to exhumation rates) are lower in the western portion of the SGM and higher in the eastern portion (Spotila et al., 2002, Fig. 7b). As average slope increases in areas with higher uplift rates, erosion rates increase and soils become thinner. Both of these responses represent negative feedback mechanisms that tend to decrease the differences that would otherwise exist between uplift and erosion rates and between erosion rates and soil production rates. If the uplift rate exceeds the potential soil production rate, soil thickness becomes zero and soil production and erosion rates can no longer increase with increasing slope (in the absence of widespread landsliding in bedrock or intact regolith). In such cases, topography with cliffs or steps may form (Wahrhaftig, 1965; Pelletier and Rasmussen, 2009; Jessup et al., 2010). However, if the potential soil production rate increases with average slope via the topographically induced stress fracture opening process, the transition to bare landscapes can be delayed or prevented, thus representing an additional negative feedback or adjustment mechanism (Heimsath et al., 2012). At the highest elevations of the range, however, soil production is slower, most likely due to temperature limitations on vegetation growth since the slopes there are among the steepest in the range. The interrelationship between these variables can be quantified without explicit knowledge of the uplift rate, since the relationship between soil thickness and average slope implicitly accounts for uplift rate (i.e., a smaller difference between uplift and erosion rates is characterized by a thinner soil). This conceptual model predicts positive correlations among potential soil production rates, erosion rates, and topographic steepness, and negative correlations of all of these variables with soil thickness."

Comment: It is worth commenting on the use of scatter in soil production data to regress P0 values for individual samples. Because these numbers were collected at specific points in the landscape (i.e., they are not basin-averaged data), one must consider if the local sources of scatter. Suppose one measured 10 P values in close proximity (e.g., in a 15 m radius): how variable would those P values be? We don't actually know how representative the P values are on a local scale, but we know soil thickness can have quite a bit of local variability, chemical weathering can have substantial local variability, and you can have substantial local variability in the production of 10Be (from where snow falls, any transience in erosion history, etc.). So I do not think Heimsath and Whipple's concern about interpreting the P0 values is unwarranted: I share this concern. So, in summary, I am worried that the potential uncertainties in P values makes it difficult to come to strong conclusions about influences of other factors on P0, that even if you believe the P0 values are representative the correlation with D is rather weak, and that I do not feel the effects of aspect have been sufficiently separated from gradient effects.

Response: I agree with Simon that some variability in P or $P_0$ values is due to methodological uncertainty such as snow shielding, etc. I also agree that, if that variability were dominant it would be dangerous to attempt to interpret $P_0$ values (because, for example, snow shielding varies with aspect and hence a methodological bias could be misinterpreted as an aspect control on soil production processes). However, I don't think that errors associated with the methodology are anywhere close to the order-of-magnitude variations in $P_0$ values observed in the data.

Please see my responses to reviewers 1 and 2 for additional information relevant to the proposed revision.

---

## Author Comment (AC5) · 16 Nov 2016

[revised manuscript text omitted]
_{av}$ (°) | $P$ (m/Myr) | $P$ error (m/Myr) | $h$ (cm) | $P_{0,resid}$ (m/Myr) | $P_{0,resid}$ error | $D$ (km) | $P_{0,pred}$ (m/Myr) |
|---|---|---|---|---|---|---|---|---|---|---|---|
| SG-1 | 34.2090 | -117.7714 | 1072 | 30 | 300 | 38 | 0 | 300 | 38 | 28.4 | 174 |
| SG-2 | 34.2118 | -117.7685 | 1072 | 30 | 240 | 19 | 0 | 240 | 19 | 29.3 | 174 |
| SG-6 | 34.1868 | -117.7632 | 947 | 35 | 460 | 148 | 0 | 818 | 263 | 28.0 | 522 |
| SG-7 | 34.1856 | -117.7662 | 950 | 35 | 373 | 34 | 5 | 427 | 39 | 26.4 | 522 |
| SG-10 | 34.207 | -117.7621 | 855 | 18 | 68 | 5 | 38 | 221 | 16 | 30.2 | 154 |
| SG-101 | 34.2852 | -118.1519 | 1673 | 35 | 156 | 32 | 0 | 278 | 57 | 21.7 | 522 |
| SG-102 | 34.2852 | -118.1519 | 1673 | 35 | 251 | 51 | 30 | 564 | 115 | 21.7 | 522 |
| SG-103 | 34.3717 | -118.0710 | 2015 | 26 | 78 | 17 | 23 | 159 | 35 | 9.5 | 169 |
| SG-104 | 34.3717 | -118.0710 | 2015 | 26 | 21 | 9 | 30 | 53 | 23 | 9.5 | 169 |
| SG-105 | 34.3707 | -118.0701 | 2005 | 26 | 48 | 12 | 43 | 182 | 46 | 9.2 | 169 |
| SG-106 | 34.3706 | -118.0692 | 1990 | 26 | 51 | 13 | 36 | 156 | 40 | 9.1 | 169 |
| SG-107 | 34.3569 | -118.0631 | 1804 | 26 | 164 | 29 | 15 | 261 | 46 | 7.9 | 169 |
| SG-108 | 34.3543 | -118.0580 | 1625 | 26 | 113 | 23 | 15 | 180 | 37 | 8.0 | 169 |
| SG-110 | 34.2931 | -118.0199 | 1725 | 26 | 106 | 20 | 10 | 145 | 27 | 21.6 | 169 |
| SG-111 | 34.2930 | -118.0202 | 1721 | 26 | 63 | 15 | 44 | 246 | 59 | 21.9 | 169 |
| SG-112 | 34.2932 | -118.0211 | 1729 | 26 | 44 | 9 | 27 | 102 | 21 | 21.4 | 169 |
| SG-113 | 34.2908 | -118.0218 | 1650 | 26 | 44 | 10 | 34 | 126 | 29 | 22.4 | 169 |
| SG-115 | 34.2832 | -118.0263 | 1390 | 15 | 138 | 33 | 10 | 188 | 45 | 23.4 | 145 |
| SG-153 | 34.3273 | -117.7998 | 2194 | 29 | 90 | 18 | 20 | 167 | 33 | 28.3 | 173 |
| SG-154 | 34.3460 | -118.0060 | 1800 | 12 | 43 | 10 | 44 | 168 | 39 | 18.4 | 133 |
| SG-155 | 34.3469 | -118.0059 | 1790 | 12 | 50 | 13 | 54 | 267 | 69 | 18.4 | 133 |
| SG-156 | 34.3476 | -118.0061 | 1780 | 12 | 12 | 3 | 45 | 48 | 12 | 18.4 | 133 |
| SG-200 | 34.3586 | -117.9922 | 1710 | 13 | 68 | 13 | 16 | 112 | 21 | 18.0 | 137 |
| SG-201 | 34.3589 | -117.9920 | 1710 | 13 | 69 | 13 | 10 | 94 | 18 | 18.0 | 137 |
| SG-202 | 34.3590 | -117.9922 | 1706 | 13 | 72 | 13 | 15 | 115 | 21 | 18.0 | 137 |
| SG-203 | 34.3592 | -117.9923 | 1702 | 13 | 109 | 20 | 10 | 149 | 27 | 17.9 | 137 |
| SG-07-009 | 34.3215 | -118.0866 | 1132 | 15 | 69 | 12 | 28 | 164 | 29 | 5.4 | 145 |
| SG-07-011 | 34.3320 | -117.9483 | 2137 | 20 | 93 | 18 | 12 | 135 | 26 | 30.7 | 159 |
| SG-07-012 | 34.3318 | -117.9481 | 2128 | 20 | 84 | 17 | 20 | 156 | 32 | 31.0 | 159 |
| SG-07-013 | 34.3318 | -117.9481 | 2120 | 24 | 121 | 24 | 20 | 225 | 45 | 31.0 | 166 |
| SG-07-014 | 34.3318 | -117.9481 | 2115 | 24 | 156 | 32 | 24 | 328 | 67 | 31.0 | 166 |
| SG-07-015 | 34.3259 | -117.9517 | 1897 | 24 | 139 | 26 | 0 | 139 | 26 | 35.8 | 166 |
| SG-07-016 | 34.3276 | -117.9507 | 1965 | 24 | 130 | 25 | 0 | 130 | 25 | 33.9 | 166 |
| SG-07-017 | 34.3304 | -117.9498 | 2068 | 24 | 147 | 28 | 0 | 147 | 28 | 31.8 | 166 |
| SG-07-019 | 34.3484 | -118.0045 | 1773 | 12 | 10 | 2 | 49 | 46 | 9 | 20.8 | 133 |
| SG-07-020 | 34.3482 | -118.0035 | 1785 | 12 | 46 | 12 | 50 | 217 | 57 | 22.4 | 133 |
| SG-07-021 | 34.3471 | -118.0030 | 1802 | 12 | 71 | 16 | 35 | 210 | 47 | 20.9 | 133 |
| SG-07-023 | 34.3627 | -117.9108 | 1958 | 31 | 427 | 145 | 25 | 839 | 285 | 31.5 | 308 |
| SG-07-024 | 34.3615 | -117.9107 | 1912 | 31 | 315 | 61 | 5 | 360 | 70 | 31.0 | 308 |
| SG-07-025 | 34.3614 | -117.9110 | 1889 | 31 | 266 | 54 | 7 | 321 | 65 | 31.0 | 308 |
| SG-07-031 | 34.3348 | -117.9695 | 1973 | 21 | 210 | 41 | 13 | 314 | 61 | 27.0 | 161 |
| SG-07-032 | 34.3348 | -117.9695 | 1973 | 21 | 210 | 38 | 3 | 230 | 42 | 27.0 | 161 |
| SG-07-033 | 34.3348 | -117.9695 | 1973 | 21 | 92 | 18 | 20 | 171 | 33 | 27.0 | 161 |
| SG-07-034 | 34.3348 | -117.9695 | 1973 | 21 | 132 | 25 | 5 | 154 | 29 | 27.0 | 161 |
| SG-07-035 | 34.3264 | -117.9690 | 1703 | 21 | 146 | 28 | 3 | 160 | 31 | 37.8 | 161 |
| SG-07-038 | 34.3307 | -117.9700 | 1847 | 21 | 178 | 33 | 3 | 195 | 36 | 32.5 | 161 |
| SG-07-041 | 34.3307 | -117.9700 | 1847 | 21 | 83 | 15 | 12 | 120 | 22 | 32.5 | 161 |
| SG-07-042 | 34.3307 | -117.9700 | 1847 | 21 | 98 | 18 | 12 | 142 | 26 | 32.5 | 161 |
| SG-07-044 | 34.3524 | -117.8792 | 2077 | 23 | 143 | 26 | 0 | 143 | 26 | 31.3 | 165 |
| SG-07-045 | 34.3521 | -117.8791 | 2058 | 38 | 594 | 125 | 8 | 737 | 155 | 31.4 | 527 |
| SG-08-100 | 34.3639 | -117.8379 | 1934 | 31 | 79 | 16 | 25 | 155 | 31 | 38.0 | 308 |
| SG-08-101 | 34.3648 | -117.8383 | 1880 | 31 | 338 | 90 | 8 | 419 | 112 | 38.5 | 308 |
| SG-08-102 | 34.3648 | -117.8383 | 1880 | 31 | 329 | 69 | 10 | 430 | 90 | 38.5 | 308 |
| SG-08-105 | 34.3712 | -117.8581 | 2494 | 35 | 96 | 23 | 0 | 170 | 41 | 38.2 | 209 |
| SG-08-106 | 34.3714 | -117.8578 | 2491 | 35 | 118 | 25 | 0 | 210 | 45 | 38.2 | 209 |
| SG-08-108 | 34.3720 | -117.8571 | 2442 | 35 | 166 | 34 | 20 | 285 | 58 | 37.9 | 209 |
| SG-08-110 | 34.3723 | -117.8631 | 2398 | 32 | 163 | 31 | 5 | 187 | 35 | 37.3 | 206 |

---

## Author Comment (AC6) · 16 Nov 2016

In my proposed revision posted on Nov 16, I included an incorrect scaling factor when computing the bedrock damage index, $D$. The correct equation for $D$ (i.e., the one that yields a result that is independent of grid resolution) is

$$D = \sum_{\mathbf{x}'} \left( \Delta x / \left| \mathbf{x} - \mathbf{x}' \right| \right)^{0.8} \tag{6}$$

My proposed revision included an extra factor of $\Delta x$ and an incorrect sentence that should be removed ("$D$ has units of length since it is the sum of all fault lengths in the vicinity of a point, weighted by a power function of inverse distance."). In fact, $D$ should be dimensionless and the values and units reported in the *ESurfD* paper were correct. This error does not affect any of the conclusions of the revision (since the damage index was included only as a discussion point) nor does it affect the relative values of $D$ since the error was a constant factor. The correct figure 5 and supplementary table are below. I apologize for the error.

[Figure]

**Table 1. Data used in the paper. Sample ID, location, elevation ($z$), mean slope ($S_{av}$), $P$ values, and soil thickness ($h$) are from Heimsath et al. (2012).**

| Sample ID | Latitude (°) | Longitude (°) | $z$ (m) | $S_{av}$ (°) | $P$ (m/Myr) | $P$ error (m/Myr) | $h$ (cm) | $P_{0,resid}$ (m/Myr) | $P_{0,resid}$ error | $D$ (m/m) | $P_{0,pred}$ (m/Myr) |
|---|---|---|---|---|---|---|---|---|---|---|---|
| SG-1 | 34.2090 | -117.7714 | 1072 | 30 | 300 | 38 | 0 | 300 | 38 | 24.8 | 174 |
| SG-2 | 34.2118 | -117.7685 | 1072 | 30 | 240 | 19 | 0 | 240 | 19 | 25.6 | 174 |
| SG-6 | 34.1868 | -117.7632 | 947 | 35 | 460 | 148 | 0 | 818 | 263 | 24.5 | 522 |
| SG-7 | 34.1856 | -117.7662 | 950 | 35 | 373 | 34 | 5 | 427 | 39 | 23.1 | 522 |
| SG-10 | 34.207 | -117.7621 | 855 | 18 | 68 | 5 | 38 | 221 | 16 | 26.4 | 154 |
| SG-101 | 34.2852 | -118.1519 | 1673 | 35 | 156 | 32 | 0 | 278 | 57 | 19.0 | 522 |
| SG-102 | 34.2852 | -118.1519 | 1673 | 35 | 251 | 51 | 30 | 564 | 115 | 19.0 | 522 |
| SG-103 | 34.3717 | -118.0710 | 2015 | 26 | 78 | 17 | 23 | 159 | 35 | 8.3 | 169 |
| SG-104 | 34.3717 | -118.0710 | 2015 | 26 | 21 | 9 | 30 | 53 | 23 | 8.3 | 169 |
| SG-105 | 34.3707 | -118.0701 | 2005 | 26 | 48 | 12 | 43 | 182 | 46 | 8.0 | 169 |
| SG-106 | 34.3706 | -118.0692 | 1990 | 26 | 51 | 13 | 36 | 156 | 40 | 9.2 | 169 |
| SG-107 | 34.3569 | -118.0631 | 1804 | 26 | 164 | 29 | 15 | 261 | 46 | 9.2 | 169 |
| SG-108 | 34.3543 | -118.0580 | 1625 | 26 | 113 | 23 | 15 | 180 | 37 | 9.1 | 169 |
| SG-110 | 34.2931 | -118.0199 | 1725 | 26 | 106 | 20 | 10 | 145 | 27 | 18.9 | 169 |
| SG-111 | 34.2930 | -118.0202 | 1721 | 26 | 63 | 15 | 44 | 246 | 59 | 19.1 | 169 |
| SG-112 | 34.2932 | -118.0211 | 1729 | 26 | 44 | 9 | 27 | 102 | 21 | 18.7 | 169 |
| SG-113 | 34.2908 | -118.0218 | 1650 | 26 | 44 | 10 | 34 | 126 | 29 | 19.6 | 169 |
| SG-115 | 34.2832 | -118.0263 | 1390 | 15 | 138 | 33 | 10 | 188 | 45 | 20.5 | 145 |
| SG-153 | 34.3273 | -117.7998 | 2194 | 29 | 90 | 18 | 20 | 167 | 33 | 24.7 | 173 |
| SG-154 | 34.3460 | -118.0060 | 1800 | 12 | 43 | 10 | 44 | 168 | 39 | 16.1 | 133 |
| SG-155 | 34.3469 | -118.0059 | 1790 | 12 | 50 | 13 | 54 | 267 | 69 | 16.1 | 133 |
| SG-156 | 34.3476 | -118.0061 | 1780 | 12 | 12 | 3 | 45 | 48 | 12 | 16.1 | 133 |
| SG-200 | 34.3586 | -117.9922 | 1710 | 13 | 68 | 13 | 16 | 112 | 21 | 15.8 | 137 |
| SG-201 | 34.3589 | -117.9920 | 1710 | 13 | 69 | 13 | 10 | 94 | 18 | 15.7 | 137 |
| SG-202 | 34.3590 | -117.9922 | 1706 | 13 | 72 | 13 | 15 | 115 | 21 | 15.7 | 137 |
| SG-203 | 34.3592 | -117.9923 | 1702 | 13 | 109 | 20 | 10 | 149 | 27 | 15.6 | 137 |
| SG-07-009 | 34.3215 | -118.0866 | 1132 | 15 | 69 | 12 | 28 | 164 | 29 | 8.6 | 145 |
| SG-07-011 | 34.3320 | -117.9483 | 2137 | 20 | 93 | 18 | 12 | 135 | 26 | 26.9 | 159 |
| SG-07-012 | 34.3318 | -117.9481 | 2128 | 20 | 84 | 17 | 20 | 156 | 32 | 27.1 | 159 |
| SG-07-013 | 34.3318 | -117.9481 | 2120 | 24 | 121 | 24 | 20 | 225 | 45 | 27.1 | 166 |
| SG-07-014 | 34.3318 | -117.9481 | 2115 | 24 | 156 | 32 | 24 | 328 | 67 | 27.1 | 166 |
| SG-07-015 | 34.3259 | -117.9517 | 1897 | 24 | 139 | 26 | 0 | 139 | 26 | 26.5 | 166 |
| SG-07-016 | 34.3276 | -117.9507 | 1965 | 24 | 130 | 25 | 0 | 130 | 25 | 29.6 | 166 |
| SG-07-017 | 34.3304 | -117.9498 | 2068 | 24 | 147 | 28 | 0 | 147 | 28 | 27.8 | 166 |
| SG-07-019 | 34.3484 | -118.0045 | 1773 | 12 | 10 | 2 | 49 | 46 | 9 | 18.2 | 133 |
| SG-07-020 | 34.3482 | -118.0035 | 1785 | 12 | 46 | 12 | 50 | 217 | 57 | 19.6 | 133 |
| SG-07-021 | 34.3471 | -118.0030 | 1802 | 12 | 71 | 16 | 35 | 210 | 47 | 18.3 | 133 |
| SG-07-023 | 34.3627 | -117.9108 | 1958 | 31 | 427 | 145 | 25 | 839 | 285 | 27.5 | 308 |
| SG-07-024 | 34.3615 | -117.9107 | 1912 | 31 | 315 | 61 | 5 | 360 | 70 | 27.2 | 308 |
| SG-07-025 | 34.3614 | -117.9110 | 1889 | 31 | 266 | 54 | 7 | 321 | 65 | 27.1 | 308 |
| SG-07-031 | 34.3348 | -117.9695 | 1973 | 21 | 210 | 41 | 13 | 314 | 61 | 23.6 | 161 |
| SG-07-032 | 34.3348 | -117.9695 | 1973 | 21 | 210 | 38 | 3 | 230 | 42 | 23.6 | 161 |
| SG-07-033 | 34.3348 | -117.9695 | 1973 | 21 | 92 | 18 | 20 | 171 | 33 | 23.6 | 161 |
| SG-07-034 | 34.3348 | -117.9695 | 1973 | 21 | 132 | 25 | 5 | 154 | 29 | 23.6 | 161 |
| SG-07-035 | 34.3264 | -117.9690 | 1703 | 21 | 146 | 28 | 3 | 160 | 31 | 29.5 | 161 |
| SG-07-038 | 34.3307 | -117.9700 | 1847 | 21 | 178 | 33 | 3 | 195 | 36 | 28.4 | 161 |
| SG-07-041 | 34.3307 | -117.9700 | 1847 | 21 | 83 | 15 | 12 | 120 | 22 | 28.4 | 161 |
| SG-07-042 | 34.3307 | -117.9700 | 1847 | 21 | 98 | 18 | 12 | 142 | 26 | 28.4 | 161 |
| SG-07-044 | 34.3524 | -117.8792 | 2077 | 23 | 143 | 26 | 0 | 143 | 26 | 27.4 | 165 |
| SG-07-045 | 34.3521 | -117.8791 | 2058 | 38 | 594 | 125 | 8 | 737 | 155 | 27.5 | 527 |
| SG-08-100 | 34.3639 | -117.8379 | 1934 | 31 | 79 | 16 | 25 | 155 | 31 | 33.3 | 308 |
| SG-08-101 | 34.3648 | -117.8383 | 1880 | 31 | 338 | 90 | 8 | 419 | 112 | 33.6 | 308 |
| SG-08-102 | 34.3648 | -117.8383 | 1880 | 31 | 329 | 69 | 10 | 430 | 90 | 33.6 | 308 |
| SG-08-105 | 34.3712 | -117.8581 | 2494 | 35 | 96 | 23 | 0 | 170 | 41 | 33.4 | 209 |
| SG-08-106 | 34.3714 | -117.8578 | 2491 | 35 | 118 | 25 | 0 | 210 | 45 | 33.4 | 209 |
| SG-08-108 | 34.3720 | -117.8571 | 2442 | 35 | 166 | 34 | 20 | 285 | 58 | 33.2 | 209 |
| SG-08-110 | 34.3723 | -117.8631 | 2398 | 32 | 163 | 31 | 5 | 187 | 35 | 32.6 | 206 |

---

## Author Response (AR2)

I wish to thank the AE for his open-mindedness in allowing me to submit a major revision to the discussion paper. I apologize for the time it has taken me to make the modest changes he has asked for. I promise that any future revisions will be completed more promptly. I have a strong incentive to get this paper published, in part because I do not wish the flawed discussion paper to remain the paper of record of my work on this topic.

Q: "One of the main issues concerns the computation of what Pelletier calls residuals and is referred to as "computing multiple intercepts" from a single regression by several of the reviewers. Although I see what the author has been trying to do, I also appreciate the point made by the reviewers that the value of P0 obtained by regression depends of course on the assumed value for h0 used in equation (1). Pelletier should call his residuals P' or Pr, to avoid confusion and refrain from making the assertion that by analysing the residuals, he is providing further constraints on what controls P0, as defined by Heimsath."

A: Please note that the two values of $h_0$ I have adopted come from the Heimsath et al. analysis. I have not assumed any value of $h_0$. Rather, I have assumed that the value of $h_0$ is sufficiently uniform within the two areas of the study site identified by Heimsath et al. (2012) (i.e., those with slopes above and below 30 degrees) that we can be confident in the trends of $P_{0,resid}$ (now called Pr) that I have identified. I understand perfectly well the issue that the reviewers have raised. The reviewers are concerned that a significant amount of the variation that I am attributing to Pr values is instead due to spatial variations in $h_0$. However, in order for the relationship between Pr and slope to be significantly affected, $h_0$ would have to have a systematic dependence on slope. As I have already noted in my previous rebuttal, Heimsath et al. (2012) clearly demonstrated that this was not the case. These authors considered two end member slope regimes and found that the average $h_0$ values for these two regions differed by only 0.05 m. As I noted previously, this difference in $h_0$ values corresponds to a difference in Pr values that is more than 100 times smaller than the actual variation in Pr values for a soil 30 cm in thickness. More broadly, $h_0$ values have now been estimated for many sites in widely different climates and have found to vary at most by approximately a factor of 2, compared to variability in $P_0$ values of approximately 3 orders of magnitude. In the revised paper I have added the following paragraph on this point: "Estimating Pr values using the residuals of the regressions of Heimsath et al. (2012) assumes that h0 has sufficiently limited variation within the two subsets of the study site considered by Heimsath et al. (2012) (i.e., those with Sav values above and below 30°) that any such variation would not affect the conclusions of the paper. For example, in order for the relationship between Pr and Sav (i.e., Figs. 2A-2C) to be significantly affected by variations in h0, h0 would have to have a systematic dependence on Sav. For example, if systematically lower values of h0 occur at steeper slopes and this effect is not accounted for, the result could be a biasing of Pr values downward in such regions. Heimsath et al. (2012) clearly demonstrated that no such systematic dependence exists. These authors considered two end member slope regimes and found that the average h0 values for these two regions differed by only 0.05 m (0.32 m vs. 0.37 m). At a soil thickness of 0.3 m, this difference corresponds to Pr differences of approximately 10% (i.e., exp(-0.30/0.32) vs. exp(-0.30/0.37)). This difference is more than 100 times smaller than the variation in Pr values. The difference becomes even smaller for soils thinner than 0.3 m."

The stated basis for Heimsath and Whipple's disagreement with me was, in some cases, simply that they had looked for a correlation (e.g., with climate) and found none, so therefore none must exist. I respectively ask that they consider my work on its merits. I propose that I have made a useful advance in this paper. I have provided a process-based explanation (i.e., topographic-stress-induced opening of fractures) for the slope dependence on Pr in the SGM. My analysis is an alternative to the theory proposed by Heimsath et al. (2012) that erosion rates control potential soil production rates both in the SGM and globally (the latter based on correlations between erosion rates and potential soil production rates globally). I believe the

evidence demonstrates that slope, not erosion rate, controls potential soil production rates, and only in regions of significant compressive stress. I appreciate that erosion rate and slope are closely correlated in the SGM, but such a correlation does not hold everywhere. My alternative is a very different view from Heimsath et al. (2012) but is supported using the same data they presented. I believe that the availability of two clear alternative models will spur new investigations on this problem.

I am somewhat confused by the recommendation that I change the variable $P_0$ since I already changed it to $P_{0,resid}$ in my previous submission to make it clear that my analysis uses residuals to estimate the soil maximum or potential sol production rate. I take the AE's recommendation to mean that I cannot use $P_0$ in any form whatsoever (even modified to indicate the use of residuals). In response, I have changed the symbol from $P_0$ to $Pr$. I am concerned that this change will be confusing to readers since it is well established in the literature that $P_0$ refers to the soil production rate as soil thickness approaches zero, which is equivalent to the maximum soil production rate in the case of an exponential soil production function. I am not aware of any publication, including any by Heimsath et al., that defines $P_0$ as the y intercept of a fit to an exponential soil production function. Every time that $P_0$ (or its precursors $epsilon_0$ (Heimsath et al. ) or $W_0$ (Furbish and Fagherazzi)) have been defined, it is as a property of nature (i.e., the soil production in the limit of zero soil thickness), not as a mathematical construct (the y-intercept of a regression). I believe that variables should always be defined as properties of nature rather than as the results of a particular method of data analysis. An analogy can be made here with the exponents m and n of the stream power law. These exponents are estimated in multiple ways, just as $P_0$ can be estimated in multiple ways. If we used a different symbol to represent the same property of nature each time a different method of estimation was used, the result would be chaos. In numerical models the values of m, n, or $P_0$ are often prescribed rather than estimated using data. In all models that I am aware of that include soil production, $P_0$ is used to represent the potential soil production rate regardless of how that variable is constrained (i.e., simply prescribed or fit to data). Such a use will, of course, no longer be allowed if this brand new definition of Heimsath is adopted as the only possible definition of $P_0$.

Q: "Pelletier argues now that the main factor controlling the residuals is tectonic stress. He shows that a simple relation obtained by Savage and Swolfs to predict stress can be used to predict the residuals. I note that in his revised version Pelletier makes references to Figure 2 whereas he means 3, I think. Pelletier also claims that there is an additional climatic control which he assesses by comparing the already corrected residuals with those further corrected by a climate correction factor. I am still somewhat confused on how that coefficient is determined."

A: I apologize for reversing the order of Figures 2 and 3 in my proposed revision. This has been corrected. I would be happy to clarify how the climate factor has been determined, but I would need more information from the AE on what sentences are confusing.

Q: "In the first version of the paper submitted to EsurfD, Pelletier argued for a control by fracture density which he estimated by introducing a damage index. He has revised not only the definition of the damage index, but also lessened his conclusion concerning the importance of damage/fracture density, arguing now for a control by topographic stresses alone. In this way, he has responded constructively to many of the reviewers criticisms."

A: Thank you. Please note that I have not redefined the damage index. The damage index in this revision is the same as the discussion paper. Please see AC6 for more on this issue.

Q: "Another major point of disagreement with the reviewer(s) is the measure of fit to data used by the authors (regression coefficient), whereas one of the reviewers argues for another, more appropriate measure, based on so-called Nash-Sutcliffe statistics. Pelletier argues that the measure proposed by the reviewer is inappropriate for assessing regressions. I am not an expert on this, but interestingly, the two methods yields the same fitness measure and Pelletier argues that it is appropriate."

A: There is not much more I can say on this point. As I argued in my earlier rebuttal, the N-S statistic does not apply to regression models (by definition).

Q: "All reviewers wonder why Pelletier has not performed a more classical multi variate analysis of the residuals, rather than privileging a step-by-step reduction of the residuals by incrementally incorporating potential processes through a simple mathematical description of the process. The main reason, I believe, of Pelletier's choice is that the relationships between variables (such as slope, climate, etc.) and residuals may not be linear at all and/or depend on thresholds which would be difficultly extracted from a simple cluster or multi variate analysis. I don't think, however, that this is clearly stated in the revised manuscript and it should (if my interpretation is correct) or an alternate explanation should be given."

A: I agree entirely with the AE that "the relationships between variables (such as slope, climate, etc.) and residuals may not be linear at all and/or depend on thresholds which would be difficultly extracted from a simple cluster or multi variate analysis." It was not a choice to forgo the multivariate linear fit to the data (with or without log transformation) proposed by reviewer 2. This approach would have been plainly inconsistent with the complex nonlinear relationships in the data, which my analysis and Heimsath et al.'s (2012) analysis clearly demonstrate are present. I have added the following paragraph to the discussion on this point: "This paper adopts a stepwise regression and cluster analysis approach that builds upon the regression analysis that Heimsath et al. (2012) used to characterize the dependence of soil production rate on soil thickness. Stepwise regression is a standard approach in statistics in which the residuals of a statistical regression are computed and additional controls tested for. I did not apply simultaneous multivariate linear regression (with or without log transformation) because such an approach would have been inconsistent with the complex nonlinear relationships in the data documented by Heimsath et al. (2012) and the analyses presented here."

I would like to make clear that I am not opposed to trying any additional analyses that the reviewers think are important. It is just not clear to me that an alternative approach has been suggested that is consistent with the complex nonlinear trends in the data nor is it clear to me why the approach I have taken has been judged to be inadequate.

Q: "There a few (unfortunate) comments by one of the reviewers that almost suggest that the author may have not properly reported the data (difference between a table and a diagram); the author has verified and confirmed that this is not the case. I checked a few points and saw no difference between table and plot. I will specifically ask the reviewers to refrain from making such comments in further reviews."

A: I had no problem with this comment. It seemed to me to be a minor, unintentional oversight by the reviewer (or by myself – I cannot say for sure that there was no discrepancy). I have much greater concern over the reviewer's accusation that my work is akin to data fabrication and his refusal to honor the GSA Ethical Guidelines for Publication in his role as a reviewer for a previous version of this paper. More generally, I am concerned about potential conflicts among some of the reviewers that may prevent a fair and impartial judgement of my paper. I respectfully ask the AE and Editor to consider these concerns.

[revised manuscript text omitted]
_{\text{av}}}{1-(S_{\text{av}}/S_\text{c})^2} = \frac{L}{\kappa} \cancel{P_{\text{0,pred}}} P_{\text{r,pred}} \exp\left(-\frac{h_1}{h_0 S_{av}^b}\right)$$

(10)

Given a map of steepness obtained by solving equation (10), soil thicknesses and erosion rates can be mapped using equations (7) and (8), respectively. Note that the $S_{\text{av}}$ value obtained by solving equation (10) is not a prediction in the usual sense, since $S_{\text{av}}$ is an input to eqn. (10) via $\cancel{P_0}\underline{P}_{\text{r,pred}}$. The model can be considered to capture the effects of topographic steepness if the predicted and observed values of $S_{\text{av}}$ have broadly similar absolute values and patterns of spatial variation.

[revised manuscript text omitted]

15  Heimsath et al. (2012) argued that $P_r$ values (analogous to what they termed $SPR_{max}$ values) increase with erosion rates not just in the SGM, but globally based on the strong correlation between $P$ and $E$ values (their Fig. 4b). However, the results of this paper suggest that  it is slope that controls $P_r$ values, not erosion rate. Slope and erosion rate are highly correlated in the SGM, but this correlation is not universal. The results of this paper also suggest that the process  by which slope leads to an increase in $P_0$ values  in the SGM (i.e.,

20  topographically induced stress opening of fractures) is likely not operative in 
[revised manuscript text omitted]

---

## Author Response (AR3)

I appreciate the value that the reviewers and AE have found in the revised version of the paper, which demonstrates the likely role of topographically induced stress on potential soil production rates in the SGM. I thank everyone for all their hard work, which I think has made for a much improved paper.

Q: *"Please explain what the "average slope" as defined by Heimsath means, i.e. how it is calculated and whether this is an appropriate measure to relate to local stress intensity (all three reviewers comment on that point)"*
A: *Regarding the first issue, i.e., whether the average slope in the model is equivalent to the average slope as calculated by Heimsath et al:*

It is clear from the context of their work that the average slope calculated by Heimsath et al. is an average of the gradients of hillslopes surrounding each sample location over a spatial scale that includes ridgetops and side slopes. I don't know precisely what spatial scale Heimsath et al. used for averaging, but it appears to be ~1 km. This is corroborated by patterns in the data, e.g., there are several clusters of samples (e.g., SG-105 to 108, SG-110 to 113) with individual data points separated by less than approximately 1 km that have precisely the same average slope value.

I emailed Arjun to ask for additional information on how he computed average slope. He noted that his calculation used only hillslope patches (valley bottoms were excluded) and graciously agreed to check his notes and get back to me with more details. By the time of the revision deadline (including an extension to the deadline), I had not heard back from him. When I receive his reply I will add the information he sends if there is still time.

In any case, I believe the average slope I use in the modeling is equivalent to the average slope computed by Heimsath et al. (2012). As stated in the manuscript, the average slope in the model is computed from the ridgetop to the steepest point in the model geometry. In the SGM, as in any region of narrow, V-shaped valleys, the steepest portion of the hillslope tends to occur at or near the base of the slope. Heimsath et al. computed average slopes from the ridgetops to the base of slopes (valley bottoms were excluded) over a spatial scale that included ridgetops and sideslopes. As such, the calculations are consistent.

I have added the following text on this issue:

"The average slope computed from the model geometry is consistent with the average slope computed by Heimsath et al. (2012). The average slope in the model is computed from the ridgetop to the point of maximum slope in the model geometry. In the SGM, as in any region of narrow, V-shaped valleys, the steepest portion of the hillslope tends to occur at or near the base of the slope. Heimsath et al. (2012) computed their average slope from hillslope patches (valley bottoms were excluded) over a length scale that included ridgetops and side slopes. As such, the calculations are consistent."

*Regarding the second issue, i.e., whether average slope is the appropriate variable for quantifying local stress intensity:*

I tried hard in the previous version of the paper to emphasize that local stress intensity is a function of both slope and curvature. However, the reviewers are correct that more clarity was needed on this issue. I have included the following text in the revision:

"It is important to note that the local stress modification in the Savage and Swolfs (1986) model is a function of both the local curvature and the slope averaged over a spatial scale that

includes ridgetops and side slopes. Within an individual hillslope, local curvature controls the sign of stress modification, with extension occurring beneath ridgetops and compression beneath valley bottoms. The extension that occurs beneath ridgetops is the most important response of the model for the purposes of this paper since all of the Pr data come from locations at or near ridgetops (i.e., 24 of the 58 data points are on ridgetops, with the remaining data points located within approximately 100 m from ridgetops). The magnitude of the extension near ridgetops is controlled by the landscape-scale slope (quantified by Savage and Swolfs (1986) as $b/a$), the slope averaged over a length scale that includes ridgetops and side slopes is the variable most consistent with $b/a$."
And to the discussion section I have added:

"Savage and Swolfs (1986) used a convex-concave geometry, defined by a conformal transformation, in which the slope increases linearly with distance from the divide to the steepest point on the hillslope. In higher-relief portions of the SGM characterized by more planar hillslopes, slopes increase abruptly over a relatively short distance from the ridgetop, then more slowly with increasing distance from the ridgetop. This difference introduces some uncertainty into the application. The model might overestimate the magnitude of topographically induced stress in high-relief portions of the SGM because a more planar slope has a lower curvature than a more parabolic slope and larger curvatures tends to increase extensional stress. On the other hand, more planar hillslopes localize curvature near the ridgetops, which might tend to increase bending stresses that drive extension over and above that predicted by the model for locations near ridgetops."

Q: "*Please respond or take into account the second point made by Reviewer 1 who disputes your argument that the relationship between slope and production rate is due to extensional stress fracturing*"
A: Please note that I did not argue that the relationship between slope and production rate is due to extensional stress fracturing. Rather, I stated that the rocks in the SGM are pervasively fractured and that extension can open up those pre-existing fractures. See my responses to reviewer 1 for detailed responses to his/her concerns.

Q: "*Could you easily check whether hill orientation with respect to the regional steps field direction plays (or not) a role in setting production rate (through stress fracturing) as suggested by Reviewer 2; this is a good test of your hypothesis and, in my opinion, would not require much work to verify*"
A: I did not find a relationship between Pr values and local slope aspect (except for the tendency for lower Pr values to occur on north-facing slopes compared to south-facing slopes, which remains unexplained but could be related to the fact that south-facing slopes are steeper, on average, compared to north-facing slopes (see figure below)) or the orientation of the closest ridgeline.

This might appear to be evidence against the hypothesis of the paper. However, bear in mind that a topographic profile along the principal stress direction that runs through any sample location (even a sample location with a local ridgeline nearly parallel to the principal stress direction) will exhibit substantial topographic variability (increasing with average slope) since ridgelines do not run in straight lines. That is all that is required for compressive stress reduction

near ridgetops, i.e., that the sample location be in a relatively high topographic position in a region of moderate to high relief along the direction of principal stress. Topography is (to some extent) fractal, such that the local ridge-and-valley profile is embedded within larger-scale topographic variations, all of which contribute to the topographically induced stress (although local scales are more important because average slope and curvature values, which control the magnitude of compressive stress reduction, decrease with increasing spatial scale). Further complicating any simple test of how Pr values might correlate with the orientation of the nearest ridgeline with respect to the principal stress direction is the fact that simply identifying which is the local ridgeline can be unclear, as sample locations often sit in saddles between two ridgelines that run in perpendicular directions (see example below).

[Figure]

Figure. Demonstration that south-facing slopes are, on average, steeper than north-facing slopes in the SGM. (A) Shaded relief image of the only publically available lidar for the SGM. (B) Plots of the slope-area relationship derived from the DEM in (A), with south- and north-facing slopes considered separately.

[Figure]

Figure. Demonstration of some of the complexity involved in trying to test how the orientation of the hillslope and/or nearest ridgeline with respect to the principal stress direction might control Pr values. In this case, the sample location exists within a saddle between a prominent ridgeline that runs SW-NE and a smaller (but closer) ridgeline that runs W-E.

Q: "*Please pay attention to the various remarks concerning the applicability of the Savage and Swolfs relationship to the study area, as well as the use of a correct measure of slope (see my first point. Reviewer 3 makes very constructive points concerning the way you have analyzed the data; the Reviewer's suggestions should help clarify your approach and help supporting your interpretation. Reviewer 3 also makes a very interesting suggestion concerning the important section 2.2 of your manuscript; I urge you to consider this point carefully and, if possible, implement his suggestion(s) in your revised manuscript.*"

A: See responses below. Note that in some cases I did not agree with the suggestions of reviewer 3, but in all such cases I have provided very specific reasons for not doing so, which I hope the AE finds compelling.

**Reviewer 1 (denoted reviewer 3 in system):**

Q: *"P2 line 2: "relatively uncommon in grantitc rock types." This needs a citation."*
A: Rephrased: "Slope failures in bedrock or intact regolith are common in some fine-grained sedimentary rocks (e.g., Griffiths et al., 2004; Roering et al., 2005) but may be less common in massive lithologies such as granite."

Q: *"P3, line 4-5: This is strange wording. $P_r$ is the limit to soil production, and earlier it is stated that this limits erosion. Here $P$ and $E$ are said to greatly exceed $P_r$. I'm sure this statement us being used to highlight that $P_r$ is in fact not the limit to soil production but I feel there should be a phrase or some rewording that makes this intention (if that is the intention) clear. A simple insertion of "apparent $P_r$" might do the trick."*
A: The sentence does not state that P and E greatly exceed Pr values. The sentence states that P and E values *from rapidly eroding portions of the range* exceed Pr values *from slowly eroding portions of the range*, which is correct as stated. Pr values from one area do not necessarily limit soil production in other areas, because Pr values can and almost certainly do vary spatially.

Q: *"P5, line 10-12: Not all ridges in the field site are oriented perpendicular to the most compressive stress direction. There are even a few oriented parallel. How does this affect the model prediction? I'm curious if one might expect a different signal depending on the orientation of the ridgelines. Note: I doubt one could see something like that in the data given the noise, but I think it is worth commenting upon."*
A: See response to AE's request for a similar test.

Q: *"P5, lines 13-15: The Savage and Swolfs paper is quite dense but they impose a particular geometry to their ridges. It isn't clear to me if the ridges they model have geometry similar to the ridges in the field area, and thus it isn't clear if the relationship between their maximum slope and average slope can be meaningfully applied to the field site. Their ridges look somewhat Gaussian (they are not, but they are convexo-concave), whereas in the San Gabriels the hillslopes have a short wavelength convexity at the top and are linear on the side slopes. Some comment should be made about how the Savage and Swolfs model is an approximation of real topography and how uncertain the stress field is of real landscapes is as a result of differences between real landscapes and their idealised landscape. P5, line 15: While I do think converting an average slope into a specific slope in the Savage and Swolfs model is something of an approximation, I do find this connection between the switch from compression to tension fascinating."*
A: I agree with the reviewer that hillslopes in the high-relief portion of the SGM have a short wavelength convexity at the top and are more linear on the side slopes than the model geometry of S&S. However, lower-relief portions of the SGM (which are also represented in the dataset) have hillslopes that are more nearly parabolic, and hence more similar to the Savage and Swolfs geometry. I have addressed this concern with the following text: "Savage and Swolfs (1986) used a convex-concave geometry, defined by a conformal transformation, in which the slope increases linearly with distance from the divide to the point of maximum slope. For the specific mathematical model of Savage and Swolfs (1986), the average slope computed from the ridgetop to the point of maximum slope is equal to $b/4a$. In higher-relief portions of the SGM characterized by more planar hillslopes, slopes increase abruptly over a relatively short distance from the divide, then more

slowly with increasing distance. This difference introduces some uncertainty into the application. The model might overestimate the magnitude of topographically induced stress in high-relief portions of the SGM because a more planar slope has a lower curvature than a more parabolic slope, and larger curvatures tends to increase the magnitude of compressive stress reduction. On the other hand, more planar hillslopes localize curvature near the ridgetops, which might tend to increase the bending stresses that drive compressive stress reduction near ridgetops (where all of the sample locations come from) over and above that predicted by the model."

Q: "*P6, line 3: Something I do not understand is that in the Savage and Swolfs paper, the horizontal and shear stresses vary substantially as a function of position (e.g., their figure 4). There is not much variation in $\sigma_{xx}$ in figure 3 here. If these stresses vary horizontally (from ridgetop to channel) then if these stresses are affecting soil production rates they should do so to different extents on different parts of the hillslope, should they not? I suggest some clarification here.*"
A: Agreed. In the revised paper I have modified the text in several places to clarify that compressive stress reduction is maximized under ridgetops.

Figure 3 of Savage and Swolfs varies more substantially because their figure includes the entire shape of the ridge-valley transect, while I include only the ridge portion (because all of the Pr data are from near-ridge locations and because valleys in the SGM are V-shaped, not U-shaped as Savage and Swolfs assume).

Q: "*P6, line 17: Does this model do better than a simple linear fit? Again, I think this is an interesting approach but the model described by equation (4) has 4 parameters that must be fit to the data (and this excludes modifications for climatic influences).*"
A: I have demonstrated in the revision that the equation is more accurate than a linear fit, even accounting for the larger number of parameters (using a reduced chi-squared measure). Sentence added: "The null hypothesis that Pr,S values can be fit as well or better by a linear relationship can be rejected: the reduced-χ2 value, which takes into account different numbers of degrees of freedom, of the log-transformed values of equation (5), is less than half (45%) of the reduced-χ2 for a least-squares linear fit."

Q: "*Page 12 lines 6-7: Clunky sentence. I suggest rewriting.*"
A: Reworded: "Stepwise regression is the process of computing the residuals of a regression and testing for additional controls, via additional regression and the calculation of a new set of residuals, until no additional explanatory variable can be identified. Stepwise regression is one method for testing the residuals of a regression for additional controls, which is a recommended step in all regression analyses."

Q: "*Page 13 line 9-11: The thresholding behaviour is particularly interesting in light of the results presented earlier in the paper and it would be useful to specifically mention the previous authors in addition to Savage and Swolfs that found this behaviour.*"
A: I don't know of any other studies that specifically found a threshold relationship between slope and some measure of topographically induced stress other than Savage and Swolfs.

Q: "*Page 13, lines 16-17: I'm not sure if this sentence is a reflection of the content of the paper. The paper does not show stresses cause fractures which then lead to enhanced weathering. What it shows is that previous models of topographically-induced stresses suggest transitions from compressive to tensile strength at hillslope angles similar to those at which $P\_r$ values increase. This study doesn't present fracturing data. I think this section needs more cautious language because at the moment it is describing processes that aren't really addressed in the paper.*"

A: The manuscript merely stated that the results *suggest* that stresses cause fractures which then lead to enhanced weathering. I think this is correct as written. However, I have adopted the reviewer's suggestion and modified this sentence to: "The results presented here show that previous models of topographically induced stresses suggest transitions from compressive to tensile strength at hillslope angles similar to those at which Pr values increase. This similarity suggests that in the SGM, the release of compressive stress in steep landscapes may cause fractures beneath ridges to open, thereby allowing weathering agents to penetrate into the bedrock or intact regolith more readily."

**Reviewer 2 (4 in system):**
Q: "*It is not clear to me that the extensional stresses that are being invoked are occurring where the soil production rates are being measured. The average slope is used in both this study and in Heimsath et al., 2012. Average slope is measured as the average from the ridgetop to the maximum slope here, but I am not sure that this is how Heimsath et al., measured it ("the average slope over hillslopes adjacent to each sample location" is not clear. More importantly, the Savage and Swolfs model suggests that the maximum extension should be occurring on the ridgetop above the steep slopes, not on the steep slopes themselves (e.g. page 13 line 16). It seems that some of Heimath et al.'s data were indeed collected from ridgetops, but some samples came from the steep hillslopes below them. The implication of the S&S model is that the maximum extension (and hence maximum Sr in this approach) should occur on ridgetops and not on steep hillslopes. Since ridgetops are, by definition, less steep than the surrounding hillslopes, this would negate the assertion that soil production rates are fastest on steep slopes because of extensional stress. It is possible that this is just a question of definitions, but if so then it needs to be explained further.*"
A: Please see response to AE's comment #1.

Q: "*The correspondence of EVH to Pr/Pr,S is not convincing. The similarity is only marginal and is controlled by four points at high elevation. Related to this, I am not sure that it is appropriate to say that temperature limitations can be determined. The climatic index (equation 5) is somewhat arbitrary and could be expanded upon.*"
A: Without more information from the reviewer it is difficult to know how my analysis could be improved to make it more convincing to him/her. I think it is highly significant that vegetation height shows a similar "hump" to that of Pr/Pr,S, although I acknowledge that the humps are offset by approximately 300 m. Some differences between the curves are to be expected due to the fact that EVH is influenced by the recent fire history, which temporarily reduces EHV in locations that have experienced fire in recent decades. Still, over a 1-km range of elevations, both EHV and Pr/Pr,S exhibit broadly similar increases and then decreases that suggest a causal connection

between vegetation cover and weathering rates. The alternative hypothesis (i.e., that Pr values are unrelated to vegetation cover) does not seem more plausible to me.

The results of the cluster analysis demonstrating climatic control at high elevations of the SGM are statistically significant even though they are based on just four data points (i.e., they are significant because all four points are so much lower than the Pr/Pr,S at lower elevations). The reviewer claims that this result is marginal. Is he/she suggesting that I adopt a significance level higher than the standard threshold of 95% or does he/she just not believe the result?

Regarding the reviewer's statement that my manuscript concluded that temperature limitations on Pr can be determined, please note that the manuscript stated only that the results were likely (not definitively) the result of temperature limitations (except for a somewhat firmer statement in the abstract that has been reworded). See also my response to reviewer 3 on this point.

Q: "*One of the other main issues is the assertion of a causal link between fracture opening and soil production (end of section 2.1). I do not accept that the results show a causal relationship between hillslope gradient and soil production rate through extensional stress fracturing (or reactivation). The higher R2 for Pr to Pr,pred than Pr to D only shows that the topographic stress fracture model provides a better correlation than fault damage model, but it could be that a third, unaccounted for, variable is the causal link. It is, however, an interesting thought experiment that is shown to be consistent with existing soil production data. If presented in such a format, Pelletier could present predictions, providing a powerful tool for guiding future soil production rate studies. This would involve restructuring the paper around the concept of topographic stress fracturing and the implications for the real world. Much of this is already included. It would be particularly useful for the community a set of testable model predictions were presented.*"

A: This concern is echoed by reviewer 3 (Simon Mudd), who suggested a rewording of Page 13, lines 16-17 to soften/clarify the conclusions. I addition to rewording on p. 13, I have carefully checked the manuscript to make sure that I have not claimed a definitive causal relationship between hillslope gradient and Pr values via topographical stress fracture opening (rather, the results are merely consistent with this hypothesis). The reviewer is correct that there could be some third control, related to slope but unrelated to topographically induced stress. However, in geosciences we always face the problem that a correlation between two variables that we explain by some mechanism (one that, in this case, has a solid foundation in theory) could, instead, be due to some other mechanism. Absent any suggestion of an alternative mechanism from this or any other reviewer, I think I have done the best I can (i.e., I cannot disprove all possible alternative hypotheses, particularly if none are identified).

Q: "*I am confused by the sentence on lines 3-4 of page 11. If the SGM is characterised by an exponential soil production function, then the maximum soil production rate (and presumably at least some erosion) should occur when there is no soil cover. If that is the case, then there is no need to presume that soil ever existed at the h=0 sites.*"

A: The distinction I am making is between persistent soil cover (soil that can be measured on any given day by a visit to the site) and episodic soil cover (soil that forms and persists only as long as it takes for it to be transported down the slope, which may be only seconds). As stated in the manuscript, soil (i.e., mobile debris) must be present in order for erosion to occur. This is clear

from the fact that immobile regolith, by definition, does not erode physically (i.e., it is immobile). In order for the landscape to erode, some mobile debris must be present, though it may only be present for seconds until rockfall or some other type of mass wasting moves it down the slope. I have tried to clarify this point by rewording the sentence to: "Here I use a soil-depth-independent transport relation because such models are highly sensitive to the presence/absence of soil and areas of thin or no soil are likely to have episodic cover (e.g., rapid mass wasting following incipient soil production) that makes measuring or estimating long-term averaged soil depths difficult."

Q: "*Terminology is not consistent throughout. Both Pr and P0 are used.*"
A: Fixed.

Q: "*Pg 14 line 5 – Should say "If soil production rates…".*"
A: Typo corrected.

**Reviewer 3 (5 in the system):**
Q: "*In the entire manuscript, both soil and regolith seems to be interchanged. It would be good to choose one of them and explain why because they are not the same.*"
A: I am confused by this comment, which notes that the two terms are not the same yet asks me to choose just one. Weathered material above fresh bedrock is comprised of *intact regolith* and *soil*. Both terms must be used in the context of describing the process of fresh bedrock breakdown into soil because intact regolith is the intermediate state between the two. Please note that the word regolith never appears in the manuscript without the modifier "intact" in front of it.

Q: "*p2: line 10: I would rather say: between 100 and 2500m/Myr.*"
A: Removed.

Q: "*Pelletier proposed an alternative approach to analyze soil production rates obtained from CRN data. Rather than regressing the soil production function through all the data, the residual Pr is calculated separately for all the measured P values using h0 values as obtained from the regressions in the Heimsath et al. paper in 2012. I see no harm in this approach to test for additional controls on SPR. However, even in this revised manuscript, at some points I still feel uncomfortable with the way these Pr data are further analyzed.*"
A: Calculating residuals *requires* first regressing the soil production function through all the data (as a first step). Therefore, I disagree with the reviewer my analysis is an alternative to regressing the soil production function through all the data. As stated in the previous round of review, computing the residuals and testing for additional controls is a recommended step in regression analysis and is performed after the initial step of obtaining the regression formula. What I am doing is not an alternative to regression but a recommended part of it.

Q: "*Please add an explanation similar to "To compare this metric with soil production, Pelletier calculates P0 from every data point by regressing the soil production function, using a slope of h0 previously regressed in the Heimsath et al paper, to its h = 0 intercept.*"

A: I prefer to leave the explanation as it is, since I am confused by what is meant by "regressing a function to its intercept."

Q: "*Eq. 2: why not write it with conventional units (cm) to comfort the reader i.e.: Pr= Pe0.031h*"
A: Writing it in the way the reviewer suggests would involve units of $cm^{-1}$, not cm. I have a preference for defining scale parameters in terms of units of length or time rather than their inverse (e.g., $\exp(-h/h\_0)$ rather than $\exp(-\alpha h)$) because I think it is easier to understand length or time units. Of course if the reviewer or AE insists on this change I will make it.

Q: "*I guess the author has reasons for it, but I am wondering why he is not plotting all the Heimsath data, calculate one soil production function from it, and use the thereby derived h0 value for the remaining part of his analysis. Actually, by using the two h0 values for Sav<30 and Sav>30 as derived by Heimsath et al., an a priory assumption is already made that weathering is higher for steeper regions. Isn't it the main goal of this paper to illustrate this using the residual values and could this point not be made stronger using a single soil production function and the therefrom derived h0 value? It is indeed remarkable that the 0-soil depth samples were excluded from the regressions presented in the Heimsath 2012 work. Including them and redrawing the SP function for S>30 shows more or less no trend raising the relevant question whether the data of SGM even support the exponential soil production function at S>30°. Therefore, I find it remarkable that the author uses the h0 value derived from the SPR function of this S>30° observations. A question which couples back to my previous remark.*"
A: I don't see how using two different $h\_0$ values for Sav<30 and Sav>30 necessarily implies that weathering is higher for steeper regions. First, the two $h\_0$ values differ by only about 10%. Second, $h\_0$ values define how quickly soil production rates fall off with increasing soil thickness – they do not define the absolute values of soil production rates (these are set by $P\_0$ or Pr).

I disagree with the reviewer's contention that plotting all of the data for Sav>30 reveals no trend. The data exhibit a hump, with the means of the clusters with h=0 and h>15 cm lower than the cluster with 0<h<15. Treating the data with h=0 separately allows (but does not require) a humped production function. It would be OK to consider all of the data at once, but given that the data exhibit a hump it would only be acceptable to do this if one were to consider a mathematical formula consistent with a humped production function. Absent any general agreement on the mathematical form that governs the humped production function, I think fitting the data points with h>0 separately from those with h=0 is the most defensible approach.

Q: "*Page 5, line 3: should be 1.78Pr , I guess...*"
A: 1.78*P* is correct as stated. As with the previous round of review, I do not understand why reviewers would suggest that I write Pr = 1.78Pr, which is mathematical nonsense.

Q: "*Overall, it is not very clear to me how exactly Sav is calculated and whether Sav used in the equations (eg. Eq. 2 versus Eq. 3) is rather "Sav is defined by Heimsath et al. (2012) as the average slope over hillslopes adjacent to each sample location." as mentioned on page 4 or "expressed in terms of the average slope from the drainage divide to the location of maximum slope rather than the shape parameter b/a used by Savage and Swolfs (1986). " as mentioned on page 5.*"

A: See responses to AE and other reviewers on this point.

Q: "*Page 5, Line 23: Why not inserting the equation of Savage and Swolfs (1986) in your text? That would increase overall readability of the paper.*"

A: Done. Added: "Savage and Swolfs (1986) studied role of topography in modifying local stresses in a model ridge and valley geometry that uses a conformal transformation that includes length scales *b* and *a* that define the vertical and horizontal extents of the ridge, respectively. Because the data from Heimsath et al. (2012) are acquired from locations at or near ridgetops, I focused only on the portion of the Savage and Swolfs (1986) solution between the ridgetop and the point of maximum slope, i.e., the broad, U-shaped valley bottoms flanking the central ridge were not considered. The average slope, $S_{av}$, computed from between the ridgetop and the point of maximum slope, is equal to $b/4a$ in the mathematical framework of Savage and Swolfs (1986). A key result of Savage and Swolfs (1986) is their prediction of a gradual decline in the horizontal compressive stress near ridgetops as $b/a$ increases between 0 and 2 (their Figure 4) based on their equation (36):

$$\frac{\sigma_{xx}}{N_1} = \frac{2-b/a}{(2+b/a)(1+b/a)} \tag{3}$$

where $N_1$ is the regional maximum compressive stress and $S_{av}$ has units of m/m in equation (3). Substituting $4S_{av}$ for $b/a$ in equation (3) yields:

$$\frac{\sigma_{xx}}{N_1} = \frac{2-4S_{av}}{(2+4S_{av})(1+4S_{av})} \tag{4}$$

Note that the tangent of the slope angle (units of m/m) is averaged to obtain $S_{av}$ in all cases in this paper. However, after this averaging, $S_{av}$ is reported in degrees in some cases to facilitate comparison with the results of Heimsath et al. (2012)."

Q: "*Page 6, Line3: "Gravitational stresses can be included", are they also included in the analysis? If not I suggest leaving the following paragraph out or at least summarizing it.*"

A: I added an explicit statement that gravitational stresses are not included. I think it is crucial that I include at least a quantitative discussion of how including gravitational stresses would modify the results, which is what the following paragraph does. I agree that this paragraph is somewhat dense and technical, but I feel strongly that this discussion is necessary.

Q: "*Eq. 4, it could be helpful for the readers to color the different line segments of Fig. 2 as calculated under the three conditions.*"

A: I have added text in the caption to make this point: "The piece-wise curve plots equation (5), with the three segments of the curve corresponding to the three conditions in the equation."

Q: "*If I understand the procedure correctly, I am afraid that I do not support the way in which the z>2300m data points are being treated and I feel this is an issue which should be resolved (or better explained in case I misunderstand this). Eq. 4 (Pr) is calculated excluding the >2300m data points. Next, the same Pr values are used to compare the >2300 m points from the <2300 points. Of course, there is a difference in the mean value of the difference between the predicted Pr value and the data points of the z>2300 points because they were not included when calculating Pr. This procedure makes no sense as such. Maybe a simple variance analysis could do the trick?*"

A: The reviewer is obviously correct that the means of any two populations will not be precisely identical. However, a t test for unequal means tests whether or not two populations are sampled from *statistically distinct sets* as measured by the similarity of their mean values and taking into account the variances within each population. Two populations can certainly have (and always will have) means that are not precisely identical, but the test I employed determines whether or not the two populations represent statistically distinct sets (for example, comparing rolls of the dice between two sets of dice, one of which might be loaded or biased). To address this issue I have reworded the sentence: "Assuming a significance level of 0.05, the null hypothesis that the cluster of blue points is sampled from the same statistical set as that of the remaining points with Sav > 30° (i.e., that both sets are governed by the same process or controlling variables) can be rejected based on the standard t test with unequal variances (t = 0.021)."

Q: "*Because of my previous remark, I feel troubled with Eq. 5. If temperature and vegetation cover is indeed controlling weathering rates as proposed by Pelletier, why not including a temperature factor or vegetation factor directly into Eq. 5 rather than using a discrete proxy for climate. Has the author tested such approach? I find too much weight is now given to a so called cluster of only 4 residual points in order to prove the importance of climate.*"
A: I worked hard to include a vegetation factor into the analysis but in the end I could not find a definitive answer. Primarily this is because vegetation cover depends on the recent fire history of a mountain range (areas that have burned recently have vegetation cover unrelated to temperature and precipitation) in addition to factors such as temperature and precipitation. Moreover, there are many ways of quantifying vegetation (via type, canopy height, above-ground biomass, etc.) and each gives somewhat different answers. As for temperature, the only spatially distributed temperature data available for the SGM are interpolations (such as PRISM or WorldClim) that are based almost entirely on elevation-based regressions. As such, I think it is more straightforward and honest to use elevation directly as a controlling variable rather than a modeled variable (temperature) that is based heavily on elevation.

Q: "*Of the entire paper, I find this section least convincing. Here the author stretches his empirical findings to the limit, based on some questionable assumptions (eg. assuming a steady state between uplift and erosion is far less evident than often assumed (Mudd, 2016)) and a weak correlation between soil depth and Sav. I see what the author tries to do in Figure 7 but find a visual comparison and evaluation of the obtained results not a good evidence for the proposed theory. I would find a simple model exercise, at a 2D profile or a landscape scale, very helpful here (e.g. with one of the many LEMs the author developed in previous work). In such an exercise the author could investigate whether or not, the adapted soil production rates, depending on topographically induced stress, does indeed allow to reproduce increasing weathering rates with simultaneously increasing erosion rates (without direct control of erosion on weathering rates). If such a trend could be observed, this would indeed offer a valid and alternative theory to explain the observation of Heimsath 2012 that weathering rates increase with increasing erosion rates. If the author wants to keep his current approach, I strongly recommend some more quantitative approaches to interpret the findings. Of special interest could be a plot comparing predicted erosion rates with measured catchment wide denudation rates (which are available for this region: Dibiase, 2010).*"

A: Please note that the steady state I assumed in section 2.2 is a soil thickness steady state, not a topographic steady state as the reviewer states. In fact, I made a special point of noting that the modeling in section 2.2 makes no assumption regarding uplift rates. A soil-thickness steady state condition is assumed in all *in situ* CRN calculations and is likely to be more widely applicable than a topographic steady state (see Heimsath et al., 2002 for a discussion).

Section 2.2 is useful because it demonstrates how the various correlations identified using individual data points plays out across the landscape at larger spatial scales. I think there is value in this. Moreover, this section demonstrates the role that soil thickness plays, which is not explicitly considered in Section 2.1. Soils always tend to thin in areas of steeper slopes, thereby allowing the soil production rate to increase in concert with the erosion rate. Now that Heimsath et al. (2012) have demonstrated that in the SGM potential soil production rates also increase with slope, this begs the question of whether (or how much) soils still thin as slope steepens. The analysis of Section 2.2. demonstrates that this type of thinning still occurs, just not to the extent that it would were it not for the positive correlation between Pr values and average slope.

In the revision I have compared the model-based erosion rates with CRN-based catchment-averaged erosion rates. I thank the reviewer for this excellent suggestion. The added text is as follows:

[revised manuscript text omitted]